# QuArch: A Benchmark for Evaluating LLM Reasoning in Computer Architecture

## Abstract

The field of computer architecture, which bridges high-level software abstractions and low-level hardware implementations, remains absent from current large language model (LLM) evaluations. To this end, we present QuArch (pronounced 'quark'), the first benchmark designed to facilitate the development and evaluation of LLM knowledge and reasoning capabilities specifically in computer architecture. QuArch provides a comprehensive collection of 2,671 expert-validated question-answer (QA) pairs covering various aspects of computer architecture, including processor design, memory systems, and interconnection networks. Our evaluation reveals that while frontier models possess domain-specific knowledge, they struggle with skills that require higher-order thinking in computer architecture. Frontier model accuracies vary widely (from 34% to 72%) on these advanced questions, highlighting persistent gaps in architectural reasoning across analysis, design, and implementation QAs. By holistically assessing fundamental skills, QuArch provides a foundation for building and measuring LLM capabilities that can accelerate innovation in computing systems.

## 1 Introduction

Benchmarks that elicit reasoning are among the most impactful large language model (LLM) evaluations today since they require critical thinking that goes beyond surface-level knowledge and pattern matching. As a result, state-of-the-art (SoTA) progress is tracked on benchmark suites requiring multi-step reasoning, where models with explicit test-time deliberation (i.e., "thinking" variants) consistently climb leaderboards. Widely adopted datasets such as GSM8K (Cobbe et al., 2021), AIME (Mislav Balunović, 2025), SWE-bench (Jimenez et al., 2024), GPQA (Rein et al., 2023), and MMLU-Pro (Wang et al., 2024a) probe this structured reasoning to serve as proxies for measuring math, software engineering, and natural and physical science expertise.

Reasoning is equally central to *computer architecture*, which emphasizes evaluating trade-offs within a multi-objective optimization design space. For example, computer architects decide how to organize and balance components of systems (e.g., compute, memory, interconnects) and their power, performance, and area trade-offs. However, computer architecture remains an area without any LLM benchmarks to date.

Existing benchmarks in computing systems target engineering tasks for software or chip implementation such as code generation (Jimenez et al., 2023; Yang et al., 2024; OpenAI, 2024; He et al., 2025), register-transfer level (RTL) generation (Liu et al., 2023b; Pinckney et al., 2025b), system-on-chip (SoC) integration (Alvanaki et al., 2025), and chip verification (Wan et al., 2025). While these are important, they primarily evaluate whether a model can produce or manipulate programmatic artifacts, not whether it can reason about the principles that guide design decisions. Computer architecture plays a different role in the computing stack: it serves as the vital interface between software and hardware to define how these complex pieces interact, where careful orchestration of system components and their trade-offs *informs and influences implementation*. These decisions rely on conceptual understanding and analytical reasoning that is guided by application workloads and technology trends, rather than just code synthesis. Importantly, the skills required by architects to navigate these multi-objective design space problems can be systematically evaluated through a question–answering (QA) paradigm.

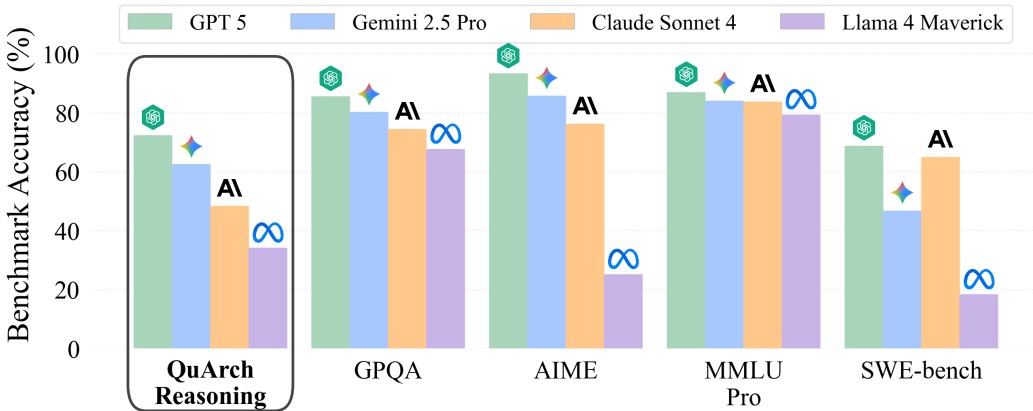

Figure 1: Reported results (Vals AI, Inc., 2025) for different models across QUARCH and multiple other SOTA benchmarks to date.

To this end, we introduce QUARCH: a question-answering benchmark to assess the architectural knowledge and reasoning capabilities of LLMs required in computing systems design. Figure 1 presents reported performance of frontier models across other reasoning domains in comparison to QUARCH, demonstrating that reasoning models are not yet able to solve advanced architecture questions. This gap underscores the need for focused evaluation on architectural reasoning to translate LLM progress into agentic methodologies that can accelerate innovation in computing systems.

QUARCH aims to capture the broad domain knowledge and skill set that architects possess by constructing the benchmark around four foundational competencies: Recall, Analyze, Design, and Implement. Existing computing systems benchmarks primarily target technical implementation skills (Table 1), but all four competencies—recalling foundational principles, analyzing workloads and constraints, designing systems that balance objectives, and implementing solutions via code—are complementary and critical for effective architecture design. Although we focus on computer architecture, these skills are broadly applicable to many systems tasks. Our framework enables systematic evaluation that holistically measures an agent's ability so that model progress can be measured with fine-grained skills and compared over time.

In summary, our work makes the following contributions: ❶ **QUARCH is the first benchmark designed to evaluate advanced computer architecture knowledge and reasoning in LLMs and is comprised of 2,671 expert-validated QAs.** 1,124 questions were curated through academic crowdsourcing and community competitions, and 1,547 questions were synthetically generated and human-verified. ❷ **To promote holistic evaluation of AI agents for systems tasks, we formalize a skills framework to systematically assess 10 frontier models on QUARCH.** Our evaluation reveals that even flagship LLMs today struggle with skills requiring higher-order thinking. Notably, QUARCH uncovers a significant performance gap between LLMs' architectural knowledge and reasoning abilities. ❸ **We conduct an in-depth analysis to offer key insights and observations on model trends and failure points**. This includes incorrect architectural assumptions made, difficulties with modeling system state, absence of architecture-semantics in code execution, and heterogeneity in LLM topic expertise. ❹ **We establish a trustworthy and scalable methodology for evaluating the correctness of free-form responses in QUARCH by comparing LLM judgments with human domain-expert verdicts across 100 QAs and 10 frontier models.** We show that LLM judgments agree with human experts at a rate of 85.35%, which is comparable to human-human grading agreement rates of 90.7% on the QUARCH benchmark.

## 2 QUARCH

### 2.1 TOWARDS AGENTIC SYSTEMS DESIGN: BUILDING AN AI ARCHITECT

**Skill Requirements.** To systematically assess progress towards agentic design of computing systems, we first introduce a conceptual framework to decompose the fundamental skills that computer

Table 1: ML benchmarks & datasets across the computing stack. QUARCH broadens the scope of current benchmarks by focusing on conceptual and analytical reasoning skills required for computer architecture and systems design. Benchmarks above QUARCH target more software-oriented tasks, while those below focus on more hardware-centric, chip design tasks.

| Benchmark & Dataset for Computing Systems | Focus in Computing Stack | Conceptual & Analytical QA | Design QA & Program Impl. | Multimodal Assessment | Expert Verified | Benchmark Size |
|---|---|---|---|---|---|---|
| SWE-bench (Jimenez et al., 2023) | Software Eng. | ✗ | ✓ | ✗ | ✗ | 2294 |
| SWE-bench Verified (OpenAI, 2024) | Software Eng. | ✗ | ✓ | ✗ | ✓ | 500 |
| SWE-Perf (He et al., 2025) | Performance Eng. | ✗ | ✓ | ✗ | ✗ | 140 |
| KernelBench (Ouyang et al., 2025) | Performance Eng. | ✗ | ✓ | ✗ | ✗ | 250 |
| CodeMMLU (Nguyen et al., 2025) | Code Reasoning | ✓ | ✗ | ✗ | ✗ | 19912 |
| CRUXEval (Gu et al., 2024) | Code Reasoning | ✓ | ✓ | ✗ | ✗ | 800 |
| **QUARCH (This Work)** | **Architecture** | ✓ | ✓ | ✓ | ✓ | **2671** |
| SLDB (Alvanaki et al., 2025) | System Design | ✗ | ✓ | ✗ | ✓ | 10 |
| CreativEval (DeLorenzo et al., 2024b) | HW Design | ✗ | ✓ | ✗ | ✗ | 120 |
| VerilogEval (Liu et al., 2023b) | RTL Generation | ✗ | ✓ | ✗ | ✗ | 156 |
| CVDP (Pinckney et al., 2025b) | RTL Generation | ✓ | ✓ | ✗ | ✓ | 783 |
| MG-Verilog (Zhang et al., 2024) | RTL Generation | ✗ | ✓ | ✗ | ✗ | 11000 |
| EDA Corpus (Wu et al., 2024a) | EDA Tooling | ✓ | ✓ | ✗ | ✓ | 1533 |
| FIXME (Wan et al., 2025) | Verification | ✗ | ✓ | ✗ | ✗ | 180 |
| ChiPBench (Wang et al., 2024b) | Layout | ✗ | ✓ | ✗ | ✗ | 20 |

architects (and systems engineers more broadly) require. Figure 2 illustrates these skills: within a single problem scenario, we exemplify how different styles of QAs exercise different skills, from recall of fundamental domain knowledge to more advanced analysis, design, and implementation.

*Recall*: Retrieving domain knowledge, definitions, and facts. *"What information is stored in a branch target buffer (BTB) to verify that a branch target address is a match?"* This includes the ability to identify components and roles in a diagram or specification such as standard digital logic elements. Critically, domain knowledge underpins advanced reasoning (Krieger, 2004; Duncan, 2007).

*Analyze*: Deducing, inferring, calculating, or interpreting data and information from a scenario to reason about workload implications and system behavior. Identifying bottlenecks and being able to explain "why" is critically important for deeper understanding. *"If the branch predictor and BTB make the same predictions, which will give better performance?"*

*Design*: Proposing, inventing, or improving an architectural feature (method, component, or policy) while satisfying system requirements and constraints. It requires balancing nuanced performance, power, area, and cost trade-offs. Synthesizing a design often can require iterating over architectural block diagrams and system specifications. *"Suggest a dynamic branch predictor that integrates into the processor's pipeline."*

*Implement*: Translating a design into executable artifacts (e.g., code/RTL/simulation scripts). Typically, this skill is used to validate a solution via modeling or measurement. *"Implement the dynamic branch predictor and BTB in a simulator and run performance benchmarks."*

Crucially, all of these skills are significant pillars exercised in different scenarios at different times by architects and systems engineers, with domain knowledge being the foundation upon which other higher-order skills can be built. For example, without first knowing the basics of how processor execution, memory hierarchy, concurrency, parallelism, and communication work, it is difficult (if not impossible) to reason about design and performance trade-offs within a complex multi-core system.

**Knowledge Breadth Requirements.** Computer architecture contains a multitude of specialized areas. Historically, architects focused on microprocessor design but expanded towards many-core systems and domain-specific accelerators (Blake et al., 2009; Dally et al., 2020) due to memory and power walls (Wulf & McKee, 1995; Esmaeilzadeh et al., 2011). Consequently, this elevated the importance of understanding memory systems, interconnects, and system-level methodology (Sangiovanni-Vincentelli, 2007; Carloni, 2015) to first-class concerns. *Effective architecture and system design requires understanding relationships across these areas and reasoning about how they interact.* For example, a processor aggressively optimized without considering the connected memory subsystem will exhibit more performance bottlenecks than if the two were optimized together. Thus, a benchmark should capture topic breadth to properly assess architectural knowledge.

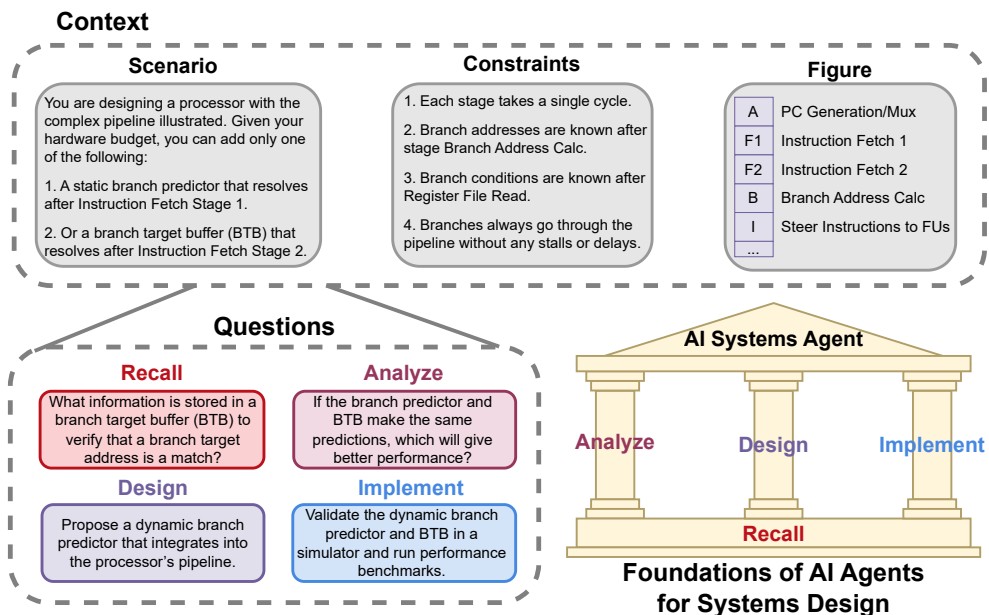

Figure 2: QUARCH QA Skills Framework. The benchmark evaluates four complementary competencies: Recall, Analyze, Design, and Implement. QAs in QUARCH contain relevant context describing the scenario, constraints, and figures when appropriate. The illustrative example shows how distinct question styles derived from the same context can probe different skills. Together, these competencies form the building blocks required for systems design in computer architecture. Balanced mastery of skills will enable more effective agents and workflows across the computing stack.

## 2.2 BENCHMARK CONSTRUCTION

Curating a computer-architecture benchmark is particularly difficult because high-quality, openly usable sources are scarce relative to other domains (Reddi & Yazdanbakhsh, 2025) and authoring or validating benchmark entries requires substantial domain expertise to ensure technical correctness. We adopt a three-pronged strategy that combines synthetic data generation, academic exams, and expert crowdsourcing and competitions (Fig. 3). Every QA is reviewed and expert-validated by doctoral students with graduate-level training in computer architecture before inclusion in QUARCH.

**Synthetic Data Generation.** We collected open-source materials to curate a large corpus of computer architecture knowledge spanning technical manuals, academic publications, and comprehensive online resources. This corpus reflects a diverse and thorough survey of publicly available knowledge in the field and serves as a foundation for QUARCH. Using this corpus, LLMs generated cloze-style multiple-choice QAs (Rogers et al., 2023) to balance educational value with practical assessment. QAs then underwent two-stage validation: LLM-as-a-judge (Zheng et al., 2023) for initial filtering (as these cloze-style QAs naturally involve little reasoning) followed by independent review of each QA by three experts. This approach enabled the identification and removal of questions lacking definitive answers or those too narrowly scoped for meaningful assessment. Prompt details are in Appendix D.6 and D.7, .

**Expert Crowdsourcing & Competitions.** We developed a web-based portal specifically for crowdsourcing architectural reasoning questions to target more advanced analysis, design, and implementation skills that are difficult to synthetically generate. QAs were collected via an open submission platform for individuals with technical backgrounds and time-boxed competitions. Similar to other recent benchmark curation methodologies such as Humanity's Last Exam (Phan et al., 2025), the interactive portal provided exemplary reasoning examples and real-time feedback on submitted questions to encourage participants to submit challenging questions and a solution rationale (Ap-

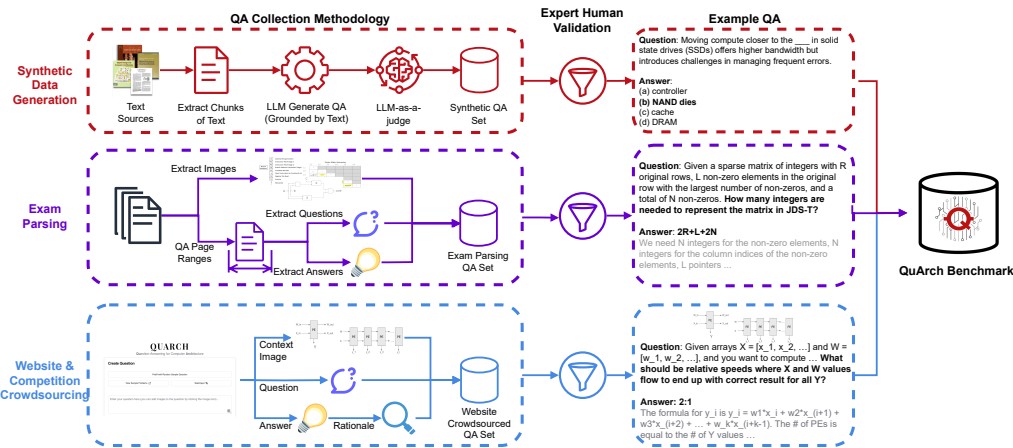

Figure 3: We construct QUARCH with a three-pronged approach including a blend of synthetic data generation, community crowdsourcing, and academic exams. All QAs are validated by a human expert to curate QUARCH's final benchmark set of 2,671 question-answer pairs.

pendix B.1). The individual submissions and competition submissions underwent expert review to check for ambiguity and correctness before final acceptance.

**Academic Exams.** We additionally curated QAs from university computer-architecture exams obtained via our community crowdsourcing process and manual web scraping. A custom pipeline was developed to convert PDFs into standalone QAs. Llamaparse was first used to extract diagrams (LlamaIndex, 2025). An LLM then segmented the exam into per-question PDFs to decompose the large exam PDF and parse each QA into context, question, and solution fields. To verify parsing, QAs underwent similar two-stage validation as our synthetic data generation process that employed LLM-as-a-judge for initial filtering followed by expert review. This pipeline yielded exam-level, multimodal QAs suitable for benchmarking. Prompt details for this pipeline are in Appendix D.8, D.9, and D.10.

## 2.3 BENCHMARK CHARACTERIZATION

We characterize QUARCH's 2,671 QA pairs along architecture topics, skill focus, question format, and modality, establishing a framework for fine-grained tracking of benchmark growth over time.

**Architecture Topic Diversity.** QUARCH captures diverse topics in 13 core areas derived from key themes in modern computer architecture research (Figure 4). Processor architecture accounts for the largest proportion of QAs (37%), followed by memory systems (25%) and interconnection networks (8%). This distribution mirrors the field's current and historical emphasis, with niche areas containing fewer QAs. Appendix C provides example QAs that show the breadth and depth of topics covered in QUARCH. The topic distribution was estimated via two-stage classification using a text embedding model and LLM labeling (Appendix D.5).

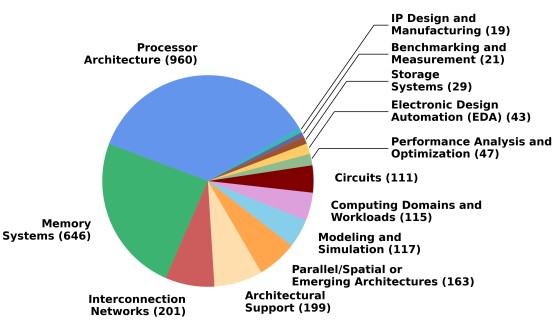

Figure 4: Distribution of topics in QUARCH.

**Skills Coverage.** Figure 5 characterizes QAs by the skills in Sec. 2.1, derived from LLM labeling (Appendix D.4) with examples for each given in Appendix C.1. In particular, QUARCH targets advanced reasoning by providing nearly 1000 analysis QAs and ∼100 design & implementation

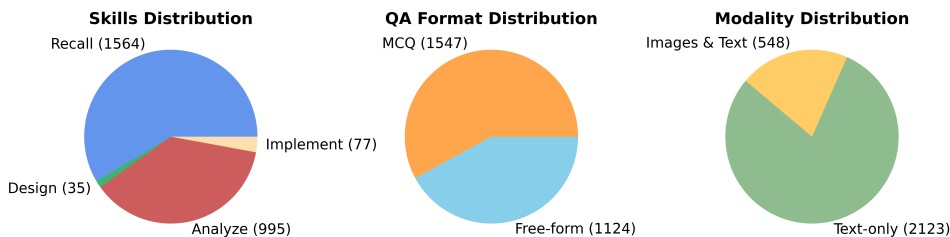

Figure 5: Breakdown of skill, format, and modality distributions in QUARCH.

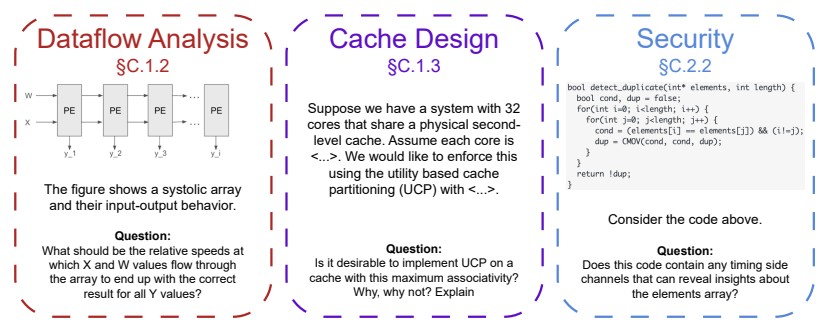

Figure 6: Example Questions from QUARCH. Full question details in Appendix C.

QAs. This disparity in number reflects the intrinsic difficulty of authoring and validating design & implementation QAs, mirroring their natural frequency in our sources (e.g., exams typically have few design QAs relative to analysis). We term *recall-focused* QAs QUARCH-RECALL and *higher-order skill* QAs (e.g., analyze, design, implement) QUARCH-REASONING.

**Question Format and Input Modalities.** In line with prior QA and code-reasoning benchmarks (Rein et al., 2023; Hendrycks et al., 2020; Nguyen et al., 2025), QUARCH includes 1,547 multiple-choice questions (MCQs), which are amenable to synthetic generation (Section 2.2) and have clear evaluation criteria. However, academic evaluation of domain-expert architecture knowledge is highly open-ended in structure, requiring deeper critical thinking, and thus cannot be formulated as MCQs. QUARCH therefore includes 1,124 free-response questions (FRQ), with examples in Appendix C. Furthermore, QUARCH contains both 2,123 text-only and 548 multimodal (images & text) questions. These multimodal examples assess image interpretation and reasoning capabilities on structured and spatial information, such as architecture datapath diagrams, circuit schematics, pipeline timing charts, roofline performance plots, and specification tables.

### 2.4 EXAMPLE QUESTIONS

Figure 6 presents sample questions from QUARCH. As shown, questions in QUARCH exercise a variety of practical skills core to computer architecture. The examples include designing dataflow accelerators, configuring memory systems, and identifying hardware vulnerabilities of low-level code. Fully expanded versions of these questions, along with others, can be found in Appendix C.

## 3 EXPERIMENTAL SETUP

**Models.** We evaluate ten frontier models from Alibaba Cloud, Anthropic, DeepSeek, Google, Meta, Mistral, and OpenAI on QUARCH across the four skills (Recall, Analyze, Design, Implement) presented in Section 2.1. Evaluation results for an additional 16 models are reported in Appendix B.4.

**Evaluation grading.** All models are evaluated in a zero-shot setting. Full evaluation prompts are provided in Appendix D. For MCQ-style responses, models must conclude with the correct choice of A, B, C, or D. For FRQ-style questions, we employ LLM-as-a-judge (Zheng et al., 2023), tasking an external model to assess the correctness of an answer with respect to the ground truth. We further motivate and rigorously validate our use of LLM-as-a-judge in Section 4.5.

Table 2: Frontier model performance on QUARCH. Reported values are the per-generation accuracy across 3 generations. All models struggle much more on QUARCH-REASONING compared to QUARCH-RECALL. We highlight the **first**, **second**, and **third** best performing models.

| Model | QUARCH-RECALL | QUARCH-REASONING | Δ |
|---|---|---|---|
| *Multimodal Models* | | | |
| GPT-5 | **89.0** | **72.4** | **-16.5** |
| GPT-5 (Non-Reasoning) | **86.3** | 49.0 | -37.3 |
| Gemini 2.5 Pro | **87.4** | **62.9** | **-24.5** |
| Gemini 2.5 Flash | 83.4 | **56.8** | **-26.6** |
| Claude Sonnet 4 | 85.5 | 48.4 | -37.0 |
| Claude 3.7 Sonnet Thinking | 85.8 | 52.1 | -33.7 |
| Llama 4 Maverick | 85.3 | 34.2 | -51.1 |
| Mistral Medium 3.1 | 84.5 | 34.1 | -50.4 |
| *Text-Only Models* | | | |
| GPT-OSS 120B | 84.2 | 64.7 | -19.5 |
| DeepSeek R1 | 86.9 | 56.1 | -30.7 |

**Metrics.** For both MCQ- and FRQ-style questions, we report model performance using "per-generation accuracy", the percentage of correct answers received out of $n$ total responses, to provide a better estimate of the *pass@k*=1 metric (Pinckney et al., 2025a) under stochastic generation parameters. In all evaluations, we generate $n = 3$ samples per question.

## 4 EVALUATION & ANALYSIS

### 4.1 MODEL PERFORMANCE

Table 2 reports headline accuracy of 10 frontier models on QUARCH. As defined in Section 2.3, QUARCH-REASONING covers higher-order skills in our framework, while QUARCH-RECALL captures domain-knowledge retrieval (rather than reasoning). QUARCH-RECALL performance is consistently strong across all frontier models. Including a recall split is useful to establish a baseline in a field that lacks a dedicated benchmark: the split distinguishes "don't know" from "can't reason," and informs whether fundamental domain knowledge is present in models. Unlike frontier models, current Small Language Models (SLMs) exhibit gaps on recall performance (Appendix B.3). Overall, the reasoning variant of GPT-5 leads on QUARCH today with Gemini models forming the next tier. We note that GPT-OSS 120B and DeepSeek R1 are evaluated only on text (no images), so their scores reflect text-only capability. We focus the rest of our analysis on QUARCH-REASONING because it offers the most headroom for today's frontier models to improve.

### 4.2 SKILL PERFORMANCE TRENDS

Table 3 provides fine-grained skill-wise performance across models, with key trends shared below:

**(1) Recall is Mastered, Higher-Order Skills are Not.** Frontier models have largely mastered recall, but fall short on advanced skills. Recall accuracy ranges between 83%-89%, suggesting architectural knowledge is present. However, analyze, design, and implement skills are lower than recall by 32.1%, 39.2%, and 38.2% on average respectively. Notably, multiple models with strong recall accuracy fall below 35% on other skills (e.g., Llama 4 Maverick, Mistral Medium 3.1). In particular, design skills exhibit the widest performance gaps, ranging between 18-87%. This suggests the design QAs, despite comprising a small proportion of the dataset, provide a strong discriminative signal that correlates with performance across the full QUARCH benchmark.

**(2) Reasoning Matters for Advanced Skills.** Results from (1) indicate that translating domain knowledge into advanced skills will require targeted training and test-time deliberation mechanisms. Comparison of GPT-5-thinking with its non-reasoning variant supports this. GPT-5 outperforms by 23%, 34%, and 30% on analyze, design, and implement QAs respectively, compared to a much smaller 3% lift on recall QAs. Moreover, GPT-5 stands as an extreme outlier on design QAs at 89%, suggesting its training corpora and test-time reasoning budget better capture the skills needed to select and justify design decisions under constraints compared to other frontier models.

Table 3: Per-generation accuracy (%) by QuArch Skill. Best performing models in each category highlighted **first**, **second**, and **third**.

| Model | Recall | QUARCH-REASONING | | |
|---|---|---|---|---|
| | | Analyze | Design | Implement |
| *Multimodal Models* | | | | |
| GPT-5 | **89.0** | **72.1** | **86.7** | **71.0** |
| GPT-5 (Non-Reasoning) | **86.3** | 49.5 | 52.4 | 40.7 |
| Gemini 2.5 Pro | **87.4** | **63.3** | **59.0** | **59.7** |
| Gemini 2.5 Flash | 83.4 | **57.3** | **57.1** | **49.8** |
| Claude Sonnet 4 | 85.5 | 49.0 | 36.2 | 46.8 |
| Claude 3.7 Sonnet Thinking | 85.8 | 53.3 | 34.3 | 45.5 |
| Llama 4 Maverick | 85.3 | 35.1 | 18.1 | 30.2 |
| Mistral Medium 3.1 | 84.5 | 34.8 | 27.6 | 27.7 |
| *Text-only Models* | | | | |
| GPT-OSS 120B | 84.2 | 65.5 | 56.1 | 57.8 |
| DeepSeek R1 | 86.9 | 57.4 | 38.6 | 47.1 |

**(3) Variations in Competencies Across Models.** Our skill framework exposes model-specific strengths and failures that a single aggregate score hides. Overall, 23 of the 26 models evaluated (Appendix B.3) score lower on either analysis or design QAs than on implementation QAs, underscoring the need to target all of these higher-order skills to holistically assess architecture competency.

## 4.3 KEY INSIGHTS & OBSERVATIONS

Results on QUARCH illustrate a clear gap in model capabilities. Based on extensive grading performed by experts (Section 4.5), we synthesize key insights into specific failure modes observed.

**(1) Struggles with architecture-semantics of code execution.** Architectural semantics of code execution refers to the deep understanding of how high-level code interacts with the underlying hardware architecture (e.g., memory access patterns, instruction scheduling, etc.) (Tschand et al., 2025). Our analysis reveals that LLMs struggle with these nuanced aspects of code execution, failing to accurately predict or analyze the architectural implications of code snippets (Appendix C.2).

**(2) Assuming unconventional architectural properties.** In computer architecture, decades of practice have cemented certain system designs, such as byte-addressable memory, as de facto defaults unless otherwise specified. However, our analysis exposes a misalignment: when prompts fail to state conventions explicitly, we observe LLMs defaulting to unconventional choices, such as word-level addressing (Appendix C.3). Models are able to succeed when provided with the default conventions explicitly, highlighting that practitioners leveraging LLMs in this domain must identify their implicit assumptions to guide the model effectively.

**(3) Modeling and tracking system state.** Building an intuition and mental model of how system components interact and the implications of their interactions is central to computer architecture. In general-domain QA, this corresponds to situational world modeling (Rogers et al., 2023): instantiating entities, tracking their locations and states, and inferring temporal and causal relations to answer queries about an evolving scenario. We find that frontier models often fail to maintain consistent system state and thus misunderstand how local actions cascade into system-level effects on latency, throughput, and correctness (Appendix C.4).

**(4) Variations in domain expertise.** Our analysis reveals that LLMs develop specialized expertise across different domains. For instance, within "Implement"-Style questions, Llama 4 Maverick performs well on Computing Domains & Workloads and struggles on Modeling & Simulation, while Mistral Medium 3.1 exhibits the opposite behavior. Importantly, Llama 4 Maverick and Mistral Medium 3.1 overall performed similarly on "Implement"-Style questions, suggesting model capabilities are more nuanced than the aggregate scores of Table 3. These findings provide the opportunity to create multi-model systems that combine the domain strengths of multiple LLMs rather than relying on a single "best" model. Spider plots visualizing these per-topic variations for all frontier models across all skills are shown in Appendix B.2.

**(5) Sensitivity to QA modality.** In computer architecture, visuals such as pipeline diagrams, cache hierarchies, and system interconnects convey structural relationships and spatial information that cannot be adequately captured through text descriptions alone (Chang et al., 2024a). Multimodal models exhibit an average 6% drop in accuracy from text-only free-response to image-only free-response questions (Appendix Table 6). This gap indicates that frontier models struggle with interpreting and reasoning about diagrams, schematics, and tables (see Appendix C.5 for failure examples).

## 4.4 CASE STUDY: DESIGNING A MEMORY CONTROLLER

To illustrate how QUARCH-REASONING performance translates to real architectural decision-making, we conduct a case study on Dynamic Random Access Memory (DRAM) controller design, a canonical target of architecture design space exploration tasks (Krishnan et al., 2023). In this task, the model must propose DRAM controller configurations (e.g. scheduling policies, refresh behavior, etc.) that would lead to energy efficient designs.

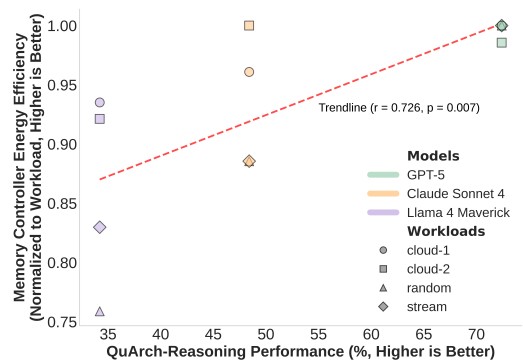

Figure 7: Correlation between QUARCH-REASONING and DRAM controller design quality. Each point is a model-workload pair, with the x-axis showing QUARCH-REASONING accuracy and the y-axis showing normalized memory-controller energy efficiency (power$^{-1}$).

The prompt lists (Appendix D.11) all design parameters and explicitly asks the model to analyze the workload's memory trace and reason about how the controller design choices affect power consumption before proposing a concrete configuration. **Critically, (1) effectively answering these questions requires architectural reasoning and influences optimal design choices; and (2) similar QAs about memory trace analysis and memory controller configurations are assessed in QUARCH (Appendix C.7).** Designs are evaluated using a DRAM simulator (Jung et al., 2015).

We run 10 independent rounds per model across 4 different workloads under a fixed evaluation budget (40 proposed designs per round). We evaluate three models with varying QUARCH-REASONING performance: GPT-5, Claude Sonnet 4, and Llama 4 Maverick. Figure 7 reports the mean energy efficiency (power$^{-1}$, normalized within each workload) of each model's best controller designs across the 10 runs versus their QUARCH-REASONING performance.

We observe a strong, statistically significant positive correlation between QUARCH-REASONING accuracy and normalized energy efficiency ($r = 0.726$, $p = 0.007$), indicating that models which reason better on QUARCH also discover more energy-efficient configurations. This correlation is important, as general QA benchmarks (e.g. MMLU) do not always correlate with downstream-task-focused agentic benchmarks (e.g. SWE-Bench) (Qin et al., 2025). This case study finds that **improvement on QuArch directly reflects improvement on computer architecture tasks.**

## 4.5 LLM-AS-A-JUDGE ANALYSIS

**Motivation.** Semantically equivalent and correct solutions to the same FRQ can differ in phrasing, as shown in Appendix C.6. Since full manual grading by domain experts is intractable, we employ LLM-as-a-judge for QUARCH.

**Human Validation.** While LLM-as-a-judge has gained popularity for evaluating FRQ-style questions (Lee et al., 2024; Zhou et al., 2023; Mañas et al., 2024; Pinckney et al., 2025b), the approach is still relatively new. We therefore validate the fidelity of LLM-as-a-Judge by measuring agreement rates between human expert and LLM judge verdicts on the correctness of generated FRQ responses. We randomly sampled 100 (8.9%) freeform QAs in QUARCH, and generated one response from the 10 models under evaluation in Sec. 4.1. We tasked a cohort of 11 domain experts

in computer architecture and hardware design to manually grade the resultant 908 responses[1] as CORRECT, PARTIALLY-CORRECT, or INCORRECT. For analysis purposes, PARTIALLY-CORRECT is recategorized as INCORRECT. Each question is graded independently by up to 3 experts and the majority consensus is taken. To control for LLM judge stochasticity, all judge evaluations in Sec. 4.1 and Appendix B.4 are likewise performed 3x and the majority vote taken.

**LLM judges agree with human experts.** We observe an agreement rate of 85.35% between LLM judges and humans (Fig. 8), when using Claude 3.7 Sonnet as the judge. We compare this agreement rate with the rate that expert humans disagree on verdicts. 84 of the 908 responses required a third expert to adjudicate between a correct and incorrect vote, corresponding to a human-to-human agreement rate of 90.7%. Since this agreement rate is broadly comparable to the frequency with which LLM-as-a-Judge consensus agrees with human consensus, we argue LLM-as-a-Judge is eminently suitable for scalable and informative benchmarking of model performance on QUARCH. Additional experiments on human expert grading difficulty, alternative LLMs as judges, and majority consensus rates are included in Appendix B.6

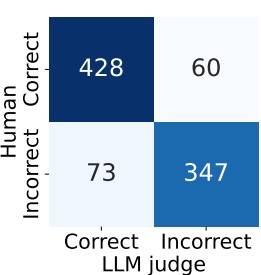

Figure 8: Confusion matrix comparing LLM-as-a-Judge with domain-expert human grading on FRQ.

## 5 RELATED WORK

**Software.** Function-level code efficiency benchmarks (Du et al., 2024; Huang et al., 2024; Shypula et al., 2024; Waghjale et al., 2024) and domain-focused performance tasks (Press et al., 2025; Ouyang et al., 2025) evaluate correctness-preserving edits and runtime gains at the function or kernel level. Repository-scale SWE benchmarks and agentic toolchains (Jimenez et al., 2024; Yang et al., 2024; Wang et al., 2025) test long-horizon code manipulation and integration. Recent QA code understanding benchmarks (Gu et al., 2024; Nguyen et al., 2025; Li et al., 2024; Dinella et al., 2024) target control/data-flow semantics, behavioral equivalence, and code review comprehension. Unlike QUARCH, which primarily targets pre-implementation, system, and architectural judgment, these works focus on code artifacts, assessing code semantics rather than system-level design.

**Hardware.** Domain-specific foundation models for chip design (Liu et al., 2023a), electronic design automation (EDA) tool interaction (Wu et al., 2024a;b), RTL generation (Chang et al., 2024b; Thakur et al., 2024; Liu et al., 2024; Blocklove et al., 2023), design optimization (Chang et al., 2023; Pei et al., 2024; DeLorenzo et al., 2024a), and security-oriented tasks (bug repair and assertions) (Tsai et al., 2024; Yao et al., 2024; Fu et al., 2023; Pearce et al., 2023; Nair et al., 2023; Meng et al., 2024; Mali et al., 2024) emphasize producing or improving implementation artifacts and driving tools. In contrast, QUARCH isolates the reasoning that guides implementation, testing whether models can reason about architectural principles and trade-offs, rather than their ability to generate HDL/RTL or steer EDA flows.

**QA benchmarks.** General-purpose and domain QA datasets (Rajpurkar et al., 2016; Trischler et al., 2016; Clark et al., 2019; Hendrycks et al., 2020; Rein et al., 2023; Huber et al., 2022; Jin et al., 2019; Cobbe et al., 2021; Zhong et al., 2020) have been instrumental for advancing and measuring LLMs (Rogers et al., 2023). QUARCH targets advancing computer architecture specifically, with expert-verified items and skill-wise evaluation capabilities not covered by existing QA benchmarks.

## 6 CONCLUSION

We introduce QUARCH, the first benchmark to directly assess computer architecture knowledge and reasoning in LLMs across four complementary skills: Recall, Analyze, Design, and Implement. Evaluating ten frontier models on 2,671 expert-validated QAs, we find consistently strong recall across models but reveal a pronounced gap in higher-order abilities that demand architectural reasoning. By providing insights into failure modes and enabling systematic tracking, QUARCH lays the groundwork for accelerating AI progress in computer architecture and, more broadly, in reasoning-centric skills for systems design.

---

[1]Non-multimodal models cannot generate responses for the multimodal proportion of sampled questions.

## ETHICS STATEMENT

QUARCH was curated from sources that permit academic use and redistribution. Synthetic items were generated from a domain corpus compiled from public materials, exam-derived items were collected from publicly accessible university course pages or contributed by instructors, and crowd-sourced items were submitted through our portal with explicit contributor consent. All expert validators who participated in question review and acceptance are co-authors of this paper. We did not recruit paid crowd workers; when individuals submitted questions via our portal, they consented to inclusion under our dataset license and to public attribution (or opted to remain anonymous). We do not collect personally identifying information beyond optional contact details for acknowledgment. No student data or private repositories were used. Where third-party figures or excerpts are included, we respect the original licenses and provide attribution. We will honor takedown requests for any inadvertently mislicensed content. This project did not involve human-subject experiments or interventions and, to the best of our understanding, does not require IRB oversight.

## REPRODUCIBILITY STATEMENT

Section 2.2 describes the methodology for constructing QUARCH that can be used to reproduce a dataset of similar quality and characteristics. Appendix D.8, D.9, and D.10 each expand on the details of the methodology overview provided in Section 2.2. Additionally, exact prompts used for evaluation results are documented in Appendix D.1, D.2, D.3, and D.4 for reproducibility.

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

APPENDIX

# Table of Contents

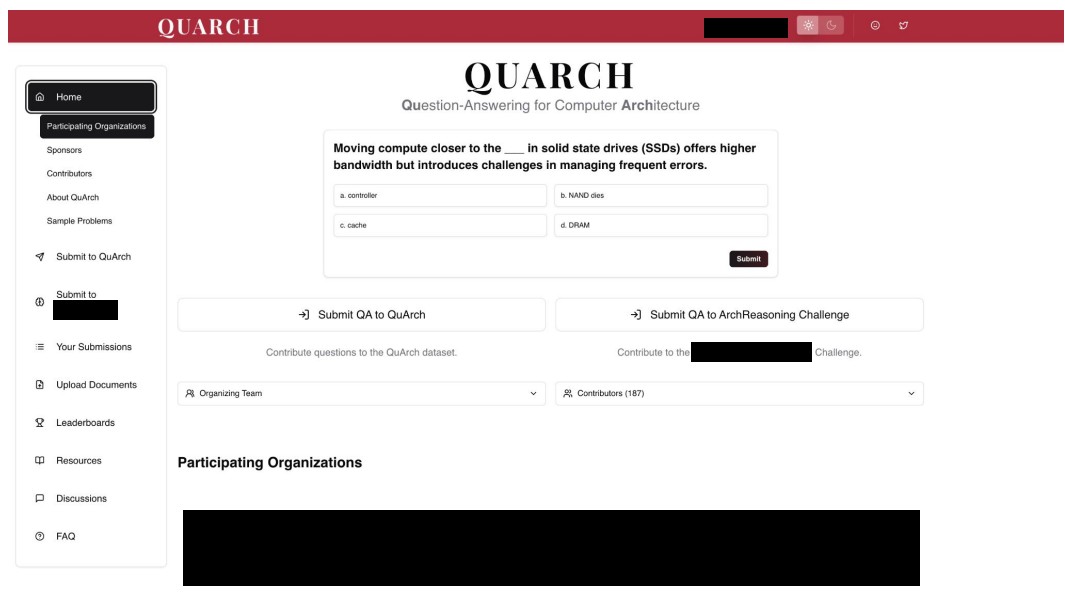

Figure 9: QuArch Website Home Page

## A    LLM USAGE

Language models were employed to refine the prose (e.g., grammar, clarity, and style) and to check formatting compliance with venue guidelines. Apart from their explicit roles described in the paper, namely for synthetic QA generation, exam parsing assistance, and evaluation (LLM-as-a-Judge), LLMs were not used to originate substantive scholarly content. All benchmark content admitted to the final release was verified by domain experts, and all prompts used in construction and evaluation are reported in Appendix D.

## B    ADDITIONAL RESULTS

### B.1    QUARCH CROWDSOURCING WEBSITE

To provide a centralized location to crowdsource questions and exams, we created a QuArch website, shown in Figure 9. When a user wants to submit a question, they are presented with a set of instructions to guide accurate, relevant, and formatted questions, shown in Figure 10. Users submit the question, four answer options, the correct answer option, and a rationale for the correct answer, as shown in figure 11. While the user-submitted question is formatted by default in multiple-choice style, we instruct them to not submit questions where the correct answer is an "all of the above" or "none of the above." This enables us to concatenate the correct answer with the rationale and format the question as a free-response as well.

Once a question is written, we enable users to seamlessly test four non-frontier LLMs on correctness, shown in Figure 12. The question and potential answer choices is sent to the respective model within the MCQ Prompt shown in Appendix D.2. Users are presented with whether the model gets the question correct and its response. We explicitly do not want to give the users access to test their question on frontier LLMs to prevent them from overfitting to only specific LLMs. By giving users signal on whether non-frontier LLMs fail, they can create difficult and correct questions that

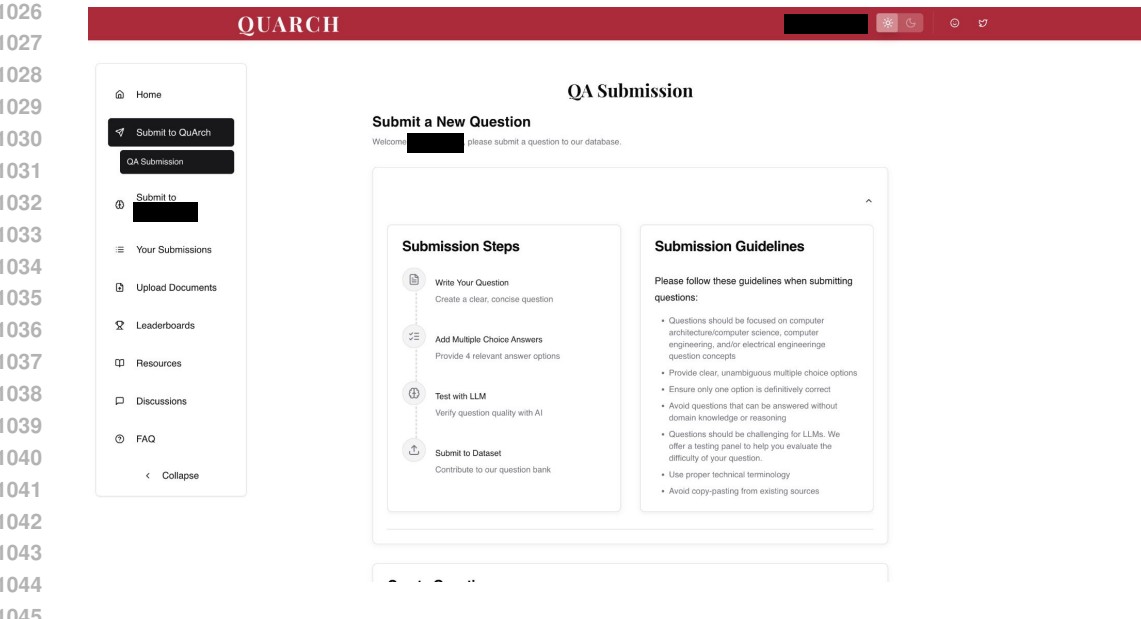

Figure 10: QuArch Website Submit Question Instructions

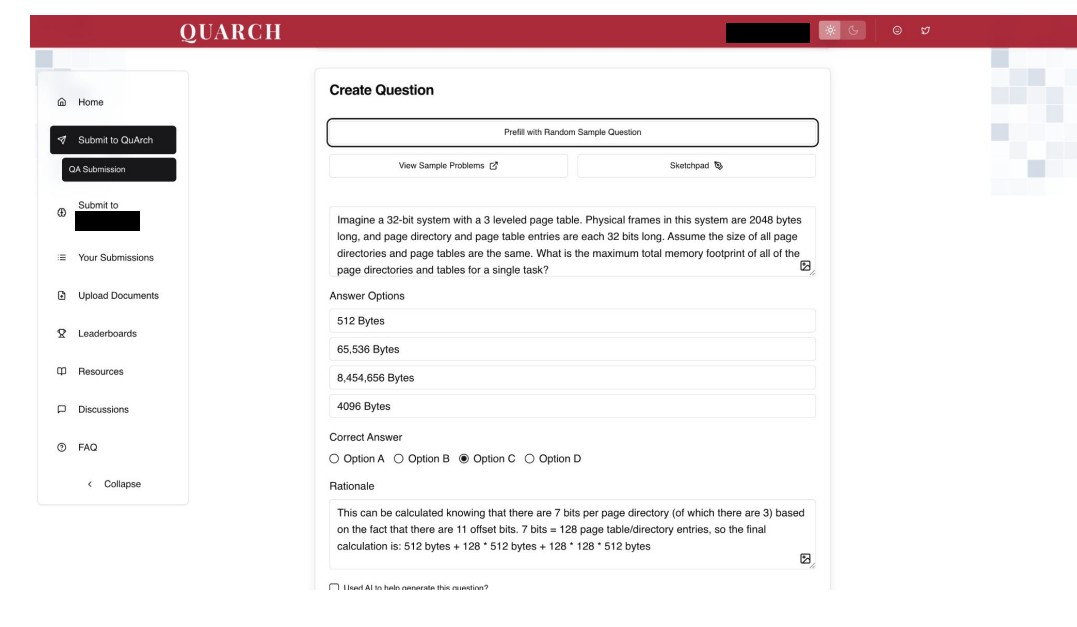

Figure 11: QuArch Website Submit Question Portal

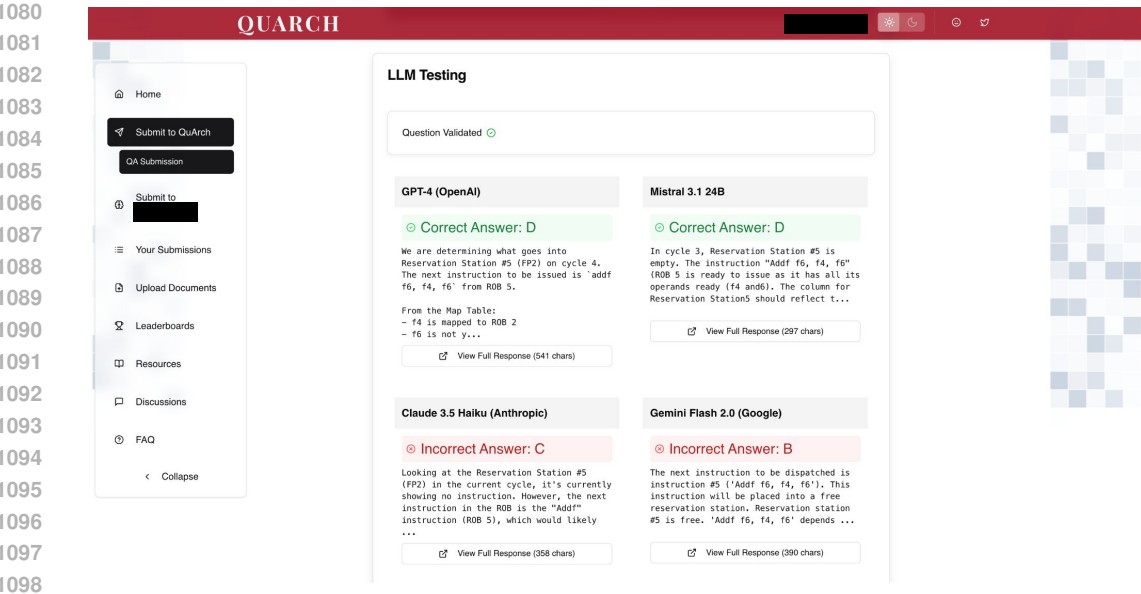

Figure 12: QuArch Website LLM Testing on Question Submission

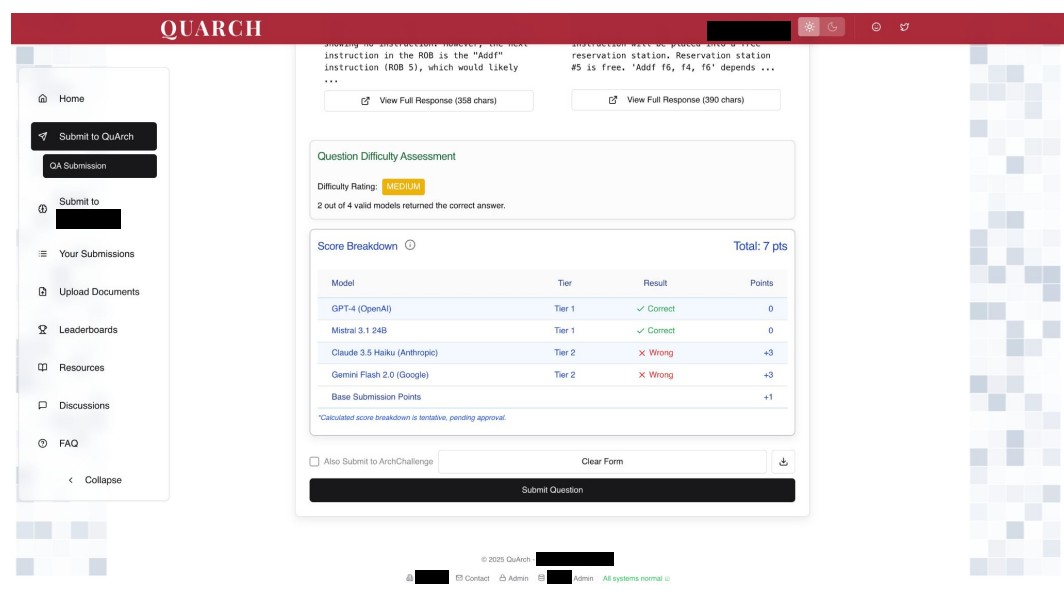

Figure 13: QuArch Website Question Submission Scoring

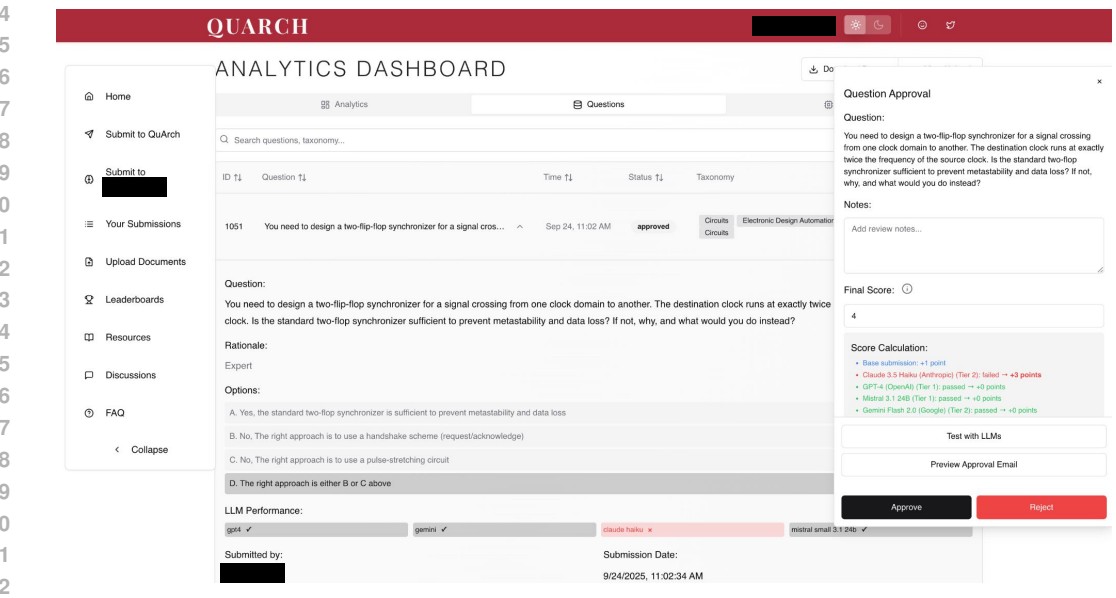

Figure 14: QuArch Website Admin Approvals

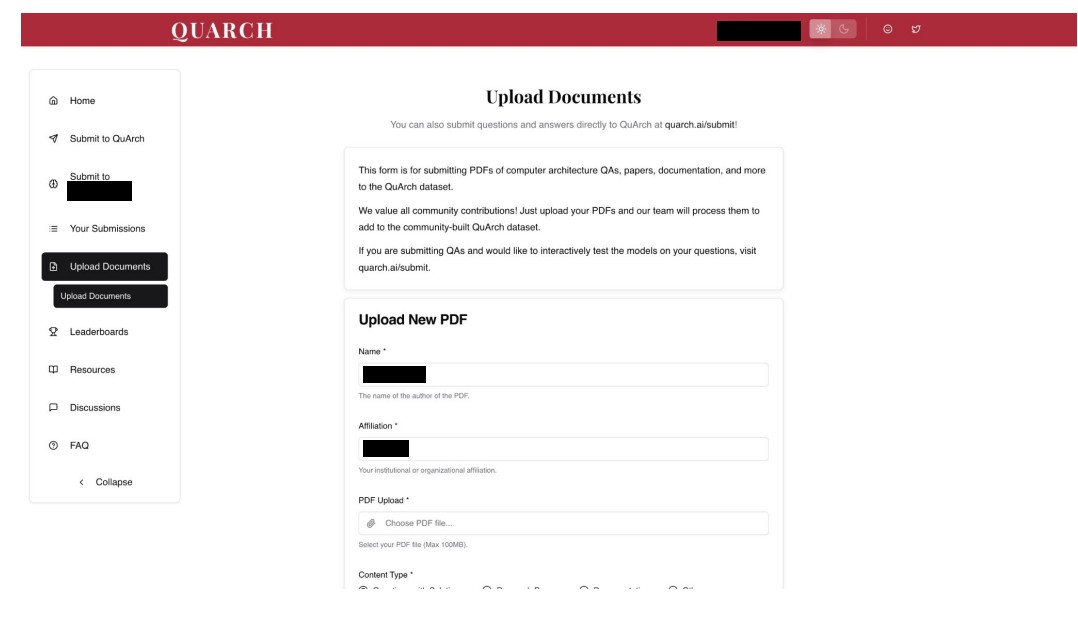

Figure 15: QuArch Website Exam Collection Portal

challenge a broad range of LLMs. The LLM performance is used to assign a score for the question, where making more frontier models fail corresponds to more points, shown in Figure 13. The website tracks submissions for each user, and we compile a leaderboard for users with the most cumulative points and question submissions.

Once a question is submitted, we created an admin approval portal, shown in Figure 14. The approval process involves checking the questions for missing assumptions, poor relevance, or an insufficient rationale. Admins also reproduce each answer manually to ensure that the question-answer pair is objective and correct. Once a submission question is approved, it is added to the QUARCH benchmark set and the user that submitted the question is notified.

Lastly, users can also submit exams in the website, shown in Figure 15. We will then parse out the questions, context images, and answers from these computer architecture course exams using the offline methodology described in Sec. 2.2. Cumulatively, the questions collected from the QuArch crowdsourcing website are taxonomized by skill and funneled into the QUARCH-Reasoning benchmark.

### B.2 TOPIC-WISE FURTHER CHARACTERIZATION

Figure 16 visualizes the topic-wise performance of the frontier models on QUARCH. As performance across all models on QUARCH-Recall is very high, very little topic-wise variability can be seen. However, in the higher-level skills (Analyze, Design, and Implement), models exhibit surprising heterogeneity in per-topic performance.

### B.3 FULL MODEL RESULTS BY SKILL

Table 4 provides the complete set of results for all 26 evaluated models across the four skill categories in QUARCH. The table illustrates how models perform differently on factual recall compared to higher-order reasoning, design, and implementation. This comprehensive view allows for comparison across both small and large language models, highlighting overall trends and providing a foundation for tracking progress over time. While we do not discuss individual results here, the table captures the broader landscape of model performance and makes clear the varying degrees of capability across skills that are critical for computer architecture reasoning.

**Performance of Small Language Models**

Small language models (SLMs) keep pace with LLMs on recall-style questions, but their performance drops sharply on analyze, design, and implement tasks—especially when multimodal reasoning is required. This gap suggests that parameter scale (and associated capacity for long-horizon reasoning and state tracking) matters far more for higher-order architectural problem solving than for fact retrieval. In practice, SLMs are well-suited for low-latency, cost-efficient assistants that handle definitions, quick checks, and targeted lookups, while agentic systems design, trade-off analysis, and figure/table interpretation still benefit from larger models or strong tool scaffolding. A pragmatic path is a cascaded workflow: route recall to SLMs, escalate complex reasoning to LLMs, and bolster SLMs with retrieval and simulators rather than relying on scale alone.

We evaluate the performance of SLMs (defined as $\leq 30B$ parameters) vs LLMs (defined as $\geq 70B$ parameters or proprietary frontier models) on the taxonomy of QUARCH questions. SLMs exhibit a 13% drop in Recall accuracy, but a 27% drop on Analyze, 26% drop on Design, and 25% drop on Implement questions compared to LLMs. This indicates that while SLMs can replicate factual knowledge nearly as well as their larger counterparts, they falter on the higher-order reasoning and synthesis that are core to the field of computer architecture. These results imply that scaling up—or supplementing smaller models with external reasoning tools—is critical for bridging the gap in advanced architectural skills.

### B.4 MODEL PERFORMANCE BY MODALITY

In addition to the main set of frontier models, we conduct a comprehensive evaluation on 26 models spanning both small language models (SLMs) and large language models (LLMs), and breakdown performance by question modality. Results are summarized in Table B.4. This broader analysis highlights consistent trends across scale, including significant gaps in higher-order reasoning and

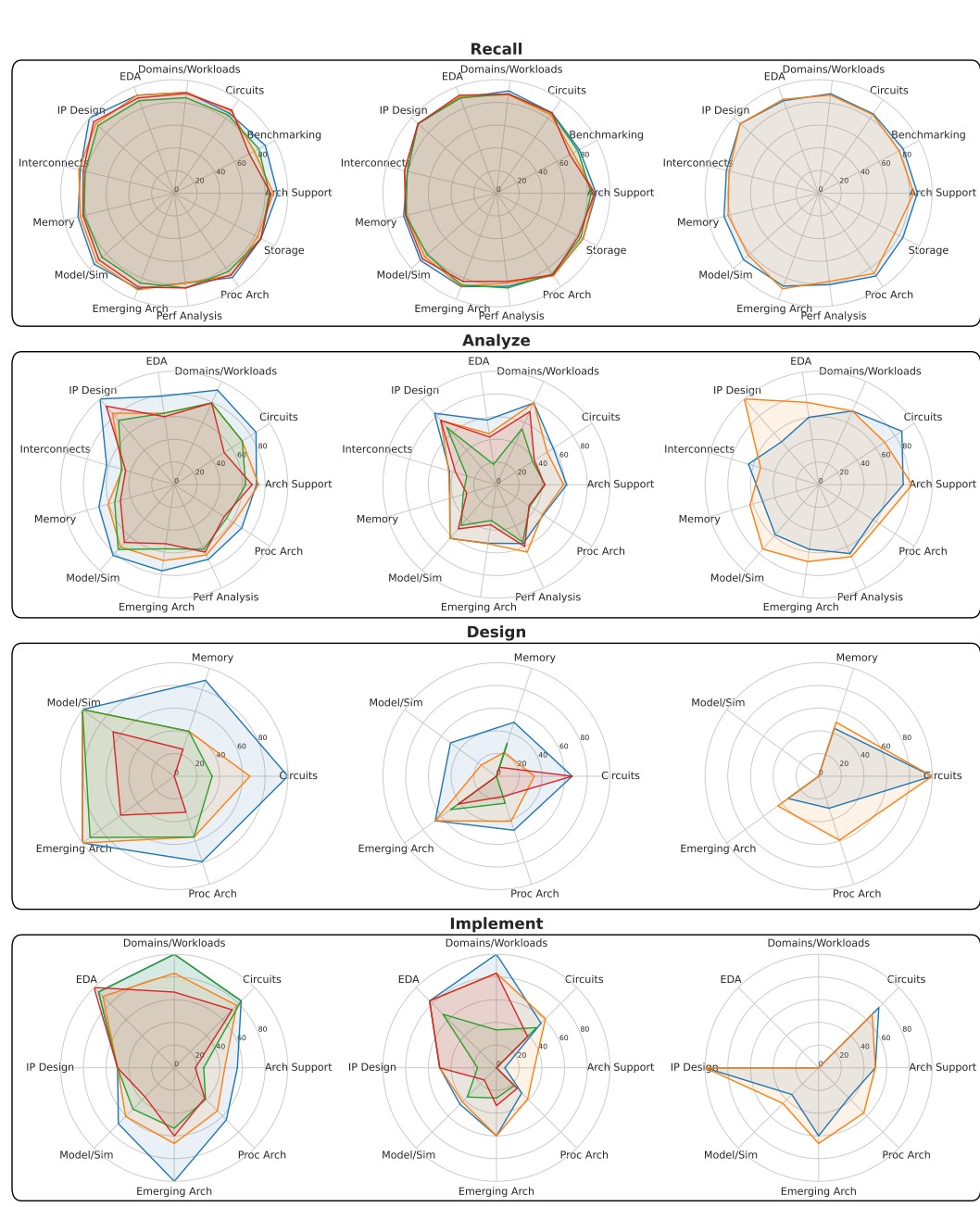

Figure 16: Topic-wise characterization of frontier models, categorized by skill. In order, the groupings of 3 radar plots correspond to "Recall", "Analyze", "Design", and "Implement" questions. Within each grouping, leftmost plot contains the best performing multimodal models, the middle plot contains the worst performing multimodal models, and the rightmost plot contains the two text-only models.

Table 4: Accuracy (%) of all evaluated models across the four skills in QuArch. Best performing models in each category highlighted **first**, **second**, and **third**.

| Model | Recall | QuArch-Reasoning | | | |
| --- | --- | --- | --- | --- | --- |
| | | Analyze | Design | Implement | Overall |
| *Closed-Source Multimodal Models* | | | | | |
| GPT-5 | **89.0** | **72.1** | **86.7** | **71.0** | **72.4** |
| GPT-5 (Non-Reasoning) | **86.3** | 49.5 | 52.4 | 40.7 | 49.0 |
| GPT-4o | 84.1 | 28.4 | 12.4 | 22.9 | 27.5 |
| Gemini 2.5 Pro | **87.4** | **63.3** | **59.0** | **59.7** | **62.9** |
| Gemini 2.5 Flash | 83.4 | **57.3** | **57.1** | **49.8** | **56.8** |
| Claude Sonnet 4 | 85.5 | 49.0 | 36.2 | 46.8 | 48.4 |
| Claude 3.7 Sonnet Thinking | 85.8 | 53.3 | 34.3 | 45.5 | 52.1 |
| Mistral Medium 3.1 | 84.5 | 34.8 | 27.6 | 27.7 | 34.1 |
| *Open-Source Multimodal Models* | | | | | |
| Gemma 3 27B IT | **75.6** | **22.4** | **15.2** | **16.0** | **21.7** |
| Gemma 3 4B IT | 62.3 | 7.6 | 2.9 | 3.5 | 7.2 |
| Llama 4 Maverick | **85.3** | **35.1** | **18.1** | **30.2** | **34.2** |
| Llama 3.2 11B | 69.4 | 8.8 | 1.0 | 4.0 | 8.2 |
| Mistral Small 3.2 24B Instruct | **78.0** | **24.3** | **16.2** | **17.7** | **23.6** |
| *Text-Only Models* | | | | | |
| GPT-OSS 120B | 84.2 | **65.5** | **56.1** | **57.8** | **64.7** |
| DeepSeek R1 | **86.9** | **57.4** | **38.6** | **47.1** | **56.1** |
| Llama 3.3 70B | 80.4 | 25.8 | 0.0 | 13.7 | 24.2 |
| Llama 3.2 1B | 36.5 | 1.7 | 0.0 | 1.0 | 1.6 |
| Mistral Codestral 2508 | 75.3 | 29.7 | 5.3 | 16.7 | 28.1 |
| Mistral Devstral Medium | 81.9 | 29.0 | 3.5 | 22.5 | 27.7 |
| Kimi K2 0905 | 84.2 | 43.9 | 35.1 | 37.3 | 43.2 |
| Qwen3 Coder 480B A35B Instruct | 82.9 | 41.7 | 17.5 | 31.4 | 40.2 |
| Qwen3 235B A22B Thinking | **85.6** | **62.2** | **50.9** | **52.9** | **61.3** |
| Qwen3 235B A22B NonThinking Instruct | **86.5** | 56.4 | 42.1 | **47.1** | 55.3 |
| Qwen3 Next 80B A3B Thinking | 84.5 | 54.9 | 36.8 | 42.2 | 53.5 |
| Qwen3 30B A3B Thinking | 82.5 | 50.0 | 31.6 | 37.3 | 48.6 |
| Qwen3 Coder 30B A3B Instruct | 78.3 | 28.2 | 8.8 | 12.7 | 26.6 |

multimodal tasks compared to LLMs. By including this wider range of models, we provide a more complete picture of the landscape and enable future work to track progress not only at the frontier but also in more lightweight, cost-efficient models.

Table 6 provides detailed comparisons between text-only question performance, image-only performance, and image-text performance. See Sec 4 for interpretation and analysis of sensitivity to input modalities.

## B.5 Partially Correct Judgments

Table 7 reports results when we extend the LLM-as-a-judge rubric to include a "Partially Correct" category. We observe that many models, particularly weaker or smaller ones, produce answers that are not fully correct but demonstrate partial understanding. For example, identifying the right concept while failing to complete all reasoning steps. Incorporating this intermediate category reveals a richer distribution of model behavior: some models that appear very weak under a strict correct/incorrect rubric (e.g., sub-30% accuracy) show substantially higher rates of partially correct answers, suggesting they are closer to reaching full correctness than raw accuracy alone would imply. At the same time, the strongest models still cluster most of their output into "Correct," with only modest use of the partially correct band. This analysis highlights that while partial correctness is less use-

Table 5: Per-generation accuracy (%) by evaluation type and modality. Best performing models in each category highlighted **first**, **second**, and **third**.

| Model | Text-only | | Image-only | Image & Text |
|---|---|---|---|---|
| | MCQ | FRQ | FRQ | FRQ |
| *Closed-Source Multimodal Models* | | | | |
| GPT-5 (High Effort) | **89.0** | **74.7** | **70.5** | **72.7** |
| GPT-5 | **86.4** | 53.8 | 44.8 | 49.4 |
| GPT-4o | 84.3 | 31.3 | 24.6 | 28.1 |
| Gemini 2.5 Pro | **87.5** | **63.9** | **62.1** | **63.0** |
| Gemini 2.5 Flash | 83.2 | **59.3** | **55.4** | **57.4** |
| Claude Sonnet 4 | 85.6 | 52.1 | 45.4 | 48.8 |
| Claude 3.7 Sonnet Thinking | 85.9 | 53.8 | 51.3 | 52.6 |
| Mistral Medium 3.1 | 84.7 | 41.2 | 27.7 | 34.6 |
| *Open-Source Multimodal Models* | | | | |
| Gemma 3 27B IT | **75.7** | **26.8** | **17.7** | **22.4** |
| Gemma 3 4B IT | 63.2 | 9.4 | 4.1 | 6.9 |
| Llama 4 Maverick | **85.9** | **36.1** | **32.1** | **34.2** |
| Llama 3.2 11B | 70.2 | 9.5 | **5.6** | 8.0 |
| Mistral Small 3.2 24B Instruct | **78.6** | **29.1** | **17.7** | **23.5** |
| *Text-Only Models* | | | | |
| GPT-OSS 120B | 84.1 | **65.3** | - | - |
| DeepSeek R1 | **87.1** | **56.2** | - | - |
| Llama 3.3 70B | 81.0 | 23.6 | - | - |
| Llama 3.2 1B | 37.1 | 0.8 | - | - |
| Mistral Codestral 2508 | 75.7 | 28.0 | - | - |
| Mistral Devstral Medium | 82.5 | 27.3 | - | - |
| Kimi K2 0905 | 84.2 | 44.2 | - | - |
| Qwen3 Coder 480B A35B Instruct | 83.3 | 40.1 | - | - |
| Qwen3 235B A22B Thinking | **85.6** | **61.7** | - | - |
| Qwen3 235B A22B NonThinking Instruct | **86.6** | 55.7 | - | - |
| Qwen3 Next 80B A3B Thinking | 84.5 | 54.1 | - | - |
| Qwen3 30B A3B Thinking | 82.7 | 48.8 | - | - |
| Qwen3 Coder 30B A3B Instruct | 78.6 | 26.9 | - | - |

ful in practice for computer architecture tasks that often require precise answers, capturing it can provide a more diagnostic view of model progress and failure modes.

### B.6 COMPARING HUMAN DOMAIN EXPERTS TO LLM-AS-A-JUDGE

We rigorously validate the fidelity of LLM judges used on QUARCH QAs compared to human expert evaluators in Sec. 4.5 and this section. We instruct LLM-as-a-Judge to reason about the accuracy of each freeform response with respect to the ground truth answer as though the response is from a student completing an academic exam (see prompts in Appendix D.3). The judge is instructed to grade each response as CORRECT, PARTIALLY-CORRECT, or INCORRECT. For our reported evaluations (Sec. 4.1), we recategorize each LLM-as-a-Judge assessment into a binary CORRECT or INCORRECT by rounding down PARTIALLY-CORRECT judge assessments to INCORRECT. The PARTIALLY-CORRECT category serves two purposes: it dis-incentivizes the judge from rounding up a nearly-correct answer to correct, and it enables analysis of fine-grained knowledge (Appendix B.5).

We generate multiple samples per question and multiple judgments per sample to control for model stochasticity, and report *pass@k*=1 across 3 samples ($n = 3$) as defined in (Pinckney et al., 2025a). For each question in QUARCH, each model under evaluation (student $s$) generates 3 responses using the model's default generation parameters. For each individual student response, judge model $j$ generates up to 3 assessments until a majority vote consensus is reached. For example, if on a given

Table 6: Per-generation accuracy (%) on the QUARCH benchmark broken down by evaluation type and modality. Best performing models in each category highlighted **first**, **second**, and **third**.

| Model | Text-only | | Image-only | Image & Text |
|---|---|---|---|---|
| | MCQ | FRQ | FRQ | FRQ |
| *Multimodal Models* | | | | |
| GPT-5 (High Effort) | **89.0** | **74.7** | **70.5** | **72.7** |
| GPT-5 | **86.4** | 53.8 | 44.8 | 49.4 |
| Gemini 2.5 Pro | **87.5** | **63.9** | **62.1** | **63.0** |
| Gemini 2.5 Flash | 83.2 | **59.3** | **55.4** | **57.4** |
| Claude Sonnet 4 | 85.6 | 52.1 | 45.4 | 48.8 |
| Claude 3.7 Sonnet Thinking | 85.9 | 53.8 | 51.3 | 52.6 |
| Llama 4 Maverick | 85.9 | 36.1 | 32.1 | 34.2 |
| Mistral Medium 3.1 | 84.7 | 41.2 | 27.7 | 34.6 |
| *Text-Only Models* | | | | |
| GPT-OSS 120B | 84.1 | 65.3 | - | - |
| DeepSeek R1 | 87.1 | 56.2 | - | - |

Table 7: Addition of "Partially Correct" Judgments in LLM-as-a-judge rubric. Results are pass@1 and using a single LLM-as-a-judge response rather than from consensus.

| Model | Correct (%) | Partially Correct (%) | Incorrect (%) |
|---|---|---|---|
| *Closed-Source Multimodal Models* | | | |
| GPT-5 | 70.4 | 19.2 | 10.2 |
| GPT-5 (Non-Reasoning) | 48.5 | 31.6 | 20.0 |
| GPT-4o | 28.5 | 40.2 | 31.3 |
| Gemini 2.5 Pro | 61.9 | 24.1 | 13.8 |
| Gemini 2.5 Flash | 56.7 | 26.7 | 16.5 |
| Claude Sonnet 4 | 49.6 | 31.8 | 18.6 |
| Claude 3.7 Sonnet Thinking | 52.1 | 30.7 | 16.9 |
| Mistral Medium 3.1 | 33.5 | 34.5 | 31.8 |
| *Open-Source Multimodal Models* | | | |
| Gemma 3 27B Instruct | 24.0 | 35.2 | 40.9 |
| Gemma 4 4B Instruct | 6.6 | 30.9 | 62.5 |
| Llama 4 Maverick | 33.7 | 37.9 | 28.5 |
| Llama 3.2 11B | 8.6 | 27.9 | 63.5 |
| Mistral Small 3.2 24B Instruct | 24.5 | 36.4 | 39.0 |
| *Text-Only Models* | | | |
| GPT OSS 120B | 66.7 | 17.5 | 15.8 |
| DeepSeek R1 | 55.9 | 25.2 | 18.8 |
| Llama 3.3 70B | 24.0 | 39.5 | 36.2 |
| Llama 3.2 1B | 0.3 | 13.4 | 86.3 |
| Mistral Codestral 2508 | 27.9 | 37.7 | 33.9 |
| Mistral Devstral Medium | 27.0 | 37.6 | 35.2 |
| Kimi K2 0905 | 45.4 | 25.8 | 28.8 |
| Qwen 3 Coder 480B Instruct | 40.7 | 34.8 | 24.5 |
| Qwen 3 235B A22B Thinking | 57.7 | 16.6 | 25.6 |
| Qwen 3 235B A22B NonThinking Instruct | 54.4 | 26.5 | 18.5 |
| Qwen 3 Next 80B Thinking | 56.1 | 20.6 | 23.2 |
| Qwen 3 30B A3B Thinking | 48.0 | 21.7 | 30.1 |
| Qwen 3 Coder 30B A3B Instruct | 28.9 | 34.7 | 36.2 |

problem, two model samples are each judged by majority vote to be CORRECT, and the third sample is majority vote INCORRECT, *pass@k*=1 on that problem is $\frac{2}{3}$.

We use Claude-3.7-Thinking as our LLM judge for all benchmark evaluations in this paper. We compare Claude against Gemini-2.5-Pro as an alternate candidate for our judge LLM (using a con-

sensus size of 3 for both), and observe Gemini-2.5-Pro achieves a slightly lower agreement rate of 84.73% against human experts, hence our adoption of Claude as our chosen judge. We also inspect the frequency of necessitated tie-breaking under our consensus size of 3 (where a tie consists of one correct and one incorrect verdict) across QUARCH's entire QA dataset and all models assessed in the main text and appendix, and observe across 65,659 responses, the first two judgments from Claude-3.7-Thinking matched 89.0% of the time (and hence did not require a third judgment to adjudicate).

In order to ascertain if LLM judge disagreements with human experts increased in frequency in relation to grading difficulty, we additionally asked each human expert to assign each problem a score between 1-5 on the difficulty of grading the question (not the difficulty of the question). This trend did not emerge in our data; rather, the domain expertise and familiarity with academic content in the human cohort led to 1 and 2 being the most frequently assigned scores for grading difficulty. We believe this preliminary result leads to three potential directions for future work in alignment between the performance of LLM-as-a-Judge and domain experts: (1) judge prompt optimization by both domain experts and automated methods (Opsahl-Ong et al., 2024), such as by informing the judge that students may try to earn extra points on a question they can't answer by including relevant-sounding jargon to mimic understanding (we see this behavior exhibited by some SLMs), (2) characterizing question difficulty in QUARCH and exploring whether harder questions are also harder to accurately grade by both LLMs and humans, and (3) investigating the tradeoff across LLM judge generation parameters between verdict determinism and verdict accuracy under consensus when comparing against human expert verdicts as ground truth.

## C  EXAMPLE QUESTIONS

### C.1  EXAMPLES OF QUESTION SKILLS TAXONOMY

#### C.1.1  RECALL QUESTION

---

**Example 2: Storage Systems**

**Question:** Moving compute closer to the ___ in solid state drives (SSDs) offers higher bandwidth but introduces challenges in managing frequent errors.

**Answers:**
(a) controller
(b) **NAND dies**
(c) cache
(d) DRAM

---

**Rationale:** This question falls under the recall category as it requires retrieval of factual knowledge about SSD architecture, specifically the trade-offs between bandwidth optimization and error management when positioning compute resources relative to different storage components.

#### C.1.2  ANALYSIS QUESTION

---

**Example 4: Systolic Array Analysis**

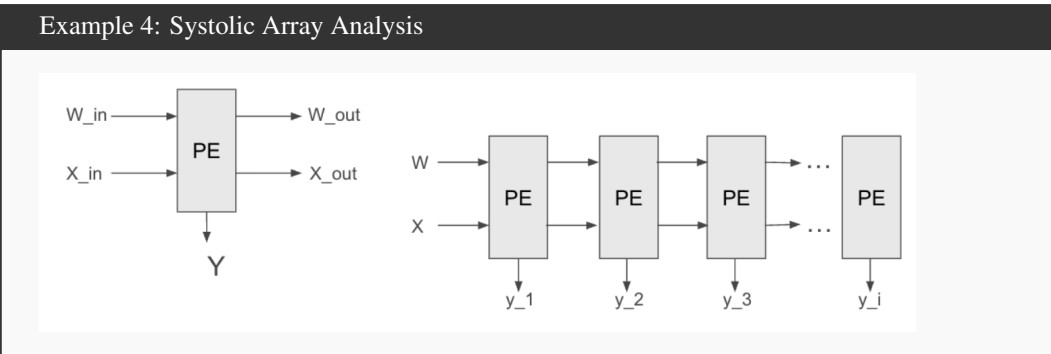

---

**Context:** Given arrays $X = [x\_1, x\_2, x\_3, \ldots, x\_n]$ and $W = [w\_1, w\_2, w\_3, \ldots, w\_k]$, you want to compute
$Y = [y\_1, y\_2, y\_3, \ldots, y\_(n + 1 - k)]$ using the formula:
$y\_i = w\_1 x\_i + w\_2 x\_(i + 1) + w\_3 x\_(i + 2) + \cdots + w\_k x\_(i + k - 1)$

The figure shows a systolic array of processing elements (PEs) and their input-output behavior.

**Question:** *What should be the relative speeds at which $X$ and $W$ values flow through the array to end up with the correct result for all $Y$ values?*

**Correct Solution:** 2:1

This question is formulated using discussion on Fig 7 in the paper "Why Systolic Architectures?" by HT Kung.

Citation: Kung, "Why systolic architectures?," in Computer, vol. 15, no. 1, pp. 37-46, Jan. 1982, doi: 10.1109/MC.1982.1653825.

Please refer to paper for full discussion.

Intuition below:

The formula for y_i is y_i = w1*x_i + w2*x_(i+1) + w3*x_(i+2) + ... + w_k*x_(i+k-1). The number of PEs is equal to the number of Y values. Each PE computes one Y value. A single PE performs the computation Y ← Y + W_in * X_in.

Since w_1 and x_1 are fed as input on the first cycle, the first PE computes y_1. The next PE needs to compute y_2 = w_1*x_2 + w_2*x_3 + ... w_k * x_(k+1). So, after the first cycle, x_2 and w_1 should enter the second PE. This means that x should flow at twice the speed of w to align correctly for the calculations of each y_i. Therefore, the relative speed is 2:1.

**Rationale:** This is an analysis question because it requires breaking down the systolic array's computational flow and examining how data dependencies between X and W arrays must be synchronized across multiple processing elements

### C.1.3 DESIGN QUESTION

**Example 4: Cache Partitioning and Associativity Trade-offs in Multicore Systems**

**Context:** Suppose we have a system with 32 cores that share a physical second-level cache. Assume each core is running a single single-threaded application, and all 32 cores are concurrently running applications. Assume that the page size of the architecture is 8KB, the block size of the cache is 128 bytes, and the cache uses LRU replacement. We would like to ensure each application gets a dedicated space in this shared cache without any interference from other cores. We would like to enforce this using the utility based cache partitioning (UCP) to partition the cache. Assume we would like to design a 4MB cache with a 128-byte block size. Recall that UCP aims to minimize the cache miss rate by allocating more cache ways to applications that obtain the most benefit from more ways, as we discussed in lecture.

**Question:** Consider the maximum associativity of the cache such that each application is guaranteed a minimum amount of space without interference. Is it desirable to implement UCP on a cache with this maximum associativity? Why, why not? Explain.

**Correct Solution:** No, it is not desirable to implement UCP with this maximum associativity because the overhead of UCP for 32 applications on this cache will likely outweigh its

benefits. UCP will only work with LRU replacement policy. But implementing LRU on top of a 32 k-way cache is impractical. Also the number of counters needed by UCP and the partitioning solution space for UCP are very large for such a cache.

**Rationale:** This question qualifies as a design question because it requires the analysis and formulation of architectural strategies for cache partitioning in multicore systems. Rather than executing a specific algorithm or implementation, the focus is on evaluating system-level trade-offs, exploring alternative approaches, and proposing optimal solutions under varying associativity constraints.

### C.1.4 IMPLEMENTATION QUESTION

#### Example 4: Linked-List Manipulation via Self-Modifying Code on EDSACjr

**Context:** In this question, you will implement linked-list operations using self-modifying code on an EDSACjr machine. The memory layout is shown in the figure on the right. You have access to the named memory locations as indicated. Linked-list nodes consist of two words: the first is an integer value, the second is an address pointing to the next node. _HEAD contains the address of the first node of the list (or $INVALID$ if it is empty). The next field of the last node is _INVALID. All valid addresses are positive. You may create new local and global labels as explained in the EDSACjr handout. Table A-1 shows the EDSACjr instruction set.

Table A-1 shows the EDSACjr instruction set.

| Opcode | Description | Bit Representation |
|---|---|---|
| ADD $n$ | Accum ← Accum + M[$n$] | 00001 $n$ |
| SUB $n$ | Accum ← Accum - M[$n$] | 10000 $n$ |
| STORE $n$ | M[$n$] ← Accum | 00010 $n$ |
| CLEAR | Accum ← 0 | 00011 00000000000 |
| OR $n$ | Accum ← Accum \| M[$n$] | 00000 $n$ |
| AND $n$ | Accum ← Accum & M[$n$] | 00100 $n$ |
| SHIFTR $n$ | Accum ← Accum shiftr $n$ | 00101 $n$ |
| SHIFTL $n$ | Accum ← Accum shiftl $n$ | 00110 $n$ |
| BGE $n$ | If Accum ≥ 0 then PC ← $n$ | 00111 $n$ |
| BLT $n$ | If Accum < 0 then PC ← $n$ | 01000 $n$ |
| END | Halt machine | 01010 00000000000 |

Context Figures

| Macro | Description |
|---|---|
| STOREADR $n$ | Replace the address field of location $n$ with the contents of the accumulator |
| LOADADR $n$ | Load the address field of location $n$ into the accumulator |

Figure 18: Caption for image 2

Figure 17: Caption for image 1

You may also use the following macros if required:

- **STOREADR n**: Replace the address field of location n with the contents of the accumulator
- **LOADADR n**: Load the address field of location n into the accumulator

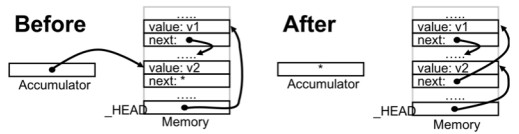

Write a macro for **LISTPUSH**, which pushes the node pointed to by the accumulator to the head of the list. LISTPUSH takes one argument, the memory address of the new node, which is available in the accumulator. As shown in the figure below, LISTPUSH stores the current _HEAD pointer in the new node's next field, and updates the _HEAD pointer to point to the new node. Implement the macro using the EDSACjr instruction set and macros provided above. Do not refer to "value" or "next"; they are for illustration only. You need not worry about memory allocation; the new node's address is provided in the accumulator.

**Question:**

```
.macro LISTPUSH
    STORE _TMP   ;; store accumulator (address of the new node)
.end
```

**Correct Solution:** 2:3

```
.macro LISTPUSH
    STORE _TMP  ;; store accumulator (address of the new node)
    ADD _ONE  ;; accum <- address of the new node's next field
    STOREADR _STN ;; address field of location _STN
                      has the address
                ;; of the new node's next field
    CLEAR
    ADD _HEAD  ;; accum <- M[_HEAD], current head pointer
_STN: STORE 0  ;; 0 will be replaced with the node's next field
            ;; address. M[_TMP + 1] <- accum
    CLEAR
    ADD _TMP   ;; retrieve address of new node in accumulator
    STORE _HEAD ;; M[_HEAD] <- accum; Update the head pointer
                ;; to the new node
.end
```

**Rationale:** This is an implementation question because it requires constructing actual, working code or macros using the EDSACjr instruction set to perform specific memory operations. Rather than theorizing or analyzing abstract concepts, it requires to translate the algorithm into concrete steps that the hardware can execute.

## C.2 FAILURE MODE 1: STRUGGLES WITH ARCHITECTURE-SEMANTICS OF CODE EXECUTION

### C.2.1 EXAMPLE 1

**Example: Reuse Intervals in Matrix Multiplication**

**Context:**
For a given compute kernel, we define a tensor's reuse interval (RI) as the number of different elements of that tensor that have been referenced between each re-reference of the same element. For example, consider the following:

```
for m in [0, M)
  for n in [0, N)
    Z[m, n] = A[m] * B[n]
```

Since A's element is used at every iteration of the inner loop, its RI is 1. Each element of B is re-referenced after N references, so its RI is N. Z has "infinite" reuse interval (i.e., no data reuse) since no element is re-referenced throughout the computation:
RI of A = 1
RI of B = N
RI of Z = infinite / no reuse
Consider the following Matrix-Matrix multiply pseudocode, which multiplies two dense matrices A and B to produce Z:

- Multiply two matrices A and B to produce Z

- First matrix A is MxK

- Second matrix B is KxN

- Thus, resulting matrix Z is MxN

```
for m in [0, M)
  for n in [0, N)
    for k in [0, K)
      Z[m, n] += A[m, k] * B[k, n]
```

**Question:**
What are the reuse intervals for the three matrices? Provide your answers in terms of M, N, and K.

**Correct Solution:**
RI of A = K
RI of B = N*K
RI of Z = 1

**Incorrect Frontier Model Response:**
To determine the reuse intervals (RIs) for the matrices A, B, and Z in the given Matrix– Matrix multiply pseudocode, let's analyze the loop structure and how each matrix is accessed.

The pseudocode is as follows:

```
for m in [0, M)
  for n in [0, N)
    for k in [0, K)
      Z[m, n] += A[m, k] * B[k, n]
```

*1. Matrix A (MxK).*

- The element `A[m, k]` is accessed within the innermost loop (`for k in [0, K)`).

- For a fixed m, the row `A[m, k]` is accessed for each k from 0 to K-1. Thus, for each iteration of the outer loop (`for m in [0, M)`), the same row is not re-referenced until M iterations have passed.

- Within the inner loop, each element of a row of A is accessed once. So for a given m, elements are accessed once as k varies from 0 to K-1. For the next m, the same element is re-referenced after K iterations of the inner loop.

- More precisely: the re-reference happens after the inner loop (`for k`) has completed for all N iterations of n, because m is fixed during those N iterations. Hence the reuse interval for A is N*K.

- A finer analysis shows that each element `A[m, k]` is accessed for every n in [0, N). Thus, it is reused N times before the next k is accessed.

Hence, the correct RI for A is N.
*2. Matrix B (KxN).*

- `B[k, n]` is accessed within the innermost loop (`for k in [0, K)`).

- For a fixed n, B[k, n] is accessed for each k. Each element of a column is accessed once as k varies, and re-referenced after K iterations for a different k.

- Considering the loop structure: for a given k and n, B[k, n] is accessed for every m in [0, M). Thus, it is reused M times.

Hence, the RI for B is M.

*3. Matrix Z (MxN).*

- Z[m, n] is accessed within the innermost loop (for k in [0, K)).

- For a fixed m, n, Z[m, n] is updated K times. It is not re-referenced until the computation for that pair is completed, which happens after K iterations of the innermost loop.

Hence, the RI for Z is K.

*Therefore, the reuse intervals are:*

- RI of A = N

- RI of B = M

- RI of Z = K

These values represent how many times each element of the respective matrices is reused before being re-referenced.

**Question Summary and Relevance:** This question examines understanding of reuse distance analysis in nested loop kernels, a fundamental concept in computer architecture for predicting cache behavior and memory bandwidth requirements. The reuse interval (RI) counts the number of distinct memory references between successive accesses to the same data element, directly determining cache hit rates through the stack distance model.

For matrix multiplication—the dominant kernel in scientific computing and deep learning—accurate reuse analysis guides critical optimization decisions: *cache blocking factors*, *prefetch distances*, and *loop transformation strategies*. The question specifically tests whether models can trace memory access sequences through triple-nested loops and correctly apply the reuse interval definition to determine that A has $RI = K$ (reused after each inner loop completion), B has $RI = N \times K$ (reused only after all $N \times K$ iterations), and Z has $RI = 1$ (immediately reused for accumulation).

**Analysis of Model Response:** The model's errors reveal a fundamental conceptual confusion between **reuse count** (how many times an element is accessed) and **reuse interval** (distinct accesses between reuses). For example, when analyzing matrix $A$, the model incorrectly reasons that because "$A[m, k]$ *is accessed for every n in* $[0, N)$... *it is reused N times,*" concluding $RI = N$. This conflates usage frequency with reuse distance—the correct analysis requires tracing the actual reference stream to see that $A[m, k]$ is accessed once, then $K$ different elements are accessed before returning to $A[m, k]$ in the next $n$ iteration, giving $RI = K$. Similarly, for matrix $B$, the model focuses on $B$ being "reused $M$ times" across different $m$ values, missing that the same $B[k, n]$ element isn't accessed again until after $N \times K$ intervening references.

### C.2.2 EXAMPLE 2

---

**Example: Timing Side Channel from Short-Circuit Evaluation**

**Question:**
Consider the code given below:

```
bool detect_duplicate(int* elements, int length){
    bool cond, dup = false;
    for(int i=0; i<length; i++) {
        for(int j=0; j<length; j++) {
            cond = (elements[i] == elements[j]) && (i!=j);
            dup  = CMOV(cond, cond, dup);
        }}
```

```
1782
1783        return !dup;
1784   }
```

Assume that CMOV is implemented as the `cmov` x86 instruction, a secure constant-time instruction. Assume the length of the array is already known publicly and is not sensitive information. Does this code have timing side channels that can reveal insights into the contents of the `elements` array?

**Correct Solution:**
Yes, there is a timing side channel.

**Incorrect Frontier Model Response:**

Analysis of Timing Side Channels in the Code
The provided code aims to detect duplicates in an array without leaking information through timing side channels. Below, I analyze whether the code has timing side channels that could reveal insights into the contents of the `elements` array.
*Key Features of the Code.*

- The function `detect_duplicate` uses two nested loops that iterate exactly `length × length` times, regardless of the array values. This fixed iteration count ensures that the loop structure itself does not introduce timing variations based on the data.

- The critical operation inside the inner loop is a comparison (`elements[i] == elements[j]`) and a logical AND with (`i != j`), followed by a conditional move (CMOV) instruction. The CMOV instruction is specified as a secure constant-time instruction (like the x86 `cmov`), meaning it executes in constant time regardless of the condition outcome.

- The memory access pattern is determined solely by the indices `i` and `j`, which are fixed for a given `length`. Since `length` is public and non-sensitive, the sequence of memory accesses (to `elements[i]` and `elements[j]`) is identical for every invocation with the same `length`. Thus, cache behavior (e.g., hits or misses) is predictable and does not depend on the array values.

- There are no conditional branches that depend on the array values. All instructions (comparisons, logical operations, and CMOV) are executed unconditionally, minimizing the risk of variable execution time due to branch prediction or speculation.

*Reasoning.*

- Constant-Time Operations: The comparison `elements[i] == elements[j]` is typically implemented as a constant-time integer comparison on modern processors. Similarly, the logical AND and CMOV operations are constant-time under the given assumptions. Therefore, the time taken per inner iteration is invariant to the array values.

- Fixed Workload: The total number of iterations is $length^2$, which is constant for a given `length`. Since the work per iteration is constant-time, the overall execution time is determined solely by `length`, not by the array contents.

- Cache Considerations: While cache timing could theoretically vary based on access patterns, here the access pattern is fixed for a given `length`. The values in the array do not influence the addresses accessed, so cache behavior cannot reveal information about the values themselves.

- Return Value: The function returns `!dup`, but this computation occurs after the loops and is trivial (a single negation). It does not affect the timing of the main computation.

*Conclusion.* Under the assumptions that:

- CMOV is a secure constant-time instruction,

- integer comparisons are constant-time,

- and `length` is public and non-sensitive,

the code does not have timing side channels that can reveal insights into the contents of the `elements` array. The execution time is predictable and depends only on `length`, not on the presence or absence of duplicates or the specific values in the array.

*Additional Note.* If the processor exhibits timing variations for CMOV or comparisons due to microarchitectural factors (e.g., value-dependent aliasing), these are considered negligible or non-exploitable for side channels in this context, given the problem constraints. However, in absolute terms, constant-time code should be verified on specific hardware, but the code here is designed to be constant-time.

**Question Summary and Relevance**: This question tests understanding of ***timing side-channel vulnerabilities*** in security-critical code, a fundamental concern in computer security and cryptographic implementations. The problem presents a duplicate detection function that attempts to be constant-time using the CMOV (conditional move) instruction—a hardware primitive specifically designed to avoid branch-based timing attacks.

- **Secure computation:** Modern processors need to support constant-time operations for cryptographic code to prevent secret extraction through timing analysis.
- **Hardware–software security boundary:** Tests whether one understands that hardware guarantees (CMOV being constant-time) can be undermined by language-level ss.
- **Compiler behavior:** The short-circuit evaluation of `&&` is a compiler optimization that can inadvertently break security guarantees.
- **Real-world impact:** Such vulnerabilities have been exploited in attacks against RSA, AES, and other cryptographic implementations.

The question is particularly clever because it combines a secure primitive (CMOV) with an insecure pattern (short-circuit evaluation), testing whether one can reason about the interaction between hardware features and high-level language semantics.

**Analysis of Model Response:** The model concluded that the code was secure, citing the constant-time nature of CMOV and the public array length, but it failed to account for the short-circuit evaluation of the logical AND operator in C. This oversight demonstrates that the model failed to effectively reason about the architecture-semantics interplay in code execution. Specifically, the model did not incorporate how code semantics (like conditional evaluation) introduce data-dependent timing variations, leading to an incorrect assessment of side channels.

### C.2.3 EXAMPLE 3

**Example: SIMD Utilization and Warp Divergence**

**Context:**

We define the SIMD utilization of a program that runs on a GPU as the fraction of SIMD lanes that are kept busy with active threads during the run of a program. As we saw in lecture and practice exercises, the SIMD utilization of a program is computed across the complete run of the program. The following code segment is run on a GPU. A warp in the GPU consists of 64 threads, and there are 64 SIMD lanes in the GPU. Each thread executes a single iteration of the shown loop. Assume that the data values of the arrays A and B are already in vector registers so there
are no loads and stores in this program. Both A and B are arrays of integers. (Hint: notice that there are 6 instructions in each thread.)

```
for (i = 0; i < 4096; i++) {
    if (B[i] < 8888) { // Instruction 1
        A[i] = A[i] * C[i]; // Instruction 2
        A[i] = A[i] + B[i] // Instruction 3
        C[i] = B[i] + 1; // Instruction 4
    }
```

```
        if (B[i] > 8888) { // Instruction 5
            A[i] = A[i] * B[i]; // Instruction 6
        }
    }
```

**Question:**
What needs to be true about array B to achieve the minimum possible SIMD utilization?
Show your work. (Please cover all cases in your answer.)

**Correct Solution:**
1 in every 64 of B's elements are greater than 8888, and 1 in every 64 of B's elements are less than 8888, and the rest of the elements are 8888.

**Incorrect Frontier Model Response:**
Looking at this problem, I need to analyze how SIMD execution works with conditional branches and determine what values in array $B$ would minimize the number of active threads across all instructions.

*Understanding the Code Structure*
The code has 6 instructions total:

1. if $(B[i] < 8888)$ (condition check)
2. $A[i] = A[i] * C[i]$ (executed if $B[i] < 8888$)
3. $A[i] = A[i] + B[i]$ (executed if $B[i] < 8888$)
4. $C[i] = B[i] + 1$ (executed if $B[i] < 8888$)
5. if $(B[i] > 8888)$ (condition check)
6. $A[i] = A[i] * B[i]$ (executed if $B[i] > 8888$)

*SIMD Execution with Branches*
With 4096 iterations and 64 threads per warp, we have 64 warps total.
In SIMD execution:

- All threads in a warp execute the same instruction simultaneously
- When branches occur, threads not taking a path become inactive
- The warp must execute all paths that any thread takes

*Analysis by Cases*
For each thread, based on $B[i]$ value:
*Case 1: $B[i] < 8888$*

- Thread executes: Instructions 1, 2, 3, 4, 5 (instruction 6 skipped)
- Active for 5 out of 6 instructions

*Case 2: $B[i] = 8888$*

- Thread executes: Instructions 1, 5 only (both conditional blocks skipped)
- Active for 2 out of 6 instructions

*Case 3: $B[i] > 8888$*

- Thread executes: Instructions 1, 5, 6 (instructions 2, 3, 4 skipped)
- Active for 3 out of 6 instructions

*Calculating Active Thread-Instructions per Warp*
For a warp with:

- $n_1$ threads where $B[i] < 8888$
- $n_2$ threads where $B[i] = 8888$
- $n_3$ threads where $B[i] > 8888$
- $n_1 + n_2 + n_3 = 64$

Active thread-instructions:

- Instruction 1: 64 threads (all execute condition)
- Instructions 2,3,4: $n_1$ threads each = $3n_1$ total
- Instruction 5: 64 threads (all execute condition)
- Instruction 6: $n_3$ threads

Total per warp = $64 + 3n_1 + 64 + n_3 = 128 + 3n_1 + n_3$

*Minimizing SIMD Utilization*

To minimize active thread-instructions, minimize: $3n_1 + n_3$

Since $n_1 + n_2 + n_3 = 64$:

- Set $n_1 = 0$ (no threads with $B[i] < 8888$)
- Set $n_3 = 0$ (no threads with $B[i] > 8888$)
- Set $n_2 = 64$ (all threads with $B[i] = 8888$)

*Answer*

For minimum SIMD utilization: All elements of array $B$ must equal 8888

*Verification*

- Total possible thread-instruction slots = $64 \times 6 \times 64 = 24{,}576$
- With all $B[i] = 8888$: Active thread-instructions = $64 \times 128 = 8{,}192$
- Minimum SIMD utilization = $\dfrac{8{,}192}{24{,}576} = \dfrac{1}{3} \approx 0.333$

This occurs because when $B[i] = 8888$, threads only execute the two condition checks (instructions 1 and 5) and skip both conditional blocks entirely, minimizing the number of active thread-instructions across the program execution.

**Question Summary and Relevance:** This problem illustrates SIMD utilization and warp divergence in GPUs, a central topic in computer architecture. SIMD (Single Instruction, Multiple Data) execution keeps all lanes busy when threads follow the same path, but utilization drops when conditional branches cause divergence. The question asks what values in array B minimize utilization across all warps, emphasizing the cost of divergent execution paths. The relevance is twofold:

- it tests understanding of control flow divergence and its impact on performance

- it highlights how data distribution (values in B) directly affects instruction activity.

Such reasoning connects hardware execution models to software efficiency, which is a core skill in computer systems design.

**Analysis of Model Response:** The model response ("Student D") misinterpreted the optimization goal. It assumed that minimizing the *number of active instructions per thread* would directly minimize SIMD utilization, and therefore concluded that setting all $B[i] = 8888$ yields the minimum utilization. However, this reasoning overlooks warp-level execution semantics: if all threads skip the same conditional blocks, the warp does not diverge, and utilization is actually higher (per executed instruction) than in the case where threads diverge.

The core mistake was conflating *per-thread activity minimization* with *warp-level utilization minimization*. The correct answer requires mixing values of $B$ such that different threads in a warp diverge (some with $B[i] < 8888$, some with $B[i] > 8888$, and the rest with $B[i] = 8888$). This forces the warp to execute all branches, but with minimal active participation in each, thereby reducing the overall SIMD utilization.

### C.2.4 EXAMPLE 4

---

**Example: Combinational vs. Sequential from Verilog**

**Question:**

```
1  module one (input clk, input a, input b, output reg [1:0] q);
2    always @ (*)
3      if (b)
4        q <= 2'b01;
5      else if (a)
6        q <= 2'b10;
7  endmodule
```

Does the following code result in a sequential circuit or a combinational circuit? Please explain why.

**Correct Solution:**
This code results in a **sequential** circuit because a latch is required to store the old value of q if both conditions are not satisfied.

**Incorrect Frontier Model Response:**
*Answer and Concise Explanation:*
This code results in a combinational circuit, but with an important caveat - it creates an unintended latch.

*Reasoning:*

1. *Sensitivity List Analysis*: The always @ (*) statement creates a combinational sensitivity list that triggers whenever any input used within the always block changes. This is the key indicator of combinational logic.

2. *No Clock Edge*: There is no clock edge (like posedge clk or negedge clk) in the sensitivity list. Although clk is declared as an input, it is never used in the always block, so it doesn't affect the circuit behavior.

3. *Data Type Clarification*: The reg data type for output q does not imply sequential behavior - it's simply a data type that can hold values in procedural blocks.

*Critical Issue - Incomplete Case Coverage:*

The code has a significant problem:
- When b = 1: q = 2'b01
- When b = 0 and a = 1: q = 2'b10
- When b = 0 and a = 0: No assignment to q

This incomplete case coverage creates an unintended latch because when both inputs are 0, the output q retains its previous value, introducing memory behavior.

*Conclusion:*

While this synthesizes as combinational logic (due to the always @ (*) construct), it creates a problematic latch that should be avoided in good design practice. To make it purely combinational, an else clause should be added to handle the case when both a and b are 0.

---

**Question Summary and Relevance:** The given Verilog code asks whether the design results in a sequential or combinational circuit. At first glance, the use of always @(*) suggests combinational intent since the block triggers on input changes rather than a clock edge. However, because not all input conditions assign a value to the output q, the synthesis tool infers a latch to hold the

previous value when both `a` and `b` are `0`. This subtlety makes the design sequential, even without an explicit clock.

In the context of computer architecture, understanding such distinctions is critical. Hardware description languages (HDLs) like Verilog are used to describe datapaths, control units, and memory systems. Misclassifying sequential vs. combinational logic can lead to unintended latches, incorrect timing behavior, or hardware inefficiencies. This impacts pipeline stages, critical path delays, and ultimately system performance and reliability—fundamental aspects of architecture design.

**Analysis of Model Response:** The model's response (Student D) incorrectly concluded that the code produces a purely combinational circuit with an unintended latch. While it correctly identified the incomplete case coverage, it failed to reconcile the key fact that *incomplete assignments within an* `always @(*)` *block result in latch inference*. This is not "still combinational with a caveat," but a defining feature of sequential logic, since the latch introduces memory behavior.

## C.3 FAILURE MODE 2: ASSUMING UNCONVENTIONAL ARCHITECTURAL PROPERTIES

### C.3.1 EXAMPLE 1

---

**Example: Big-Endian Ordering**

**Context:**
Suppose a virtual memory system employs a 64KB page size and a two-level hierarchical page table. The physical address of the base of the Level-1 page table (0x01000) is stored in a dedicated register called the Page Table Base Register. Both the virtual and physical addresses are 20 bits wide. The figure below illustrates the page table structure and the breakdown of a virtual address in this system.

Each page table entry (PTE) in both Level-1 and Level-2 tables occupies 4 bytes, and the memory is byte-addressable. All pages and page tables are assumed to reside in main memory. Each entry in the Level-1 page table contains the physical base address of a Level-2 page table, while each entry in a Level-2 page table holds the PTE corresponding to a data page. (The diagram is not drawn to scale.) As shown, the L1 index and L2 index are used to select the appropriate 4-byte entry in the Level-1 and Level-2 page tables, respectively.

A Level-2 page table entry can be decomposed into the following fields (status bits can be ignored for this discussion).

---

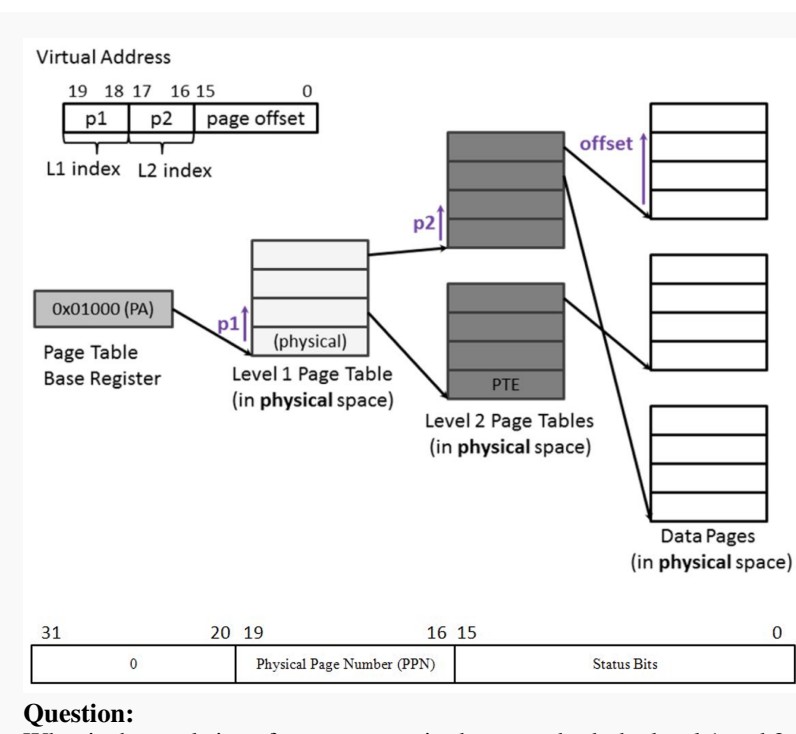

**Question:**
What is the total size of memory required to store both the level 1 and 2 page tables?

**Correct Solution:**
L1: 4 lines each 4B → 16B
L2: 4 × 4 × 4 B = 64B → Total = 16 + 64 = 80B

**Incorrect Frontier Model Response:**
To determine the total memory required for both levels of page tables, we analyze the virtual address structure and calculate the number of entries in each level.
*Step 1: Analyze the Virtual Address Breakdown*
From the provided diagram, the 20-bit virtual address is divided as follows:

- Bits 19–18: L1 index ($p_1$) – 2 bits

- Bits 17–16: L2 index ($p_2$) – 2 bits

- Bits 15–0: Page offset – 16 bits (corresponding to the 64KB page size: $2^{16}$ bytes)

*Step 2: Calculate Level 1 Page Table Size*

- Number of entries in Level 1 = $2^{\text{number of L1 index bits}} = 2^2 = 4$

- Each entry size = 4 bytes

- Total Level 1 size = $4 \times 4 = 16$ bytes

*Step 3: Calculate Level 2 Page Tables Size*

- Number of Level 2 page tables = Number of entries in Level 1 = 4

- Number of entries in each Level 2 page table = $2^{\text{number of L2 index bits}} = 2^2 = 4$

- Each entry size = 4 bytes

- Size of one Level 2 page table = $4 \times 4 = 16$ bytes

- Total Level 2 size = $4 \times 16 = 64$ bytes

*Step 4: Calculate Total Size*

$$\text{Total memory required} = \text{Level 1 size} + \text{Total Level 2 size}$$

$$= 16\,\text{bytes} + 64\,\text{bytes} = 80\,\text{bytes}$$

Therefore, the total size of memory required to store both the Level 1 and Level 2 page tables is 80 bytes.

**Question Summary and Relevance:** This question asks for the memory size required to store a two-level hierarchical page table in a 20-bit virtual memory system with 64KB pages. Since the system is byte-addressable, the calculation should follow conventional little-endian, byte-level memory assumptions common to modern processors. Correct reasoning requires straightforward indexing into the page tables and summing their storage requirements.

**Analysis of Model Response:** The model incorrectly assumes big-endian ordering in the interpretation of address breakdown. Instead of aligning the lower-order bits with the page offset, the response reverses the significance of address portions, misplacing table indices. While big-endian addressing is technically valid, it contradicts the universally assumed little-endian default in modern architecture, leading to a miscalculated total size. This illustrates how models, absent explicit prompt guidance, may diverge into less conventional assumptions not used in real-world practice.

### C.3.2 EXAMPLE 2

---

### Example: Word-Level vs. Byte-Level Addressing

**Context:**
In this question, you will be given a task and two code snippets in MIPS assembly language. You will have to decide which of the code snippets can be used for the task.

For all the questions assume the following initial values:
Registers:

| Register | Value |
|----------|-------|
| $s0 | 0x000000FF |
| $s1 | 0x00000004 |
| $s2 | 0x00000008 |
| $s3 | 0x0000000C |

Memory:

| Address | Value |
|---------|-------|
| 0x00000000 | 0x0000FF00 |
| 0x00000004 | 0x000000FF |
| 0x00000008 | 0xFFFFFFF7 |
| 0x0000000C | 0x12345678 |

**Question:**
Write all zeroes to memory starting at 0x0000 4000 for 1024 consecutive memory locations (up to 0x0000 5000)

(A)

```
addi $s0, $s0, 0x1000
LOOP:
    sw   $0, 0x4000($s0)
    addi $s0, $s0, -1
    bne  $s0, $0, LOOP
```

(B)

```
addi $s0, $s0, 0x4000
addi $s1, $s0, 0x1000
addi $s2, $0, 1
LOOP:
    sw   $0, 0($s0)
    sub  $s1, $s1, $s2
    bne  $s0, $s1, LOOP
```

**Correct Solution:**
A Since the assignment is on register $s0, which is constant at 0x4000, if the line were to read:

---

```
LOOP:  sw  $0,  0($s1)
```

it would be correct.

**Incorrect Frontier Model Response:**
Looking at this problem, I need to analyze which code snippet correctly writes zeroes to 1024 consecutive memory locations starting at `0x00004000`.

Let me trace through each code snippet:

*Analysis of Code Snippet (A):*
Initial state: `$s0 = 0x000000FF`

```
addi  $s0,  $s0,  0x1000      #  $s0 = 0x000000FF+0x1000 = 0x000010FF
LOOP:
    sw    $0,  0x4000($s0)   #  Write  0  to  address  (0x4000  +  $s0)
    addi  $s0,  $s0,  −1      #  Decrement  $s0  by  1
    bne   $s0,  $0,  LOOP     #  Continue  until  $s0 = 0
```

This will write to addresses:

- First iteration: `0x4000 + 0x10FF = 0x50FF`

- Second iteration: `0x4000 + 0x10FE = 0x50FE`

- . . .

- Last iteration: `0x4000 + 0x0000 = 0x4000`

Number of iterations: `0x10FF + 1 = 4352`
Memory range: `0x4000` to `0x50FF` (consecutive bytes, going backwards).

*Analysis of Code Snippet (B):*
Initial state: `$s0 = 0x000000FF, $s1 = 0x00000004`

```
addi  $s0,  $s0,  0x4000      #  $s0 = 0x000000FF+0x4000 = 0x000040FF
addi  $s1,  $s0,  0x1000      #  $s1 = 0x000040FF+0x1000 = 0x000050FF
addi  $s2,  $0,  1            #  $s2 = 1
LOOP:
    sw    $0,  0($s0)          #  Write  0  to  address  $s0
    sub   $s1,  $s1,  $s2      #  $s1 = $s1 − 1
    bne   $s0,  $s1,  LOOP     #  Continue  until  $s0 = $s1
```

Critical issue: `$s0` never changes in the loop! It remains `0x000040FF` throughout.

This will:

- Always write to the same address: `0x000040FF`

- Loop `0x000050FF − 0x000040FF = 0x1000 = 4096` times

- Write the same memory location 4096 times

*Target requirement*: Write zeroes to 1024 consecutive memory locations from `0x00004000` to `0x00004FFC` (1024 words × 4 bytes = 4096 bytes total).
*Code Snippet (A):*

- Writes to wrong address range (`0x4000` to `0x50FF` instead of `0x4000` to `0x4FFC`)

- Writes to individual bytes instead of word boundaries

- Goes backward instead of forward

- At least covers the target memory range (and more)

*Code Snippet (B):*

- Writes to only one memory location (`0x000040FF`)

- This location is outside the target range
- Completely fails to meet the requirement

Neither code snippet correctly implements the required task. However, if I must choose between them:

- Code Snippet (A) is closer to being correct because:
    1. It writes to a memory range that includes the target addresses
    2. It writes zeros to memory (even if to too many locations)
    3. Code Snippet (B) completely fails by writing to only one wrong address

*Answer: (A)* — Though flawed, it's the only option that writes to multiple memory locations in the vicinity of the target range.

**Question Summary and Relevance:** The task required writing zeroes to memory locations from `0x00004000` through `0x00005000` – a range of exactly 1024 word-aligned addresses. This detail is crucial because MIPS sw instructions demand word alignment (multiples of 4), and the iteration step must move in increments of 4 to ensure correct coverage of 1024 words. Any solution that treats the loop index as a byte counter or ignores alignment will fail to meet the specification, either by writing to invalid addresses, writing too many times, or missing required locations.

**Analysis of Model Response:** The model concludes that snippet (A) is "closer to being correct," overlooking its fundamental flaw: it decrements $s0 by one each iteration, producing byte-level addressing instead of word-level addressing. This causes misaligned sw operations at addresses such as 0x50FF, 0x50FE, etc., which would raise exceptions on a real MIPS processor. Although Student D recognizes that snippet (B) only writes to a single location, their preference for snippet (A) ignores the fact that (A) still does not meet the requirement – it overshoots the memory range, writes more than 1024 times, and uses byte steps instead of 4-byte steps. The key error in Student D's analysis is failing to recognize that word-level vs. byte-level addressing is a decisive factor that invalidates snippet (A).

### C.3.3  EXAMPLE 3

**Example: Non-Standard Bit Indexing Patterns**

**Context:**
Let A and B be two unknown 8-bit 2's complement numbers. We know the results of $A \oplus B$ and $A \wedge B$ as shown below:
$$A \oplus B = 00110100$$
$$A \wedge B = 11001001$$

**Question:**
What is the sum $A + B$ expressed in the 8-bit two's complement notation?
On an x86 system, would the carry flag be set after $A + B$? What about the overflow flag?

**Correct Solution:**
$$A + B = 11000110$$

Will carry flag be set? **Yes**
Will overflow flag be set? **No**

**Incorrect Frontier Model Response:**
The question asks for the sum $A + B$ in 8-bit two's complement notation and the state of the carry and overflow flags on an x86 system after this addition.

We are given:

- $A \oplus B = 00110100$ (Let this be $P$, where $P_i = A_i \oplus B_i$)
- $A \wedge B = 11001001$ (Let this be $G$, where $G_i = A_i \wedge B_i$)

For 8-bit binary addition, the sum bit $Sum_i$ at position $i$ and the carry-out bit $c\_out\_i$ are determined by the individual bits $A_i$, $B_i$, and the carry-in bit $c\_in\_i$. The relations are:

- $Sum_i = A_i \oplus B_i \oplus c\_in\_i$
- $c\_out\_i = (A_i \wedge B_i) \vee (A_i \wedge c\_in\_i) \vee (B_i \wedge c\_in\_i)$

We can simplify these using $P_i$ and $G_i$:

- $Sum_i = P_i \oplus c\_in\_i$
- $c\_out\_i = G_i \vee (P_i \wedge c\_in\_i)$

We can trace the carries and sum bits from the least significant bit (LSB, bit 0) to the most significant bit (MSB, bit 7). The initial carry $c\_in\_0$ is 0.

Step-by-Step Calculation of A + B and Flags:

| **Bit** $i$ | $P_i = (A \oplus B)_i$ | $G_i = (A \wedge B)_i$ | $c\_in_i$ | $Sum_i = P_i \oplus c\_in_i$ | $c\_out_i = G_i \vee (P_i \wedge c\_in_i)$ |
|---|---|---|---|---|---|
| 0 | 0 | 1 | 0 | $0 \oplus 0 = \mathbf{0}$ | $1 \vee (0 \wedge 0) = \mathbf{1}$ |
| 1 | 0 | 0 | 1 | $0 \oplus 1 = \mathbf{1}$ | $0 \vee (0 \wedge 1) = \mathbf{0}$ |
| 2 | 1 | 0 | 0 | $1 \oplus 0 = \mathbf{1}$ | $0 \vee (1 \wedge 0) = \mathbf{0}$ |
| 3 | 0 | 1 | 0 | $0 \oplus 0 = \mathbf{0}$ | $1 \vee (0 \wedge 0) = \mathbf{1}$ |
| 4 | 1 | 0 | 1 | $1 \oplus 1 = \mathbf{0}$ | $0 \vee (1 \wedge 1) = \mathbf{1}$ |
| 5 | 1 | 0 | 1 | $1 \oplus 1 = \mathbf{0}$ | $0 \vee (1 \wedge 1) = \mathbf{1}$ |
| 6 | 0 | 1 | 1 | $0 \oplus 1 = \mathbf{1}$ | $1 \vee (0 \wedge 1) = \mathbf{1}$ |
| 7 | 0 | 1 | 1 | $0 \oplus 1 = \mathbf{1}$ | $1 \vee (0 \wedge 1) = \mathbf{1}$ |

From the table:
The sum $A + B$ (represented by $Sum_7 Sum_6 \ldots Sum_0$) is `11000110` in 8-bit binary.

1. Sum $A + B$ expressed in 8-bit two's complement notation:
$A + B = 11000110$

To convert this to decimal: since the MSB is 1, it is negative.
Two's complement of 11000110: invert $\rightarrow$ 00111001, add 1 $\rightarrow$ 00111010.
Decimal value $= 32 + 16 + 8 + 2 = 58$.
Therefore, 11000110 represents $-58$.

2. Carry Flag (CF) after A + B on x86:
The Carry Flag is set if there is a carry-out from the MSB. From the table, $c\_out\_7 = 1$.
Therefore, CF = 1 (set).

3. Overflow Flag (OF) after A + B on x86:
Overflow is set if $c\_in\_7 \oplus c\_out\_7 = 1$. From the table, $c\_in\_7 = 1$, $c\_out\_7 = 1$, so $OF = 1 \oplus 1 = 0$.
Therefore, OF = 0 (not set).

Final Answer:

- Sum $A + B$: `11000110` ($-58$ in decimal)
- Carry Flag: Set (CF = 1)
- Overflow Flag: Not set (OF = 0)

**Question Summary and Relevance:** The question under consideration asks students to compute the sum of two unknown 8-bit two's complement numbers, $A$ and $B$, given only their bitwise XOR ($A \oplus B$) and AND ($A \wedge B$) results. Students must then express $A + B$ in 8-bit two's complement notation and determine the states of the carry and overflow flags on an x86 system. This problem is a classic test of digital arithmetic reasoning, requiring careful use of propagate ($P_i$) and generate ($G_i$) signals, correct bit-by-bit carry calculation, and strict adherence to conventional bit indexing (LSB =

bit 0 through MSB = bit 7). It highlights the importance of consistent bit ordering when performing flag analysis, since arithmetic correctness and flag determination depend on accurate propagation of carries from least significant to most significant bits.

**Analysis of Model Response:** The model response approach reveals a misunderstanding rooted in non-standard bit indexing patterns. While they correctly introduced propagate and generate definitions, their subsequent tracing of carries and sums implicitly reversed or misapplied the conventional numbering scheme. By failing to consistently treat the least significant bit as index 0 and the most significant bit as index 7, the student introduced misalignment between bit positions and the carries they computed. This mistake corrupted both the intermediate arithmetic reasoning and the interpretation of overflow and carry flags. The response therefore illustrates a key failure mode: even with correct formulas, adopting an unconventional or inconsistent bit indexing convention undermines the entire analysis, leading to incorrect conclusions about the final result and flag states.

## C.4 FAILURE MODE 3: MODELING AND TRACKING SYSTEM STATE

### C.4.1 EXAMPLE 1

---

#### Example: Test-and-Set States

**Context:**

You are writing a queue to be used in a multi-producer/single-consumer application. (Producer threads write messages that are read by one consumer.) We assume here a queue with infinite space. The basic code is shown below.

TST rs, Imm(rt) is the test-and-set instruction, which atomically loads the value at Imm(rt) into rs, and if the value is zero, updates the memory location at Imm(rt) to 1. This atomic instruction is useful for implementing locks: a value of 1 at the memory location indicates that someone holds the lock, and a value of 0 means the lock is free.

Producer pushes a message onto queue: (memory operations in bold)

```
void push(int** tail_ptr, int* tail_write_lock, int message) {
    while (lock_try(tail_write_lock) == false);
    **tail_ptr = message;
    *tail_ptr++;
    lock_release(tail_write_lock);
}

# R1 - contains address of data to enqueue
# R2 - contains the address of the tail pointer of queue
# R3 - address of tail pointer write lock
P1 SpinLock:TST R4, 0(R3)  # try to acquire tail write lock
P2  BNEZ R4, R4, SpinLock
P3  LD R4, 0(R2)  # get tail pointer
P4  ST R1, 0(R4)  # write message to tail
P5  ADD R4, R4, 4  # update tail pointer
P6  ST R4, 0(R2)
P7  ST R0, 0(R3)   # release lock

int pop(int** head_ptr, int** tail_ptr) {
    while (*head_ptr == *tail_ptr);
    int message = **head_ptr;
    *head_ptr++;
    return message;
}

# R1 - will receive address contained in message
# R2 - contains the address of the head pointer of queue
```

---

```
# R3 – contains the address of the tail pointer of the queue
C1 Retry: LD R4, 0(R2)  # get head pointer
C2  LD R5, 0(R3)  # get tail pointer
C3  SUB R5, R4, R5  # is there a message?
C4  BNEZ R5, Pop
C5  JMP Retry
C6 Pop: LD R1, 0(R4)  # read message from queue
C7  ADD R4, R4, 4  # update head pointer
C8  ST R4, 0(R2)
```

**Question:**

We'll now focus on the tail pointer only. Assuming a MSI invalidate coherence protocol, show the state of the tail pointer in the producer and consumer cache after each operation in the sequence below. Show any data or permissions transfers, e.g. "Memory→C" or "C invalidates P".

How many state transitions occur per message in the steady state?

| Operation | Producer tail pointer state | Consumer tail pointer state | Transfers |
|---|---|---|---|
| | I | I | |
| P1 TST try lock | | | |
| P3 LD tail_ptr | | | |
| P4 ST message | | | |
| P6 ST new_tail | | | |
| P7 ST release lock | | | |
| C1 LD head_ptr | | | |
| C2 LD tail_ptr | | | |
| C6 LD message | | | |
| C7 ST new_head | | | |
| P1 TST try lock | | | |
| P3 LD tail_ptr | | | |
| P4 ST message | | | |
| P6 ST new_tail | | | |
| P7 ST release lock | | | |
| C1 LD head_ptr | | | |
| C2 LD tail_ptr | | | |
| C6 LD message | | | |
| C7 ST new_head | | | |

How many state transitions occur per message in the steady state?

**Correct Solution:**

| Operation | Producer tail pointer state | Consumer tail pointer state | Transfers |
|---|---|---|---|
| | I | I | |
| P1 TST try lock | | | |
| P3 LD tail_ptr | S | | P ← Memory |
| P4 ST message | | | |
| P6 ST new_tail | M | | |
| P7 ST release lock | | | |
| C1 LD head_ptr | | | |
| C2 LD tail_ptr | S | S | C ← P |
| C6 LD message | | | |
| C7 ST new_head | | | |
| P1 TST try lock | | | |
| P3 LD tail_ptr | | | |
| P4 ST message | | | |
| P6 ST new_tail | M | I | P invalidates C |
| P7 ST release lock | | | |
| C1 LD head_ptr | | | |
| C2 LD tail_ptr | S | S | C ← P; Memory ← P |
| C6 LD message | | | |
| C7 ST new_head | | | |

How many state transitions occur per message in the steady state? 2 (second half of table)

**Incorrect Frontier Model Response:**

To analyze the state transitions of the tail pointer in the producer and consumer cache, we need to carefully follow the sequence of operations for both the producer and the consumer, assuming an MSI (Modified, Shared, Invalid) cache coherence protocol.

*Steady-State Analysis of Tail Pointer:*

*Producer Side:* 1. P1 TST try lock: - State Transition: The producer tries to acquire the lock on the tail pointer. - Cache State: If the lock is free (cache line state is I or S), it will transition to M (Modified) when the producer acquires the lock.

2. P3 LD tail_ptr: - State Transition: The producer loads the tail pointer into its cache. - Cache State: If the tail pointer is not in the cache, it will transition from I (Invalid) to S (Shared) or E (Exclusive) if it is not shared.

3. P4 ST message: - State Transition: The producer writes the message to the tail location. - Cache State: This write will transition the cache line to M (Modified).

4. P6 ST new_tail: - State Transition: The producer updates the tail pointer. - Cache State: The tail pointer cache line will transition to M (Modified) in the producer's cache.

5. P7 ST release lock: - State Transition: The producer releases the lock. - Cache State: The lock cache line will transition back to I (Invalid) or S (Shared) depending on other cores' states.

*Consumer Side:* 1. C2 LD tail_ptr: - State Transition: The consumer loads the tail pointer into its cache. - Cache State: If the tail pointer is not in the cache or is invalid, it will transition from I to S (Shared) or E (Exclusive).

—

*MSI State Transitions per Message in Steady State:*

1. Producer Acquires Lock (P1 TST try lock): - Transition from I or S to M (1 transition).

2. Producer Loads Tail Pointer (P3 LD tail_ptr): - Transition from I to S or E (1 transition).

3. Producer Writes Message (P4 ST message): - Transition from S or E to M (1 transition).

4. Producer Updates Tail Pointer (P6 ST new_tail): - Transition from S or E to M (1 transition).

5. Producer Releases Lock (P7 ST release lock): - Transition from M to I (1 transition).

6. Consumer Loads Tail Pointer (C2 LD tail_ptr): - Transition from I to S (1 transition), potentially causing an invalidation in the producer's cache (1 additional transition).

—

*Data/Permission Transfers:* - Memory to Cache (Memory → C): When the tail pointer is initially loaded into the cache. - Cache Invalidation (C invalidates P): When the consumer loads the tail pointer, it may invalidate the producer's copy if it was in M state.

—

*Summary of State Transitions per Message:*
1. Producer: - Acquire lock: I/S → M (1 transition) - Load tail pointer: I → S/E (1 transition) - Write message: S/E → M (1 transition) - Update tail pointer: S/E → M (1 transition) - Release lock: M → I (1 transition)
2. Consumer: - Load tail pointer: I → S (1 transition) - Invalidate producer's tail pointer: M → I (1 transition)
Total State Transitions per Message: - 5 transitions on the producer side. - 2 transitions on the consumer side.
Total: 7 state transitions per message in the steady state.

| Operation | Producer Cache State Transition | Consumer Cache State Transition | Transfers |
|---|---|---|---|
| P1 TST try lock | I/S → M | - | - |
| P3 LD tail_ptr | I → S/E | - | Memory → P |
| P4 ST message | S/E → M | - | - |
| P6 ST new_tail | S/E → M | - | - |
| P7 ST release lock | M → I | - | - |
| C2 LD tail_ptr | - | I → S | Memory → C, C invalidates P |

*Conclusion:* In the steady state, there are 7 state transitions per message considering both the producer and consumer operations. Each message involves multiple cache line state changes and potential invalidations to maintain coherence.

**Question Summary and Relevance:** This question assesses whether models can concretely reason about cache coherence behavior in a multicore setting. By stepping through the producer–consumer interaction under an MSI protocol, it forces careful tracking of how load, store, and lock operations map to specific cache-line state transitions and invalidations. The question relies on architectural intuition – seeing how simple pointer updates or lock releases induce multiple coherence transitions highlights the hidden costs of synchronization and communication, which often create a scalability bottleneck in shared-memory systems.

**Analysis of Model Response:** The model incorrectly asserted that the producer's test-and-set (TST) on the lock variable alters the MSI state of the tail pointer. This reflects a misunderstanding of coherence granularity: the MSI protocol tracks cache lines at the level of physical addresses, not program-level abstractions. Since the lock and the tail pointer occupy distinct memory locations, a TST on the lock can only trigger transitions on the lock's cache line. Attributing such transitions to the tail pointer indicates a conceptual error in tracking the architectural system state, where ordering effects of synchronization are mistakenly interpreted as coherence events on unrelated data.

C.4.2 EXAMPLE 2

Example: Identifying Data Dependencies from Instruction Trade

**Context:**

```
addi x2, x0, 17
addi x8, x0, 14
add x1, x1, x2
and x3, x4, x1
addi x1,x2, 5
add x3, x3, x3
sw x3, 4(x1)
add x7, x5, x4
sw x7, 0(x5)
add x11, x2, x3
add x9, x7, x8
add x9, x9, x9
```

```
ld x11, 12(x8)
ld x9, 0(x5)
```

**Question:**
Consider the above RISC-V instruction sequence. The total number of instruction pairs having Read after Write (RAW) dependencies on registers and memory is ?

**Correct Solution:**
There are 13 RAW dependencies on registers and 2 RAW dependencies on memory (for memory addresses 0(x5) and 26)

**Incorrect Frontier Model Response:**
I need to analyze this RISC-V instruction sequence to identify all Read-after-Write (RAW) dependencies on both registers and memory. Let me work through this step-by-step, numbering each instruction and tracking dependencies: Instructions:

```
addi x2, x0, 17     # x2 = 0 + 17 = 17
addi x8, x0, 14     # x8 = 0 + 14 = 14
add x1, x1, x2      # x1 = x1 + x2
and x3, x4, x1      # x3 = x4 & x1
addi x1, x2, 5      # x1 = x2 + 5
add x3, x3, x3      # x3 = x3 + x3
sw x3, 4(x1)        # store x3 to memory[x1 + 4]
add x7, x5, x4      # x7 = x5 + x4
sw x7, 0(x5)        # store x7 to memory[x5 + 0]
add x11, x2, x3     # x11 = x2 + x3
add x9, x7, x8      # x9 = x7 + x8
add x9, x9, x9      # x9 = x9 + x9
ld x11, 12(x8)      # load x11 from memory[x8 + 12]
ld x9, 0(x5)        # load x9 from memory[x5 + 0]
```

RAW Dependencies on Registers:
Instruction 3 reads x2 (written by instruction 1) → RAW: (1,3)
Instruction 4 reads x1 (written by instruction 3) → RAW: (3,4)
Instruction 5 reads x2 (written by instruction 1) → RAW: (1,5)
Instruction 6 reads x3 (written by instruction 4) → RAW: (4,6)
Instruction 7 reads x3 (written by instruction 6) → RAW: (6,7)
Instruction 7 reads x1 (written by instruction 5) → RAW: (5,7)
Instruction 9 reads x7 (written by instruction 8) → RAW: (8,9)
Instruction 10 reads x2 (written by instruction 1) → RAW: (1,10)
Instruction 10 reads x3 (written by instruction 6) → RAW: (6,10)
Instruction 11 reads x7 (written by instruction 8) → RAW: (8,11)
Instruction 11 reads x8 (written by instruction 2) → RAW: (2,11)
Instruction 12 reads x9 (written by instruction 11) → RAW: (11,12)
Instruction 13 reads x8 (written by instruction 2) → RAW: (2,13)
RAW Dependencies on Memory:
Instruction 14 reads from memory[x5 + 0] (written by instruction 9) → RAW: (9,14)
Summary:
Register RAW dependencies: 13 pairs
Memory RAW dependencies: 1 pair
Total RAW dependencies: 14
The answer is 14.

**Question Summary and Relevance:** This question evaluates a student's ability to identify and count Read-After-Write (RAW) dependencies in a given RISC-V instruction sequence, both at the register level and the memory level. Correctly solving it requires carefully tracking dataflow dependencies across instructions and determining when a later instruction consumes a value produced by an earlier one thus creating a stall in the processor pipeline.

**Analysis of Model Response:** The model successfully tracked register states but failed to capture memory updates, resulting in an incomplete representation of the system state. This limitation led to errors in identifying Read-after-Write (RAW) dependencies and ultimately produced an incorrect answer, illustrating the model's weakness in handling memory-based interactions and implicit architectural information.

## C.5 FAILURE MODE 4: SENSITIVITY TO QA MODALITY

### C.5.1 EXAMPLE 1

In the example below, the model failed to properly interpret the RAT (Register Alias Table) and reservation station mappings shown in the diagram, particularly overlooking the source tags and sequential ordering information. This caused it to incorrectly mark valid destination registers as unknown. However, the diagram provided sufficient information for complete mapping if the model had properly cross-referenced the source tags with RAT entries and considered the program execution order across processing units.

---

**Example: Filling in Instruction Fields From Out-of-Order Execution Snapshot**

**Context:**
The diagram below shows a snapshot at a particular point in time of various parts (reservation stations and register alias table) of the microarchitecture for an implementation supporting out-of-order execution in the spirit of Tomasulo's Algorithm. Note that there is an adder and a multiplier in this machine. The processor is supplied with a seven instruction program following reset. The state below was captured at some point in time during the execution of these seven instructions. Anything marked with a - is unknown and can't be relied upon for your answer. You should assume that the bottommost instruction in the reservation station arrived earliest and the topmost instruction in the reservation station arrived last.

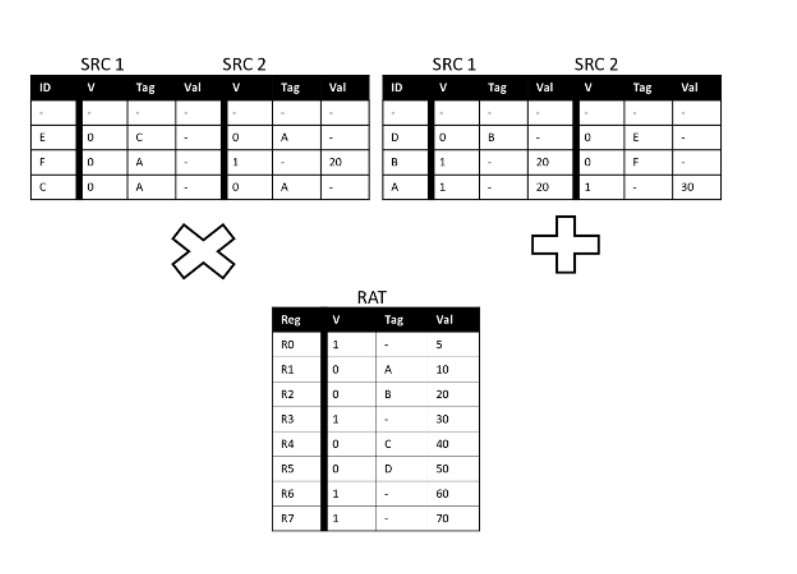

**Question:**
Fill in the instruction opcodes, source, and destination registers in the table below.

**Correct Solution:**

| OP | DEST | SRC1 | SRC2 |
|----|------|------|------|
| ADD | R3 | R1 | R2 |

---

```
ADD    R1    R2    R3
MUL    R4    R1    R1
MUL    R5    R2    R1
ADD    R2    R2    R5
MUL    R5    R4    R1
ADD    R5    R2    R5
```

**Incorrect Frontier Model Response:**
Answer to Out-of-Order Execution Instruction Reconstruction

To solve this problem, I need to analyze the reservation stations and Register Alias Table (RAT) to determine what instructions are being executed. I'll examine the dependencies between instructions and map the data flow.

Analysis of the RAT
From the RAT, I can determine which reservation stations are producing values for specific registers:
- R1: Waiting for result from RS A (Tag = A)
- R2: Waiting for result from RS B (Tag = B)
- R4: Waiting for result from RS C (Tag = C)
- R5: Waiting for result from RS D (Tag = D)

Analysis of Reservation Stations
*Addition Reservation Stations:*
1) RS A (earliest): Both sources are valid with values 20 and 30. Will produce value for R1. Operation must be ADD R1, ?, ?; since both values are ready: ADD R1, #20, #30 (or using registers with those values).
2) RS B: SRC1 valid with value 20; SRC2 waiting for RS F. Will produce value for R2. Operation must be ADD R2, #20, ?.
3) RS D: SRC1 waiting for RS B (which produces R2); SRC2 waiting for RS E. Will produce value for R5. Operation must be ADD R5, R2, ?.

*Multiplication Reservation Stations:*
1) RS C (earliest): Both sources waiting for RS A (which produces R1). Will produce value for R4. Must be using the same value twice: MUL R4, R1, R1.
2) RS F: SRC1 waiting for RS A (which produces R1); SRC2 valid with value 20. Destination register unknown. Operation must be MUL ?, R1, #20.
3) RS E: SRC1 waiting for RS C (which produces R4); SRC2 waiting for RS A (which produces R1). Destination register unknown. Operation must be MUL ?, R4, R1.

Completed Table
Reservation Station A: Opcode = ADD, Source1 = #20, Source2 = #30, Destination = R1
Reservation Station B: Opcode = ADD, Source1 = #20, Source2 = ?, Destination = R2
Reservation Station C: Opcode = MUL, Source1 = R1, Source2 = R1, Destination = R4
Reservation Station D: Opcode = ADD, Source1 = R2, Source2 = ?, Destination = R5
Reservation Station E: Opcode = MUL, Source1 = R4, Source2 = R1, Destination = ?
Reservation Station F: Opcode = MUL, Source1 = R1, Source2 = #20, Destination = ?

Note: For the unknown destination registers (marked with ?), I cannot determine their values from the provided information, as they are not shown in the RAT excerpt.

### C.5.2   EXAMPLE 2

**Example: Deadlock-Free Min-Cut**

**Question:**
The image shows a channel dependency graph for a network. What is the smallest set of edges that, when removed, eliminates deadlocks in the network?

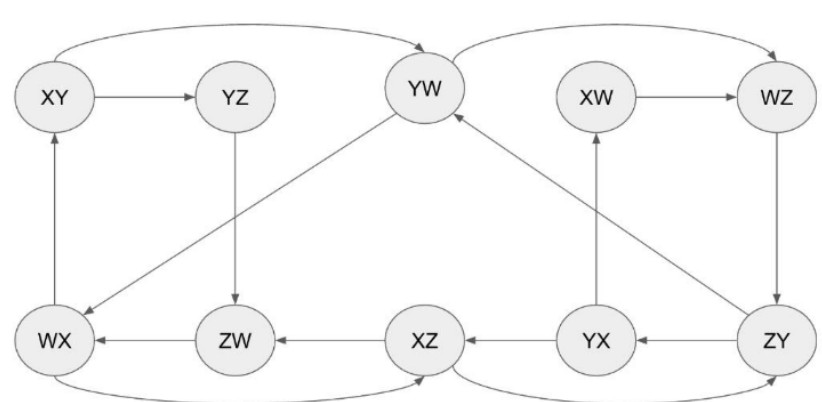

- $ZW \to WX, ZY \to YX$
- $ZW \to WX, ZY \to YX, YW \to WX, ZY \to YW$
- $YW \to WX, ZY \to YW, XY \to YZ, XW \to WZ$
- $ZW \to WX$

**Correct Solution:**
Eliminating edges mentioned in Option B removes all cycles from the graph and hence eliminates deadlocks. Since there are 8 cycles in the given graph, we need to eliminate at least 4 edges.

**Incorrect Frontier Model Response:**
Looking at this channel dependency graph, I need to identify the smallest set of edges that, when removed, eliminates all deadlocks.

First, let me identify the cycles in the graph, as deadlocks occur when there are circular dependencies:
– XY → YZ → ZW → WX → XY (4-node cycle)
– YW → WX → XZ → ZY → YW (4-node cycle)
– XW → WZ → ZY → YX → XW (4-node cycle)

To eliminate deadlocks, I need to break all cycles. The key is finding the minimum number of edges that participate in multiple cycles, as removing such edges breaks multiple cycles simultaneously.

Let me analyze which edges appear in multiple cycles:
– ZW→WX: Appears in cycles 1 and 2
– ZY→YX: Appears in cycles 2 and 3
– YW→WX: Appears in cycle 2
– XZ→ZY: Appears in cycles 2 and 3

Looking at the options:
– ZW→WX, ZY→YX: This removes edges that break all three cycles.
Removes ZW→WX (breaks cycles 1 and 2).
Removes ZY→YX (breaks cycles 2 and 3).
All cycles are broken with just 2 edges.

– ZW→WX, ZY→YX, YW→WX, ZY→YW: This is overkill (4 edges).
– YW→WX, ZY→YW, XY→YZ, XW→WZ: This is also overkill (4 edges).
– ZW→WX: This only breaks cycles 1 and 2, leaving cycle 3 intact.

The smallest set that eliminates all deadlocks is ZW→WX, ZY→YX (option 1), as it breaks all cycles with just 2 edges.

The answer is the first option: ZW→WX, ZY→YX.

The question tests the models' understanding of deadlock freedom conditions. The model failed to properly track the cycles in the graph and performed an oversimplified traversal, ignoring edge cases/cycles where deadlock could have occurred.

## C.6 SUCCESSFUL MODEL RESPONSE

In the example below, we illustrate a scenario where a model successfully generates the correct answer to a question in the dataset. Although the responses

the frontier model's response aligns conceptually with the answer key by addressing the same advantages using more verbose phrasing. Their point about "Reduced Bus Contention and Cache Invalidation" corresponds to "No ping-pong" in the answer key - both describe how LdR/StC prevents unnecessary cache line bouncing between processors by avoiding writes on failed attempts. Similarly, their emphasis on "No Bus Locking During Critical Section" matches "Everything is local" - both highlight that LdR/StC operations can be managed through local reservation tracking without requiring global bus coordination or centralized locking mechanisms.

---

**Example: Advantages of LdR/StC pair of instructions**

**Context:**
One of the common instruction sequences used for synchronizing several processors are the LOAD RESERVE/STORE CONDITIONAL pair (from now on referred to as LdR/StC pair). The LdR instruction reads a value from the specified address and sets a local reservation for the address. The StC attempts to write to the specified address provided the local reservation for the address is still held. If the reservation has been cleared the StC fails and informs the CPU.

**Question:**
Give two reasons why the LdR/StC pair of instructions is preferable over atomic read-test-modify instructions such as the TEST & SET instruction.

**Correct Solution:**

1. Bus doesn't need to be aware of them.
2. Everything is local.
3. No ping-pong.
4. No extra hardware (tied to 1)

**Correct Frontier Model Response:**
The LOAD RESERVE/STORE CONDITIONAL (LdR/StC) pair is preferable over atomic read-test-modify instructions like TEST&SET for the following reasons:

1. Reduced Bus Contention and Cache Invalidation
TEST&SET performs an atomic read-modify-write operation, which always writes to memory even if the lock acquisition fails. This results in frequent bus traffic and cache invalidations for contending processors, degrading performance.

---

LdR/StC avoids unnecessary writes: – The StC instruction only writes to memory if the reservation is still valid. – Failed StC attempts do not generate writes, reducing bus traffic and cache coherence overhead.

2. No Bus Locking During Critical Section
TEST&SET requires locking the memory bus for the entire atomic operation, blocking all other memory accesses during execution. This creates a scalability bottleneck in multi-processor systems.

LdR/StC operates without bus locking: – LdR is a standard load instruction, and StC is a conditional store. – The hardware tracks reservations passively (e.g., via cache coherence protocols), allowing concurrent memory accesses by other processors between the LdR and StC. – This enables better parallelism and scalability for contended synchronization.

## C.7 EXAMPLE QUESTIONS FOR CASE STUDY

This section presents concrete questions from QuArch that are similar to the types of questions asked in the memory controller design case study (Section 4.4). Please refer to Appendix D.11 for the prompt used for this case study. The prompt asks for the LLM to analyze the memory trace and design parameters to understand their architectural implications on power. The questions in this section require similar reasoning skills.

### C.7.1 ANALYZING MEMORY TRACES

---
**Example: DRAM Row Access Trace Optimization**

**Context:**
Recall from your required reading on Tiered-Latency DRAM that there is a near and far segment, each containing some number of rows. Assume a very simplified memory model where there is just one bank and there are two rows in the near segment and four rows in the far segment. The time to activate and precharge a row is 25ns in the near segment and 50ns in the far segment. The time from start of activation to reading data is 10ns in the near segment and 15ns in the far segment. All other timings are negligible for this problem. Given the following memory request stream, determine the optimal assignment (minimize average latency of requests) of rows in the near and far segment (assume a fixed mapping where rows cannot migrate, a closed-row policy, and the far segment is inclusive).
time 0ns: row 0 read
time 10ns: row 1 read
time 100ns: row 2 read
time 105ns: row 1 read
time 200ns: row 3 read
time 300ns: row 1 read
**Question:**
What rows would you place in near segment? Hint: draw a timeline.
**Correct Solution:**
rows 0 and 2. see above

---
**Example: Tiered-Latency DRAM Design based on Access Pattern**

**Context:**
You would like to understand the configuration of the DRAM subsystem of a computer using reverse engineering techniques. Your current knowledge of the particular DRAM subsystem is limited to the following information:
* The physical memory address is 16 bits.

---

* The DRAM subsystem consists of a single channel and 4 banks.
* The DRAM is byte-addressable.
* The most-significant 2 bits of the physical memory address determine the bank.
* The DRAM command bus operates at 500 MHz frequency.
* The memory controller issues commands to the DRAM in such a way that no command for servicing a later request is issued before issuing a READ command for the current request, which is the oldest request in the request buffer. For example, if there are requests A and B in the request buffer, where A is the older request and the two requests are to different banks, the memory controller does not issue an ACTIVATE command to the bank that B is going to access before issuing a READ command to the bank that A is accessing.

You realize that you can observe the memory requests that are waiting to be serviced in the request buffer. At a particular point of time, you take the snapshot of the request buffer and you observe the following requests in the request buffer.

Requests in the request buffer (in descending order of request age, where the oldest request is on the top):

Read 0x4C80
Read 0x0140
Read 0x4EC0
Read 0x8000
Read 0xF000
Read 0x803F
Read 0x4E80

At the same time you take the snapshot of the request buffer, you start probing the DRAM command bus. You observe the DRAM command type and the cycle (relative to the first command) at which the command is seen on the DRAM command bus. The following are the DRAM commands you observe on the DRAM bus while the requests above are serviced.

Cycle 0 — PRECHARGE
Cycle 6 — ACTIVATE
Cycle 10 — READ
Cycle 11 — READ
Cycle 21 — PRECHARGE
Cycle 27 — ACTIVATE
Cycle 31 — READ
Cycle 32 — ACTIVATE
Cycle 36 — READ
Cycle 37 — READ
Cycle 38 — READ
Cycle 42 — PRECHARGE
Cycle 48 — ACTIVATE
Cycle 52 — READ

To improve performance, you decide to implement the idea of Tiered-Latency DRAM (TL-DRAM) in the DRAM chip. Assume that a bank consists of a single subarray. With TL-DRAM, an entire bank is divided into a near-segment and far-segment. When accessing a row in the near-segment, the ACTIVATE-to-READ latency reduces by 2 cycles and the ACTIVATE-to-PRECHARGE latency reduces by 5 cycles. When accessing a row in the far-segment, the ACTIVATE-to-READ latency increases by 1 cycle and the ACTIVATE-to-PRECHARGE latency increases by 2 cycles.

Assume that the rows in the near-segment have smaller row ids compared to the rows in the far-segment. In other words, physical memory row addresses 0 through N - 1 are the near-segment rows, and physical memory row addresses N through M - 1 are the far-segment rows.

**Question:**

If the above DRAM commands are issued 5 cycles faster with TL-DRAM compared to the baseline (the last command is issued in cycle 47), how many rows are in the near-segment? Show your work.

**Correct Solution:**

59 rows have to be in the near segment.

Explanation. There should 59 rows in the near-segment (rows 0 to 58) since rows until row id 58 need to be accessed with low latency to get 5 cycle reduction. Rows 59 and 192 are in the far-segment, thus latency for accessing them increases slightly.

Here is the new command trace:

Cycle 0 – PRECHARGE - Bank 1
Cycle 6 – ACTIVATE - Bank 1, Row 50, near segment
Cycle 8 – READ - Bank 1
Cycle 9 – READ - Bank 0
Cycle 16 – PRECHARGE - Bank 1
Cycle 22 – ACTIVATE - Bank 1, Row 59, far segment
Cycle 27 – READ - Bank 1
Cycle 28 – ACTIVATE - Bank 2, Row 0
Cycle 30 – READ - Bank 2
Cycle 31 – READ - Bank 3
Cycle 32 – READ - Bank 2
Cycle 39 – PRECHARGE - Bank 1
Cycle 45 – ACTIVATE - Bank 1, Row 58, near segment
Cycle 47 – READ - Bank 1

### C.7.2 MEMORY CONTROLLER CONFIGURATIONS

**Example: DRAM Command Design**

**Context:**
You are given a memory system that has four channels, and each channel has two ranks of DRAM chips. A separate memory controller controls each memory channel. Each rank of DRAM contains eight banks. A bank contains R rows. Each row in one bank is 8KB. The minimum retention time among all DRAM rows in the system is 64 ms. In order to ensure that no data is lost, every DRAM row is refreshed once per 64 ms. Refresh of each row is initiated by a command from the memory controller. The command refreshes only the specified row. The command occupies the command bus on the associated memory channel for 5 ns and the associated bank for 40 ns.

We define refresh utilization of a resource (such as a bus or a memory bank) as the fraction of total time for which a resource is occupied by a refresh command.

**Question:**
How can you reduce the command bus utilization due to refreshes? You cannot change the refresh rate when answering this question.

**Correct Solution:**
Have each command be responsible for multiple row refreshes.

**Example: DRAM Heirarchy Configuration**

**Context:**
Recall from your required reading on Tiered-Latency DRAM that there is a near and far segment, each containing some number of rows. Assume a very simplified memory model where there is just one bank and there are two rows in the near segment and four rows in the far segment. The time to activate and precharge a row is 25ns in the near segment and 50ns in the far segment. The time from start of activation to reading data is 10ns in the near segment and 15ns in the far segment. All other timings are negligible for this problem. Given the following memory request stream, determine the optimal assignment (minimize average latency of requests) of rows in the near and far segment (assume a fixed mapping where rows cannot migrate, a closed-row policy, and the far segment is inclusive).

time 0ns: row 0 read
time 10ns: row 1 read
time 100ns: row 2 read
time 105ns: row 1 read

time 200ns: row 3 read
time 300ns: row 1 read
**Question:**
Assume now that the mapping is dynamic. What are the tradeoffs of an exclusive design vs. an inclusive design? Name one advantage and one disadvantage for each.
**Correct Solution:**
Exclusive requires swapping, but can use nearly full capacity of DRAM. Inclusive, the opposite.

---

### Example: DRAM Sizing Optimization

**Context:**
You are given a memory system that has four channels, and each channel has two ranks of DRAM chips. A separate memory controller controls each memory channel. Each rank of DRAM contains eight banks. A bank contains R rows. Each row in one bank is 8KB. The minimum retention time among all DRAM rows in the system is 64 ms. In order to ensure that no data is lost, every DRAM row is refreshed once per 64 ms. Refresh of each row is initiated by a command from the memory controller. The command refreshes only the specified row. The command occupies the command bus on the associated memory channel for 5 ns and the associated bank for 40 ns.

We define refresh utilization of a resource (such as a bus or a memory bank) as the fraction of total time for which a resource is occupied by a refresh command.

**Question:**
Only changing the number of rows per bank, find the maximum number of rows per bank for which either the bank utilization or the command bus utilization reaches 100

**Correct Solution:**
Because the command bus utilization will reach 100R*5ns*2ranks*8banks/64ms = 1 R = 800,000 We find that with 800k rows per bank, the command bus utilization reaches 100

## D   PROMPT TEMPLATES

### D.1   LLM PROMPT FOR STUDENT LLM

### Student LLM Prompt

You are an expert computer architect taking an exam. You will be provided with a question and its context. Your task is to provide a clear, accurate, and well-reasoned answer to the question.
Please provide your answer in a structured format that clearly addresses the question. If the question involves calculations, show your work step-by-step. If it involves diagrams or tables, describe them clearly.
Remember to:

1. Read the question carefully and understand what is being asked

2. Use the provided context to inform your answer

3. Show your reasoning and work where appropriate

4. Be precise and accurate in your response

5. If you're unsure about something, acknowledge the uncertainty

Question Context:
{context}
{context_images_placeholder}

Question:
{question}

> Please provide your answer:

This prompt is used to simulate the "student LLM," where models act as exam-takers solving architecture questions. The emphasis is on structured, step-by-step reasoning, accuracy, and clarity, mirroring how a human student would respond to technical exam questions.

### D.2   LLM PROMPT FOR MCQ RESPONSE

> **MCQ Prompt**
>
> You are an expert computer architect solving multiple choice questions. Please read the following question and select the best answer from the choices provided.
> Question:
> {question}
>
> Choices:
> {choices}
>
> Please conclude your response with a JSON object containing your final answer. The JSON object must match this schema: {"final_answer": "⟨ A, B, C, or D⟩"}.

This prompt is designed for multiple-choice evaluation, where correctness can be measured directly. The JSON-constrained output ensures answers are machine-readable and easy to score at scale.

### D.3   LLM PROMPT FOR FRQ LLM-AS-A-JUDGE

> **LLM-as-a-Judge Prompt**
>
> You are an expert computer architect acting as an exam grader to evaluate the quality of an answer to a computer architecture question. You will be provided with:
>
> 1. The original question and context
> 2. The correct solution
> 3. A student's answer to the question
>
> Your task is to carefully evaluate whether the student's answer is correct, partially correct, or incorrect by comparing it to the provided solution.
> Evaluation criteria:
>
> - CORRECT: The answer is accurate, complete, and demonstrates proper understanding
> - PARTIALLY-CORRECT: The answer shows some understanding but has significant errors or is incomplete
> - INCORRECT: The answer is fundamentally wrong or shows major misunderstandings
>
> Consider:
>
> - Mathematical accuracy
> - Conceptual understanding
> - Completeness of the response solely in relation to the question being asked
> - Logical reasoning
> - Whether the answer addresses what was actually asked
>
> Be fair but rigorous in your evaluation. If you are unsure, err on the side of being more critical.

Question Context:
{context}
{context_images_placeholder}

Question:
{question}

Correct Solution:
{solution}
{solution_images_placeholder}

Student's Answer:
{student_answer}

Please evaluate the student's answer and provide your reasoning. At the end of your response, write exactly one of the following words in all caps on a new line: CORRECT, PARTIALLY-CORRECT, or INCORRECT. If you do not end your response with a new line with exactly one of these options, you will not be paid for your work.

This prompt is used to evaluate model outputs under the "LLM-as-a-Judge" paradigm. Here, the model acts as a grader, comparing student answers to reference solutions and deciding between CORRECT, PARTIALLY-CORRECT, or INCORRECT. It enables scalable evaluation of free-response answers while preserving rigor and consistency.

### D.4  LLM PROMPT FOR SKILLS CLASSIFICATION

Skill Classification Prompt

You are a computer architecture expert who is a professional at categorizing exam questions based on the cognitive skill they are testing of a computer architect. Your task is to classify one question at a time into exactly one of the following four categories:

**Recall**: The question asks for a fact, definition, or direct retrieval of knowledge. Answering these questions should typically not require multi-step reasoning and can be directly answered in a single step.

**Analyze**: The question requires deducing, inferring, calculating, or interpreting information based on data or a specific scenario. Answering these questions typically requires some level of multi-step reasoning but does not require invention or innovating upon an existing solution.

**Design**: The question asks you to propose, invent, suggest, or improve a method, system component, or policy. It requires proposing new or improved solutions.

**Implement**: The question requires constructing, coding, or developing a full solution or system based on explicit requirements or specifications. These typically involve providing detailed instructions or actual code (i.e., programmable/executable artifacts).

Sometimes "Analyze" and "Design" can be confused if the question is open-ended. If the main effort is proposing or inventing, choose "Design"; if it's interpreting specific data or information, choose "Analyze".

Additionally, sometimes "Design" and "Implementation" can also be confused. If the answer involves some sort of programmatic implementation or executable artifact then choose "Implement"; if the answer is at a higher-level of abstraction than this, it is likely to be "Design".

> Your output **MUST** be **ONLY** one word, chosen from the following list:
> Recall, Analyze, Design, Implement
>
> Find an example below:
> *Recall*: A ___ cache allows any block of main memory to be placed in any line, eliminating conflict misses but requiring complex associative lookup hardware.
> *Analyze*: A fully associative cache has 4 lines and uses an LRU policy. The following sequence of memory references occurs … What is the overall hit ratio?
> *Design*: Describe a cache replacement policy that would improve the performance of the hybrid memory system more than it would DRAM.
> *Implement*: Create a cache controller that interfaces with both a processor pipeline and a DRAM chip, following the provided Verilog port and signal specifications.
>
> Now, classify this question:
> {question}

We use this prompt to consistently label each exam or benchmark question with the specific cognitive skill it targets. Our rationale in writing it was twofold: (i) provide clear, operational definitions of *Recall*, *Analyze*, *Design*, and *Implement* that are grounded in how architects approach problem solving, and (ii) minimize ambiguity by explicitly addressing common confusions between neighboring categories (e.g., Analyze vs. Design, or Design vs. Implement). This ensures that classification is reliable across reviewers and that the benchmark's skill taxonomy aligns with real-world architectural workflows.

### D.5 LLM Prompt for Architecture Topic Classification

To determine the architecture topic distribution of QUARCH, we employed a two-stage classification process that combines embedding-based similarity search with LLM reasoning for scalable categorization. In the first stage, the top 3 most relevant topic candidates are identified by embedding each question using OpenAI's text-embedding-3-large model and comparing against pre-computed taxonomy topic embeddings via cosine similarity. In the second stage, GPT-4o is used to make the final topic selections from these 3 candidates, providing both best and second-best classifications along with justifications.

This hybrid approach effectively balances accuracy with scalability by leveraging the computational efficiency of embedding-based similarity search for initial filtering while utilizing LLM reasoning capabilities for nuanced final classification decisions. Below is the prompt used for LLM categorization:

> **Architecture Topic Classification Prompt**
>
> You are a computer architecture researcher and expert. You have been asked to categorize the following question into a subfield of computer architecture. Three options have been provided and you must select the top two.
>
> Question: {question}
>
> Categories:
> 1: taxonomy_terms[0]
> 2: taxonomy_terms[1]
> 3: taxonomy_terms[2]
>
> Please provide the exact names of the two categories that you feel best fit this question. First select the best match of the three options and then choose the second best category that matches. You have been told **you have to pick no matter what from the options** and have

to provide your response should be in the format without any additional text or explanation:

[{ "best_selection": "[BEST CATEGORY HERE]",
"justification": "[JUSTIFICATION #1 HERE]" },

{ "second_best_selection": "[SECOND BEST CATEGORY HERE]",
"justification": "[JUSTIFICATION #2 HERE]" } ]

## D.6 LLM PROMPT FOR MCQ GENERATION

---

**MCQ Generation Prompt**

You are a computer architecture professor and expert researcher. You have been provided with the following excerpt about computer architecture:
"{excerpt}"

You have been asked to create one difficult, paraphrased cloze-style format multiple choice question based on this excerpt to test senior computer architects.

The questions will be used for creating an interview test for senior computer architects, and they will not get to read the excerpt for context, so any questions you create should not refer to anything specifically in the excerpt that would make the question unanswerable in the excerpts's absence.

The questions must be precise and clear so they can be answered *definitively*, so avoid using qualifying adjectives or adverbs that would make the answer ambiguous or depend on the context; make sure there is only one correct answer to the question.

Quote *word for word* the sentence(s) of the excerpt context that are comprehensive and self-contained and could be read to justify and support each answer to each question. Your goal is to create good **conceptual** questions that test **conceptual** knowledge from the excerpt.

Here is an example of a good cloze question:
___ is the typical penalty incurred for a branch mispredict.
A) 100 us
B) 5 ms
C) 5 ns
D) 100 ms
Answer: C

Provide your response in this *exact* format with *zero additional characters for formatting* before or after the opening and closing brackets:
[
{
"question": "[CLOSE QUESTION HERE]",
"option A": "[OPTION A HERE]",
"option B": "[OPTION B HERE]",
"option C": "[OPTION C HERE]",
"option D": "[OPTION D HERE]",
"answer": "{specified_answer_choice['Cloze']}",
"context": "[JUSTIFICATION HERE AS CONTEXT]",
"type": "Cloze"
}
]

---

### D.7 LLM Prompts for MCQ Filtering

---

**MCQ Filtering Prompt 1**

You are a computer architecture expert.

You have been asked the following question:
{question}

Here are the options:
{options}

As a computer architecture expert, would you need additional context to correctly answer the question?

Please answer with one word: "YES" or "NO".

Then provide one short sentence to justify your answer reasoning. Return your response in this exact format with zero other characters for formatting before or after:
{
"answer": "[ANSWER HERE: YES/NO]",
"justification": "[JUSTIFICATION HERE]"
}

---

**MCQ Filtering Prompt 2**

You are a computer architecture expert.

You have been asked the following question:
{question}

Here are the options:
{options}

As a computer architecture expert, of the provided options is there *only one* answer that is correct?

Please answer with one word: "YES" or "NO".

Then provide one short sentence to justify your answer reasoning. Return your response in this exact format with zero other characters for formatting before or after:
{
"answer": "[ANSWER HERE: YES/NO]",
"justification": "[JUSTIFICATION HERE]"
}

---

**MCQ Filtering Prompt 3**

You are a new graduate student reading about computer architecture to learn the subject.

You have been given the following quote from a computer architecture excerpt to read for context:
{context}

---

You have now been asked the following question:
{question}

Here are the options:
{options}

Does the provided context you read sufficiently help you answer the question correctly?

Please answer with one word: "YES" or "NO".

Then provide one short sentence to justify why you answered "YES" or "NO". Return your response in this exact format with zero other characters for formatting before or after:
{
"answer": "[ANSWER HERE: YES/NO]",
"justification": "[JUSTIFICATION HERE]"
}

### MCQ Filtering Prompt 4

You are a computer architecture expert.
You have been given the following quote from a computer architecture excerpt for context:
{context}

You have now been asked the following question:
{question}

Here are the options:
{options}

Choose the best correct option and provide your answer justification. Return your response in this exact format with zero other characters for formatting before or after:
{
"answer": "[ANSWER HERE: LETTER OF OPTION]",
"justification": "[JUSTIFICATION HERE]"
}

## D.8 LLM PROMPT FOR EXAM QUESTION TEXT EXTRACTION

### Exam Question Text Extraction Prompt

You are a language model assisting with the digitization of academic exam content. The input is a PDF file containing one problem of a Computer Architecture Assessment. If part of another problem is included, ignore it and only focus on filename. The problem may include any combination of the following: A context paragraph, or just a short statement (e.g., "Convert the number 42 to binary") One or more sub-questions, or be a single standalone question Context for sub-questions separate from the sub-question and separate from the original problem context Multiple questions within a subquestion Point value associations for the problem or subproblems, including extra credit points Solutions, either typed or handwritten Tables, diagrams, circuit schematics, or block diagrams
Your task is to identify and separate each exam problem into the listed components, including context, sub-questions, and solutions. At times, a subquestion can have nested subparts. Ignore any point values for any problem, question, or sub-question. Ignore any images, charts, or figures and do not attempt to extract text from them. If a provided image is not part of the problem in the pdf file and instead is part of another problem(s), omit it from the dictionary. Format your response so that it can be exported into a JSON file using the

template below. If the particular exam question lacks any of the listed components, omit them from the template.

Template for a problem which is split up into sub-problems:

```
{
    "problem": "1",
    "problem_context": <Insert any introductory paragraph or
        description exactly as it appears. If there is no context,
         d o n t  include this header>,
    "subproblems": [
    {
        "subproblem": "a" (Copy the part letter/number exactly as
            it appears on the exam),
        "subproblem_context": <Insert any introductory paragraph or
             description exactly as it appears. If there is no
             subproblem context or if the question is the only part
             of the subproblem,  d o n t  include this header. Replace
             all double quotes " here with escaped double quotes /">,
        "subproblem_question": <Insert the full question of the
             subproblem, exactly as it appears in the original.
             Replace all double quotes " here with escaped double
             quotes /">,
        "subproblem_solution": <Insert the full solution of the
             subproblem, exactly as shown in the original. Replace
             all double quotes " here with escaped double quotes /">
    },
    ...repeat as needed for additional subproblems within this
        problem
    ],
}
```

Template for a problem which is standalone and has no sub-problems:

```
{
    "problem": "1",
    "problem_context": <Insert any introductory paragraph or
        description exactly as it appears. If there is no context,
         d o n t  include this header. Replace all double quotes "
        here with escaped double quotes /">,
    "problem_question": <Insert the full question of the problem,
        exactly as it appears in the original. Replace all double
        quotes " here with escaped double quotes /">,
    "problem_solution": <Insert the full solution of the problem,
        exactly as shown in the original. Replace all double quotes
        " here with escaped double quotes /">
}
```

If any double quotes (") within strings appear within your JSON, you must replace them with escaped double quotes (/"). Return only valid JSON.

This is the prompt we use to extract the text for each question in our crowdsourced exam dataset. It ensures that problem statements, sub-questions, and solutions are consistently structured into JSON, while ignoring irrelevant formatting such as point values or images. By enforcing this schema, we can standardize raw exam PDFs into machine-readable data suitable for validation, benchmarking, and downstream analysis.

## D.9 LLM PROMPT FOR EXAM QUESTION IMAGE EXTRACTION

---

**Exam Question Image Extraction Prompt**

You are a language model assisting with the digitization of academic exam content in Computer Architecture.
**Input:**
You are provided with:
1) A PDF file containing one problem of an exam. If part of another problem is included, ignore this and only focus on the current pdf file.
2) A .json-styled txt file containing the problem's extracted text. It may contain some or all of the following empty-list fields: ``problem_context_figures", ``subproblem_context_figures", ``subproblem_solution_figures", and ``problem_solution_figures".
3) PNG images containing tables, diagrams, circuit schematics, or block diagrams that may or may not pertain to this problem. The names of the images provided are as follows, in order: {images}.

The given problem may be a standalone problem or consist of multiple sub-problems. Each problem or sub-problem may contain:
– Main context figures which are necessary to understanding the main problem, or ALL of the sub-problems.
– Sub-problem-specific figures that are separate from both the main problem context and the other sub-problems.
– Solutions to the main problem, which may be typed or handwritten.
– Solutions to a certain sub-problem, which may be typed or handwritten.

**Your task:**
Match each image file name (table, diagram, circuit schematic, or block diagram) to its correct association in the exam. Each image should have one of the four possible associations:
– The main problem question (``problem_context_figures")
– The sub-problem question (``subproblem_context_figures")
– The main problem solution (``problem_solution_figures")
– The sub-problem solution (``subproblem_solution_figures")
At times, the context or question in the main problem/sub-problem question/solution will include phrases (such as "The table below" or "The following diagram") that indicate a visual image falls under that category.
Your output should be a modified version of the given JSON, but with each of the figure fields populated with lists of the relevant image names (files ending in .png, .jpg, etc.), for example:
``problem_context_figures": [``image_name.png", ``image_name_2.png"]
No content, descriptions, or recreations of the image should be included in the output; only the file name should be included. If a provided image is not part of the current PDF file and instead is part of other problem(s), omit it. If the provided JSON includes image file names that are not present in the images provided, omit those too.
Be as precise as possible in your associations. Only include the dictionary; do not include reasoning. If none of the images given pertain to the problem, just output the original JSON given.

---

This is the prompt we use to associate figures with their correct roles in the digitized exam problems. By structuring images into categories such as problem context, subproblem context, problem solution, and subproblem solution, we can align visual information with text-based question content. This ensures that diagrams, tables, and schematics are consistently linked to their intended question or solution, enabling precise and reproducible dataset construction.

### D.10 LLM PROMPT FOR EXAM QUESTION VERIFICATION

---

**Exam Question Verification**

You are a reader tasked with verifying the accuracy of an automated document parser. A computer architecture exam PDF has been fed through the parser to produce standalone question–answer pairs in a certain JSON format, and you must compare the parsed questions to the original questions in the PDF.

**Input:** You are given (1) the original PDF of {problem_name} of the exam which may be broken down into subproblems, (2) the JSON dictionary produced by the parser, and (3) images that the parser has deemed are associated with the question and/or its subproblems. The names of the images provided are as follows, in order: {images}. Your task is to determine if the parser has correctly extracted the text and images while staying true to the original PDF.

**A problem is correct only if all of the following are satisfied:**
a) The extracted problem text is nearly identical, word-for-word, to the original PDF's text (special characters/math symbols may appear as unicode-escaped or equivalent).
b) The problem's ``question" and ``solution" fields are both populated. Exception: if the original solution is purely an image, that image must be correctly associated in ``solution_figures".
c) The problem is standalone: the ``context" and ``context_figures" provide all information needed, even if the original referenced prior problems.
d) No part of the solution is revealed in ``context" or ``context_figures".
e) All images are extracted/cropped correctly and categorized correctly as context vs. solution figures.
f) All tables are extracted correctly. If parsed as text instead of image, the table must be recreatable from the extracted text and usable to answer the question.
g) For fill-in-the-blank or fill-in-the-chart, the blank version must be provided in the question/context or ``context_figures".

**Output format:**
Return a modified version of the given JSON dictionary. Set each ``passed_llm_verification" field to `true` or `false` (unquoted). For every item marked `false`, add a ``reasoning" field explaining which conditions failed. Return exactly one dictionary and nothing else.

**Conservatism:** Err on the side of false. Avoid false positives; false negatives are acceptable.

---

This prompt is used to audit the parser's extracted question–answer pairs against the original exam source. It enforces strict, itemized criteria for textual fidelity, self-contained context, correct image/table extraction, and proper figure categorization, and it standardizes the verification output by toggling `passed_llm_verification` and adding concise `reasoning` where failures are detected.

### D.11 LLM PROMPT FOR MEMORY CONTROLLER DESIGN CASE STUDY

---

**Exam Question Verification**

You are a computer architecture expert specializing in DRAM memory controller design. You have been tasked with tuning the design of a custom DRAM memory controller to optimize for **POWER** (lower is better).

=== SEARCH BUDGET ===
You are on iteration `current_iteration` of 40.
Consider this when deciding your strategy:
- Early iterations: Consider exploring diverse configurations to understand the design space.
- Later iterations: Consider exploiting promising regions by making smaller refinements to the best configurations.

---

=== DRAM CONTROLLER DESIGN PARAMETERS ===
You control 10 DISCRETE design parameters, each represented as an INTEGER INDEX:
0: PagePolicy, Options: ["Open", "OpenAdaptive", "Closed", "ClosedAdaptive"]
1: Scheduler, Options: ["Fifo", "FrFcfsGrp", "FrFcfs"]
2: SchedulerBuffer, Options: ["Bankwise", "ReadWrite", "Shared"]
3: RequestBufferSize, Options: [1, 2, 4, 8, 16, 32, 64, 128]
4: RespQueue, Options: ["Fifo", "Reorder"]
5: RefreshPolicy, Options: ["NoRefresh", "AllBank"]
6: RefreshMaxPostponed, Options: [1,2,4,8]
7: RefreshMaxPulledin, Options: [1,2,4,8]
8: Arbiter, Options: ["Simple", "Fifo", "Reorder"]
9: MaxActiveTransactions, Options: [1, 2, 4, 8, 16, 32, 64, 128]

=== CONFIGURATIONS TRIED SO FAR ===
Here are the top 10 best configurations you've found so far, ranked by Power (W) (best first):
#1: [1.0, 2.0, 2.0, 1.0, 0.0, 0.0, 0.0, 0.0, 0.0, 0.0], Energy=0.000194 J, Power=1.697670 W, Latency=0.000114290 s
#2: [1.0, 2.0, 2.0, 2.0, 0.0, 0.0, 0.0, 0.0, 0.0, 1.0], Energy=0.000178 J, Power=2.092590 W, Latency=0.085218750 s
#3: [1.0, 2.0, 2.0, 3.0, 1.0, 0.0, 0.0, 0.0, 0.0, 2.0], Energy=0.000171 J, Power=2.408910 W, Latency=0.000070795 s
...

Here are the last 10 configurations you've tried:
#1: [1.0, 2.0, 2.0, 3.0, 1.0, 0.0, 0.0, 0.0, 0.0, 2.0]
#2: [1.0, 2.0, 2.0, 2.0, 0.0, 0.0, 0.0, 0.0, 0.0, 1.0]
#3: [1.0, 2.0, 2.0, 1.0, 0.0, 0.0, 0.0, 0.0, 0.0, 0.0]
...

IMPORTANT: You CANNOT propose a configuration that exactly matches any of the Top 10 or Last 10 configs you've tried listed above as proposing a duplicate configuration wastes an evaluation.

=== MEMORY TRACE SAMPLE ===
Below is a sample of the workload's memory trace (format: "timestamp: operation address"):
— Beginning (lines 0-29 of 10000) —
0: read 0x0
1: read 0x40
2: read 0x80
3: read 0xC0
4: read 0x100
...

— Middle (lines 4985-5014 of 10000) —
4985: read 0x190BBCC0
4986: read 0x26EEB740
4987: read 0x29CEE6C0
4988: read 0x3771780
4989: read 0x277DC1C0
...

— End (lines 9970-9999 of 10000) —
9970: read 0x4A880
9971: read 0x4A8C0
9972: read 0x4A900

```
9973: read 0x4A940
9974: read 0x4A980
...

=== YOUR TASK ===
Before proposing a new configuration, use your understanding of DRAM memory controller
architecture to reason through the following to make an informed choice:

1. **Memory Access Pattern Analysis**: Looking at the memory trace sample above, what
are key characteristics of this workload that you should consider when tuning the DRAM
memory controller?

2. **Parameter Impact Analysis**: Based on the trace sample and broader optimization
target (POWER) of yours, which parameters are most critical to tune and why?

3. **Learning from History**: Looking at previous configurations and their metrics, what
patterns do you observe? Which parameter changes led to improvements?

Based on your analysis from the THREE QUESTIONS above and the remaining search
budget, propose one NEW configuration that you expect will MINIMIZE POWER.

IMPORTANT OUTPUT FORMAT:
- First, provide your reasoning in a <reasoning>block
- Then, output ONLY a JSON list of exactly 10 integers (the indices 0..7 or 0..3 etc) on its
own line
- Do NOT include any extra text beyond the <reasoning>and JSON list of integer indices.

Example output format (enclosed in markdown code block):
```
<reasoning>
My analysis shows that ... therefore, I propose the following configuration:
</reasoning>
indices=[2, 0, 1, 3, 1, 0, 2, 1, 2, 4]
```
which in this example would correspond to the following DRAM design configuration:
- PagePolicy: "Closed" (index 2)
- Scheduler: "Fifo" (index 0)
- SchedulerBuffer: "ReadWrite" (index 1)
- RequestBufferSize: 8 (index 3)
- RespQueue: "Reorder" (index 1)
- RefreshPolicy: "NoRefresh" (index 0)
- RefreshMaxPostponed: 4 (index 2)
- RefreshMaxPulledin: 2 (index 1)
- Arbiter: "Reorder" (index 2)
- MaxActiveTransactions: 16 (index 4)
```

This prompt is used for the DRAM controller case study (Section 4.4). It tasks the LLM
with answering 3 questions and then proposing a new memory controller design configuration.
current_iteration is the i-th round of the 40 evaluation budget provided to the LLM. Thirty
lines of the workload's memory trace start, middle, and end is sampled to add to the prompt (ellipses
are used for brevity here).

