# OpenReview forum: "QuArch: A Benchmark for Evaluating LLM Reasoning in Computer Architecture"
_ICLR.cc/2026/Conference — Submitted to ICLR 2026_

### Official Review · Reviewer_ZBy2 · 2025-10-29

**Soundness:** 2
**Presentation:** 3
**Contribution:** 2
**Rating:** 2
**Confidence:** 5

**Summary:**

This paper introduces QUARCH, the first benchmark designed to evaluate the knowledge and reasoning capabilities of Large Language Models (LLMs) in the field of computer architecture. The benchmark contains 2,671 expert-validated question-answer (QA) pairs. These questions cover key areas like processor design, memory systems, and interconnection networks. The benchmark evaluation reveals that LLMs generally struggle with questions requiring higher-order thinking

**Strengths:**

1) This work presents an interesting benchmark for reasoning in computer architectures

2) This benchmark helps to demonstrate the shortcoming of LLMs in high-order thinking.

**Weaknesses:**

1) According to the sources of the benchmark, there is limited evidence that the QA tasks reflect the iterative, data-driven decision-making and trade-off analysis that human architects perform. This limits ecological validity — i.e., how well the benchmark reflects real-world reasoning in architectural design.

2) Despite the efforts, the benchmark’s scope is restricted to a relatively small set of expert-validated questions and may not scale to cover the full breadth of computer architectural domains.

3) The work argues that architectural reasoning evaluation will accelerate innovation in system design, yet provides no evidence or use cases showing how improved benchmark performance correlates with tangible advances in computer architecture research or practice.

**Questions:**

see the weaknesses.

---

> ### Author Response · Authors · 2025-11-25
> **Response to Reviewer ZBy2 [Part 1 of 2]**
>
> We thank the reviewer for engaging with our work and for highlighting the importance of evaluating higher-order reasoning in computer architecture. Below we address each concern directly.
>
> ### **Ecological Validity**
> >According to the sources of the benchmark, there is limited evidence that the QA tasks reflect the iterative, data-driven decision-making and trade-off analysis that human architects perform. This limits ecological validity — i.e., how well the benchmark reflects real-world reasoning in architectural design.
>
> QuArch is designed to evaluate the conceptual reasoning that underlies real architectural decision-making and basic blocks for real architecture designs. While the QA format abstracts away toolchain iteration, the content of the questions is directly drawn from the same artifacts used to train human architects, which includes university architecture exams and expert-authored assessments.
>
> These questions probe the core cognitive tasks that architects perform before implementation, such as:
> - Analyzing throughput of compute fabrics
> - Reasoning about cache and memory-system tradeoffs
> - Evaluating pipeline bottlenecks
> - Selecting architectural strategies under constraints
>
> We include examples such as (i) throughput analysis of a systolic array (Appendix C.1.2, Pages 26-27) and (ii) trade-off reasoning for multicore cache partitioning (Appendix C.1.3, Pages 27-28). Other questions in QuArch (Appendix C, Pages 26-52) also probe understanding architectural implications of software. These are the exact same problems that require the same analytical steps architects use in early-stage design and in design-review processes.
>
> Our results support that QuArch provides a **strong and discriminative** signal on LLMs’ architectural reasoning capabilities and serve as a well-scoped and grounded proxy for tractably measuring reasoning in architecture.
>
> Many benchmarks suffer from similar real-world limitations and abstract system complexity into smaller tasks that can serve as proxies and make evaluation tractable. Importantly, this does not undermine the benchmark’s value as many such benchmarks have driven meaningful progress. For example, VerilogEval [1] offers limited complexity when compared to the full complexity of industry-level designs, yet it has still helped push massive progress in LLMs for RTL code generation. Similarly, QuArch is the first benchmark of its kind for architecture reasoning and serves as an essential first step critical to bring about progress that offers structure, signal, and headroom for both model evaluation and future method development.
>
> ### **Benchmark Breath and Scalability**
> > Despite the efforts, the benchmark’s scope is restricted to a relatively small set of expert-validated questions and may not scale to cover the full breadth of computer architectural domains.
>
> Although no single benchmark can cover an entire field, QuArch provides unusually broad coverage for a domain-specific reasoning dataset:
>
> - 2,671 expert-validated QAs, exceeding or matching influential domain benchmarks (GPQA [2]: ~540; SWE-bench [3]: ~2,300; Table 1, Page 3 for more examples).
> - 13 significant architectural subdomains (Figure 4, Page 5), including processor architecture, memory systems, interconnects, and spatial/parallel architectures (e.g. GPUs), and modeling & simulation.
> - Both multiple-choice and free-response formats, spanning Recall, Analyze, Design, and Implement skills.
> - Both text-only and multimodal (i.e., image and text) question types spanning all skills.
>
> Our paper includes a detailed breakdown of our benchmark’s coverage (Section 2.3, Pages 5-6), which we believe is a strength: it provides transparency about what the dataset actually covers and enables targeted analysis of model strengths and weaknesses (Appendix Figure 14, Page 22). QuArch is already one of the most comprehensive benchmarks for domain-specific hardware reasoning.
>
> We emphasize that QuArch is extensible by design. Our public submission portal (Appendix B.1, Pages 17-21) is actively receiving new QA submissions from academic and industry researchers, and planned upcoming competitions will expand coverage. This ensures the benchmark grows alongside the field, similar to how LiveBench [4] evolves with model capabilities. As new architectural trends and techniques emerge, QuArch will evolve to reflect them as our infrastructure and methodology provides a scalable approach to ensure that QuArch can remain an enduring benchmark for the community.

---

> ### Author Response · Authors · 2025-11-25
> **Response to Reviewer ZBy2 [Part 2 of 2]**
>
> ### **Evidence Connecting Reasoning Performance to System-Level Impact**
>
> >The work argues that architectural reasoning evaluation will accelerate innovation in system design, yet provides no evidence or use cases showing how improved benchmark performance correlates with tangible advances in computer architecture research or practice.
>
> Our claim is not that high QuArch scores alone “design a better chip,” but that the reasoning skills QuArch measures are precisely the skills required by emerging agentic systems in architecture and code optimization. Recent work demonstrates this clearly:
> - Multi-agent systems such as ASTRA[5], MARCO[6], and ASIC-Agent[7] rely on a planner or orchestration agent that reasons about the best strategy to pursue to inform implementation.
> - GPU-kernel optimization systems (e.g., ASTRA, SwizzlePerf [8]) show that the quality of high-level reasoning about architectural principles (e.g., data reuse, vectorization, and memory hierarchy) directly governs end-to-end performance.
> - Recent AI-accelerated systems-research loops explicitly incorporate reasoning and analysis agents [9-10] and have already discovered systems algorithms that outperform human designs to accelerate innovation in system design.
>
> Importantly, QuArch reveals shortcomings with these higher-order reasoning skills: there is a clear gap between what LLMs *know* and how effectively they can *apply* that knowledge to inform architecture and systems-related decisions (Table 2). Moreover, by providing a skill framework, QuArch localizes model deficits to inform where the improvements need to be made. In this sense, QuArch provides both the diagnostic signal and the framework needed to guide progress toward more capable, reasoning-driven system design agents. Appendix Section C.2-C.5 provides detailed insights into LLM failure modes relevant for computer architecture with specific examples.
>
> We agree that connecting benchmark performance to real-world advances is an important point raised by the reviewer and will expand the discussion in our revision to explicitly connect reasoning performance to tangible advances in architectural design and optimization. The tangible impact of QuArch will emerge as future works adopt it as their evaluation benchmark just as other QA benchmarks (e.g., GPQA for physical sciences [2], GSM-8K for math [11], PubMedQA for medicine [12]) have done in their respective domains to catalyze progress.
>
> --------------
> ### **References**
>
> [1] N. Pinckney et al.,, “Revisiting VerilogEval: A Year of Improvements in Large-Language Models for Hardware Code Generation,” ACM Trans. Des. Autom. Electron. Syst., vol. 30, no. 6, p. 91:1-91:20, Oct. 2025, doi: 10.1145/3718088.
>
> [2] D. Rein et al., “GPQA: A Graduate-Level Google-Proof Q&A Benchmark,” Nov. 20, 2023, arXiv: arXiv:2311.12022. doi: 10.48550/arXiv.2311.12022.
>
> [3] C. E. Jimenez et al., “SWE-bench: Can Language Models Resolve Real-world Github Issues?,” presented at the The Twelfth International Conference on Learning Representations, Oct. 2023. Accessed: Nov. 20, 2025. [Online]. Available: https://openreview.net/forum?id=VTF8yNQM66
>
> [4] C. White et al., “LiveBench: A Challenging, Contamination-Limited LLM Benchmark,” Apr. 18, 2025, arXiv: arXiv:2406.19314. doi: 10.48550/arXiv.2406.19314.
>
> [5] A. Wei et al., “Astra: A Multi-Agent System for GPU Kernel Performance Optimization,” Sept. 09, 2025, arXiv: arXiv:2509.07506. doi: 10.48550/arXiv.2509.07506.
>
> [6] A. Rahman et al., “MARCO: Multi-Agent Code Optimization with Real-Time Knowledge Integration for High-Performance Computing,” June 25, 2025, arXiv: arXiv:2505.03906. doi: 10.48550/arXiv.2505.03906.
>
> [7] A. Allam et al., “ASIC-Agent: An Autonomous Multi-Agent System for ASIC Design with Benchmark Evaluation,” in 2025 IEEE International Conference on LLM-Aided Design (ICLAD), June 2025, pp. 23–29. doi: 10.1109/ICLAD65226.2025.00033.
>
> [8] A. Tschand et al., “SwizzlePerf: Hardware-Aware LLMs for GPU Kernel Performance Optimization,” Aug. 27, 2025, arXiv: arXiv:2508.20258. doi: 10.48550/arXiv.2508.20258.
>
> [9] A. Cheng et al., “Barbarians at the Gate: How AI is Upending Systems Research,” Oct. 10, 2025, arXiv: arXiv:2510.06189. doi: 10.48550/arXiv.2510.06189.
>
> [10] P. Hamadanian et al., “Glia: A Human-Inspired AI for Automated Systems Design and Optimization,” Nov. 17, 2025, arXiv: arXiv:2510.27176. doi: 10.48550/arXiv.2510.27176.
>
> [11] K. Cobbe et al., “Training Verifiers to Solve Math Word Problems,” Nov. 18, 2021, arXiv: arXiv:2110.14168. doi: 10.48550/arXiv.2110.14168.
>
> [12] Q. Jin et al., “PubMedQA: A Dataset for Biomedical Research Question Answering,” in Proceedings of the 2019 Conference on Empirical Methods in Natural Language Processing and the 9th International Joint Conference on Natural Language Processing (EMNLP-IJCNLP), K. Inui, J. Jiang, V. Ng, and X. Wan, Eds., Hong Kong, China: Association for Computational Linguistics, Nov. 2019, pp. 2567–2577. doi: 10.18653/v1/D19-1259.

---

### Official Review · Reviewer_ivvM · 2025-10-29

**Soundness:** 3
**Presentation:** 3
**Contribution:** 3
**Rating:** 6
**Confidence:** 2

**Summary:**

This paper proposes QuArch, a benchmark on assessing LLMs in designing computer architectures. This benchmark goes beyond the prevailing benchmarks in software design and contributes to bridging the gap in hardware designs. The authors construct and validate the dataset with experts, ensuring the dataset is of high quality. Using this dataset, the authors benchmark state-of-the-art models and reveal that although some models perform quite well on the benchmark, some others do not. They also analyze and provide insights on common behavior patterns of LLMs in this task.

**Strengths:**

1. The investigated problem is interesting: there are many datasets on software design, but it is also important to understand whether we can use LLMs in hardware design.

2. The dataset consists of more than 2k QA pairs validated by experts, providing a large scale testbed with high-quality data instances for researchers to understand and improve model behaviors.

3. The paper provides interesting insights for LLMs' common behavior patterns on this benchmark. These insights make it possible to understand/predict to some extent how LLMs will perform in related tasks. It also makes it easier for researchers to design remedies strategically to further improve LLM performance.

**Weaknesses:**

1. The head room of the benchmark does not seem to be very big. GPT-5 is already having accuracy of above 70 on the reasoning task and around 90 on the recall task. It seems to me that even if we can improve model performance, the space for improvement is quite small. Considering that benchmarks recently tend to be easily saturated with increasingly powerful models, I'm a bit worried about for how long the benchmark can be used.

2. Is there any method we can use to improve LLM performance in related tasks? It will be really useful if the authors can briefly discuss what methods an potentially be helpful, either in inference or training time.

**Questions:**

Authors are encouraged to address concerns above.

---

> ### Author Response · Authors · 2025-11-25
> **Response to Reviewer ivvM [Part 1 of 2]**
>
> We thank the reviewer for the positive feedback and for recognizing the contribution and insights of QuArch. We address your two concerns below.
>
> ### **Benchmark Headroom and Long-Term Utility**
>
> >The head room of the benchmark does not seem to be very big. GPT-5 is already having accuracy of above 70 on the reasoning task and around 90 on the recall task. It seems to me that even if we can improve model performance, the space for improvement is quite small. Considering that benchmarks recently tend to be easily saturated with increasingly powerful models, I'm a bit worried about for how long the benchmark can be used.
>
> QuArch remains a challenging benchmark and far from saturated, as it exposes substantial capability gaps among frontier models:
> - GPT-5 achieves 72% on reasoning, but the next-best frontier model is 10% lower (Table-2 Page-7), and notably the gap widens to 28% in the more advanced Design skill category (Table 3, Page 8).
>
> - A key insight from our benchmark is the persistent gap revealed between what models factually *know* about computer architecture (i.e., QuArch-Recall) and how they *apply* that knowledge (i.e., QuArch-Reasoning: analyze, design, implement). For example, there is a 17% gap between GPT-5’s recall vs reasoning abilities (Table 2, Page 7).
>
> - Despite high overall performance, GPT-5 scores only 62% on interconnection-network questions and shows other weaknesses across several architecture subdomains (Appendix Figure 14, Page 22).
>
> - We highlight key sub-domains frontier models need to further improve in with detailed failure-analysis (Appendix C.2-C.5, Pages 29-52).
>
> - Smaller open-source LLMs trail GPT-5 by large margins (Appendix Table 4, Page 23), offering substantial headroom for the research community. This headroom is important, as current works have asserted SLMs will be the drivers of future agentic systems [1].
>
> - Finally, QuArch-Reasoning remains more challenging than many widely used domain benchmarks (Figure 1, Page 2), such as GPQA (graduate-level physical sciences) [2] or AIME (mathematical problem solving) [3], both of which continue to yield meaningful signal and drive research advancements despite model advances.
>
> Moreover, QuArch is intentionally designed to be continuously updated through our public submission portal (Appendix B.1, Pages 17-21), which is actively receiving new QA submissions from researchers and industry contributors. This mirrors successful benchmarks that continuously evolve such as LiveBench [4]. Our methodology and pipeline make refreshing the dataset straightforward and sustainable.
>
> Together, these factors mean QuArch will provide meaningful differentiation and remain relevant even as models improve.

---

> ### Author Response · Authors · 2025-11-25
> **Response to Reviewer ivvM [Part 2 of 2]**
>
> ### **Methods for Improving LLM Performance on Hardware-Reasoning Tasks**
>
> >Is there any method we can use to improve LLM performance in related tasks? It will be really useful if the authors can briefly discuss what methods an potentially be helpful, either in inference or training time.
>
> We agree that discussing potential improvement strategies is valuable, and we summarize two actionable paths below.
>
> *(1) Training-time methods*
>
> - Prior work shows that even ~1K domain-specific QA pairs can significantly improve model reasoning [5]. QuArch provides 2,671 expert-vetted QAs, making it an effective SFT resource for architecture-specific reasoning.
>
> - A large subset of questions in QuArch include detailed rationales. Models can be trained on these rationales to produce correct intermediate reasoning, not just answers. This aligns with recent approaches [6] that reinforce chain-of-thought fidelity and leads to more reliable system-level reasoning.
>
> *(2) Inference-time methods*
>
> - Many agentic hardware design systems (e.g., ASTRA [7], MARCO [8], ASIC-Agent [9], Adhrafi et al. [10], Pamnani et al. [11]) rely on a planner agent that must reason about systems properties. High-quality retrieval from QuArch’s reasoning QAs can strengthen such planners.
>
> - Systems-related tasks like GPU kernel optimization (e.g., in ASTRA or SwizzlePerf [12]) has been shown to depend heavily on accurate architectural reasoning. Improving performance on QuArch-Reasoning directly strengthens the upstream decision-making these workflows rely on.
>
> QuArch is therefore not only diagnostic but also a practical foundation for effectively improving systems-oriented reasoning in LLMs for related tasks.
>
> --------------
> ### **References**
>
> [1] P. Belcak et al., “Small Language Models are the Future of Agentic AI,” Sept. 15, 2025, arXiv: arXiv:2506.02153. doi: 10.48550/arXiv.2506.02153.
>
> [2] D. Rein et al., “GPQA: A Graduate-Level Google-Proof Q&A Benchmark,” Nov. 20, 2023, arXiv: arXiv:2311.12022. doi: 10.48550/arXiv.2311.12022.
>
> [3] “opencompass/AIME2025 · Datasets at Hugging Face.” Accessed: Nov. 20, 2025. [Online]. Available: https://huggingface.co/datasets/opencompass/AIME2025
>
> [4] C. White et al., “LiveBench: A Challenging, Contamination-Limited LLM Benchmark,” Apr. 18, 2025, arXiv: arXiv:2406.19314. doi: 10.48550/arXiv.2406.19314.
>
> [5] N. Muennighoff et al., “s1: Simple test-time scaling,” in Proceedings of the 2025 Conference on Empirical Methods in Natural Language Processing, C. Christodoulopoulos, T. Chakraborty, C. Rose, and V. Peng, Eds., Suzhou, China: Association for Computational Linguistics, Nov. 2025, pp. 20286–20332. doi: 10.18653/v1/2025.emnlp-main.1025.
>
> [6] H. Lightman et al., “Let’s Verify Step by Step,” May 31, 2023, arXiv: arXiv:2305.20050. doi: 10.48550/arXiv.2305.20050.
>
> [7] A. Wei et al., “Astra: A Multi-Agent System for GPU Kernel Performance Optimization,” Sept. 09, 2025, arXiv: arXiv:2509.07506. doi: 10.48550/arXiv.2509.07506.
>
> [8] A. Rahman et al., “MARCO: Multi-Agent Code Optimization with Real-Time Knowledge Integration for High-Performance Computing,” June 25, 2025, arXiv: arXiv:2505.03906. doi: 10.48550/arXiv.2505.03906.
>
> [9] A. Allam, Y. Mansour, and M. Shalan, “ASIC-Agent: An Autonomous Multi-Agent System for ASIC Design with Benchmark Evaluation,” in 2025 IEEE International Conference on LLM-Aided Design (ICLAD), June 2025, pp. 23–29. doi: 10.1109/ICLAD65226.2025.00033.
>
> [10] N. Ashrafi, S. Bouktif, and M. Mediani, “Enhancing LLM Code Generation: A Systematic Evaluation of Multi-Agent Collaboration and Runtime Debugging for Improved Accuracy, Reliability, and Latency,” May 04, 2025, arXiv: arXiv:2505.02133. doi: 10.48550/arXiv.2505.02133.
>
> [11] C. R. Pamnani, “AI-Driven Automation for Digital Hardware Design: A Multi-Agent Generative Approach,” in Proceedings of the 2025 4th International Conference on Frontiers of Artificial Intelligence and Machine Learning, Shenyang China: ACM, Apr. 2025, pp. 26–30. doi: 10.1145/3748382.3748388.
>
> [12] A. Tschand et al., “SwizzlePerf: Hardware-Aware LLMs for GPU Kernel Performance Optimization,” Aug. 27, 2025, arXiv: arXiv:2508.20258. doi: 10.48550/arXiv.2508.20258.

---

### Official Review · Reviewer_YovW · 2025-10-31

**Soundness:** 3
**Presentation:** 3
**Contribution:** 2
**Rating:** 4
**Confidence:** 3

**Summary:**

This paper introduces QuArch, the first benchmark specifically designed to evaluate the reasoning capabilities of LLMs in the field of computer architecture. The authors address the absence of specialized evaluation tools in this domain by creating a comprehensive dataset of 2,671 expert-validated question-answer pairs. Through the evaluation of ten frontier LLMs, the study reveals that these models exhibit considerable weaknesses in higher-order reasoning tasks such as analysis, design, and implementation.

**Strengths:**

1. This work addresses the lack of LLM evaluation benchmarks in computer architecture. The inclusion of expert validation for all 2,671 question-answer pairs lends credibility to the dataset's quality and technical accuracy.
2. The authors evaluate a wide range of frontier LLMs.

**Weaknesses:**

The "Implement" skill is tested by asking models to produce artifacts like code or simulation scripts. This is a step in the right direction, but it's a simplified version of real-world implementation, which involves complex toolchains, debugging, verification, and performance tuning that cannot be fully captured in this format.

**Questions:**

The expert validation is a key strength. Could you elaborate on the inter-rater reliability among the experts who validated the QAs? Were there specific topics or question types that caused more disagreement?

---

> ### Author Response · Authors · 2025-11-24
> **Response to Reviewer YovW [Part 1 of 2]**
>
> We thank the reviewer for the feedback and for recognizing both the need for a domain-specific benchmark and the rigor of our expert-validated dataset. Below we directly address your questions and concerns.
>
> ### **Expert Validation and Inter-Rater Reliability**
>
> >The expert validation is a key strength. Could you elaborate on the inter-rater reliability among the experts who validated the QAs? Were there specific topics or question types that caused more disagreement?
>
> In QuArch, we apply expert validation in two areas: (1) expert validation of multiple-choice and free-response QAs for admission into QuArch, and (2) verifying that human judging is comparable to LLM-as-a-judge for grading LLM-generated answers.
>
> *(1) Inter-rater reliability for the accepted questions*
>
> All multiple choice questions (MCQs) included in QuArch were retained only when all three expert reviewers unanimously agreed on question quality and correctness. Rejected questions were discarded, so the final set contains no disagreements.
>
> For free form questions (FRQs), crowdsourced questions were similarly only accepted after unanimous correctness checks by three experts. Questions from university architecture exams were authored by faculty/teaching staff and thus were validated by the original exam creators.
>
> *(2) Human vs. LLM-as-a-judge grading*
>
> Expert humans saw 90.7% agreement in response grading. We evaluated human-human agreement using 100 randomly selected FRQs answered by models (908 total LLM responses). Of these, 84 responses from 35 questions required a third expert for adjudication.
>
> Regarding per-topic human-human disagreements, we find the following disagreement rates per LLM response across the top 3 topic categories:
> - Processor Architecture: 5.9% (22 / 373)
> - Memory Systems: 14.7% (29 / 197)
> - GPU/Parallel/Spatial Architectures: 9.2% (18 / 196)
>
> We interestingly observe more disagreement when grading questions related to memory systems; however, a larger sample size is required to validate this trend. Rather than particular topic subject matter, experts reported that disagreements were usually due to mechanical effort (e.g., verifying long tables or Boolean expressions).

---

> ### Author Response · Authors · 2025-11-24
> **Response to Reviewer YovW [Part 2 of 2]**
>
> ### **Scope of Implementation and Real-World Complexity**
>
> >The "Implement" skill is tested by asking models to produce artifacts like code or simulation scripts. This is a step in the right direction, but it's a simplified version of real-world implementation, which involves complex toolchains, debugging, verification, and performance tuning that cannot be fully captured in this format.
>
> We agree that real-world implementation in architecture could involve toolchains, debugging, verification, and performance tuning. Naturally, many current benchmarks focus on this dimension for this reason (e.g., emphasizing code generation and measuring success through compilable output or pass/fail test cases). We acknowledge these activities are not easily measured in a QA-style benchmark.
>
> Our goal is *not* to replicate full workflows, but to evaluate the reasoning stages that precede and enable implementation, which are currently under-examined.
>
> QuArch therefore includes simplified implementation proxies (e.g., Debugging: Appendix C.2.4, Pages 36-37 and Appendix C.5.2, Pages 50-51; Performance tuning: Appendix C.2.1, Pages 29-31 and Appendix C.2.3, Pages 33-35) while intentionally emphasizing upstream skills that are essential precursors to real implementation and where frontier models also show great weaknesses (See Table 3, Page 8).
>
> This division is consistent with trends in LLM code benchmarks, where reasoning-focused tasks are separated from toolchain-heavy agentic evaluations [1-3]. We view agentic implementation-focused benchmarks [4-5] as complementary, not overlapping. Our benchmark is designed to interoperate with such efforts: QuArch provides the reasoning substrate necessary for those downstream efforts.
>
> ### **Contribution Clarification**
>
> We believe QuArch fills a foundational gap and will help guide future work on agentic approaches for hardware design by providing holistic evaluation of an LLM’s system-level competence. QuArch is the first benchmark that systematically evaluates reasoning in computer architecture and systems across four skill types (Recall, Analyze, Design, and Implement). By disentangling these skills and providing expert-validated ground truth answers, it establishes a principled framework for localizing LLM capabilities and weaknesses that current toolchain-focused benchmarks fail to explicitly provide. QuArch currently contains 2.6K verified QAs, which is comparable to highly impactful benchmarks across the computing stack (Table 1, Page 3).
>
> --------------
> ### **References**
>
> [1] E. Dinella, S. Chandra, and P. Maniatis, “CRQBench: A Benchmark of Code Reasoning Questions,” Aug. 15, 2024, arXiv: arXiv:2408.08453. doi: 10.48550/arXiv.2408.08453.
>
> [2] A. Gu, B. Roziere, H. J. Leather, A. Solar-Lezama, G. Synnaeve, and S. Wang, “CRUXEval: A Benchmark for Code Reasoning, Understanding and Execution,” presented at the Forty-first International Conference on Machine Learning, June 2024. Accessed: Nov. 20, 2025. [Online]. Available: https://openreview.net/forum?id=Ffpg52swvg
>
> [3] D. M. Nguyen et al., “CodeMMLU: A Multi-Task Benchmark for Assessing Code Understanding & Reasoning Capabilities of CodeLLMs,” presented at the The Thirteenth International Conference on Learning Representations, Oct. 2024. Accessed: Nov. 20, 2025. [Online]. Available: https://openreview.net/forum?id=CahIEKCu5Q
>
> [4] A. Allam, Y. Mansour, and M. Shalan, “ASIC-Agent: An Autonomous Multi-Agent System for ASIC Design with Benchmark Evaluation,” in 2025 IEEE International Conference on LLM-Aided Design (ICLAD), June 2025, pp. 23–29. doi: 10.1109/ICLAD65226.2025.00033.
>
> [5] N. Pinckney et al., “Comprehensive Verilog Design Problems: A Next-Generation Benchmark Dataset for Evaluating Large Language Models and Agents on RTL Design and Verification,” June 17, 2025, arXiv: arXiv:2506.14074. doi: 10.48550/arXiv.2506.14074.

---

### Author Response · Authors · 2025-11-25
**General Response to Area Chairs & All Reviewers [Part 1 of 2]**

Dear Area Chairs and Reviewers,

We would like to thank everyone for their time and constructive feedback on our paper. We have provided a general response to common concerns that we would like to address below.
We have also addressed each reviewer’s comments individually and look forward to a productive discussion during the author response phase.

### **Significance of This Work**

QuArch contributes the following core advances to LLM evaluation in the systems domain:
- Addresses a missing layer by focusing on computer architecture, an area largely ignored in current benchmark evaluations across the systems stack (Table 1, Page 3; Section 5, Page 9).
- Offers expert-vetted, multimodal, and complex free-response questions to offer a reasoning-focused benchmark in the broader hardware domain (Section 2.2, Pages 4-5).
- Provides an organized taxonomy to study model reasoning and systematically identify failure modes (Figure 5, Page 6; Appendix Figure 14, Page 22; Appendix C.2-5, Pages 29-51).
- Exposes weaknesses that traditional code-generation benchmarks (e.g., Python or Verilog generation) fail to capture. Our analysis shows that even frontier models struggle with higher-order reasoning required in systems tasks and details specific LLM limitations relevant for computer architecture with concrete examples (Section 4.3, Pages 7-8; Appendix C.2-5, Pages 29-51).

QuArch is the first step toward holistic evaluation of LLM agents for hardware and systems tasks, complementary to implementation-focused efforts like VerilogEval [1] or SWE-bench [2].

### **Representative QAs**

QuArch’s QAs are not only high-quality and expert-validated, but they are reflective of the conceptual skills and reasoning that are at the foundation of real-world computer architecture design. While Figure 2 (Page 4) was meant to illustrate our skill framework, it did not sufficiently capture the richness of the questions themselves. Although several detailed examples appear in the Appendix C (Pages 26–52), we recognize based on reviewer feedback that more representative QAs should have been included in the main paper and will revise accordingly. Importantly, the questions (drawn from domain experts, rigorous academic exams, and conference competitions) reflect how architects engage with systems: not just by writing code, but by making inferences from diagrams, analyzing hardware-software interactions, and designing strategies to improve system performance.

For example, QA #1 below includes a block diagram and challenges the model to conduct architectural analysis of a systolic array to determine the required throughput.

> **QuArch QA #1 (Appendix C.1.2, Pages 26-27)**
> ```
> ASCII-rendition of context figure (see Appendix for actual image of block diagram):
>
>
> W  ---> +-----------+ ---> +-----------+ ---> ... ---> +-----------+
>         |    PE     |      |    PE     |               |    PE     |
> X  ---> |           | ---> |           | ---> ... ---> |           |
>         +-----+-----+      +-----+-----+               +-----+-----+
>               |                  |                           |
>               v                  v                           v
>              y_1                y_2                         y_i
> ```
> **Context:**
>
> Given arrays X = [x_1, x_2, x_3, . . . , x_n] and W = [w_1, w_2, w_3, ... , w_k], you want to compute Y = [y_1, y_2, y_3, . . . , y_(n + 1 − k)] using the formula:
>
> y_i = w_1×i + w_2×(i + 1) + w_3×(i + 2) + · · · + w_k×(i + k − 1)
>
> The figure shows a systolic array of processing elements (PEs) and their input-output behavior.
>
> **Question:** What should be the relative speeds at which X and W values flow through the array to end up with the correct result for all Y values?
>
> **Correct Solution:** 2:1

---

> ### Author Response · Authors · 2025-11-25
> **General Response to Area Chairs & All Reviewers [Part 2 of 2]**
>
> *(continuation of previous comment)*
>
> QA #2 below asks for reasoning about system-level trade-offs and different architectural strategies for cache partitioning in multicore systems.
> >**QuArch QA #2 (Appendix C.1.3, Pages 27-28)**
> >
> >**Context:** Suppose we have a system with 32 cores that share a physical second-level cache. Assume each core is running one single-threaded application, and all 32 cores are concurrently running applications. Assume that the page size of the architecture is 8KB, the block size of the cache is 128 bytes, and the cache uses LRU replacement. We would like to ensure each application gets a dedicated space in this shared cache without any interference from other cores. We would like to enforce this using the utility based cache partitioning (UCP) to partition the cache. Assume we would like to design a 4MB cache with a 128-byte block size. Recall that UCP aims to minimize the cache miss rate by allocating more cache ways to applications that obtain the most benefit from more ways, as we discussed in lecture.
> >
> >**Question:** Consider the maximum associativity of the cache such that each application is guaranteed a minimum amount of space without interference. Is it desirable to implement UCP on a cache with this maximum associativity? Why, why not? Explain.
> >
> >**Correct Solution:** No, it is not desirable to implement UCP with this maximum associativity because the overhead of UCP for 32 applications on this cache will likely outweigh its benefits. UCP will only work with LRU replacement policy. But implementing LRU on top of a 32 k-way cache is impractical. Also the number of counters needed by UCP and the partitioning solution space for UCP are very large for such a cache.
>
> Other questions in the Appendix probe understanding the architectural implications of software. Appendix C.2.3 (Pages 33-35) asks about the impact of warp divergence on GPU SIMD utilization, and Appendix C.4.1 (Pages 43-46) asks about the implications of cache coherence policy in state transitions of instructions. Multimodal questions are also included. Appendix C.5.1 (Pages 48-49) asks models to complete a step of Tomasulo’s Algorithm [3] in the reservation station table, and Appendix C.5.2 (Pages 50-51) asks them to find the min-cut that eliminates deadlocks in a channel dependency graph. **Notably, we observe multiple frontier models get these questions incorrect.**
>
> **We believe these examples illustrate how QuArch assesses the technical reasoning skills essential to architectural design and practice.** We will update the main draft to directly include a more representative set of real questions, which will better highlight the benchmark’s depth and diversity, practical relevance, and what it means to perform well on the benchmark.
>
> --------------
> ### **References**
>
> [1] N. Pinckney, C. Batten, M. Liu, H. Ren, and B. Khailany, “Revisiting VerilogEval: A Year of Improvements in Large-Language Models for Hardware Code Generation,” ACM Trans. Des. Autom. Electron. Syst., vol. 30, no. 6, p. 91:1-91:20, Oct. 2025, doi: 10.1145/3718088.
>
> [2] C. E. Jimenez et al., “SWE-bench: Can Language Models Resolve Real-world Github Issues?,” presented at the The Twelfth International Conference on Learning Representations, Oct. 2023. Accessed: Nov. 20, 2025. [Online]. Available: https://openreview.net/forum?id=VTF8yNQM66
>
> [3] J. L. Hennessy and D. A. Patterson, Computer Architecture: A Quantitative Approach. Elsevier, 2011.

---

### Author Response · Authors · 2025-12-03
**Summary of Rebuttal Discussion for Area Chair**

Dear Area Chair,

We thank the reviewers for their thoughtful feedback and questions. To assist with changes made to the review process, we provide a summary of (1) our work’s contributions and (2) our responses to reviewer concerns. We've linked the detailed responses for convenience.

## **Summary of Contributions**
QuArch contributes the following core advances to LLM evaluation in the systems domain:
* Addresses a missing layer by focusing on computer architecture, an area ignored in current benchmark evaluations across the systems stack (Table 1, Page 3; Section 5, Page 10).
* Offers expert-vetted, multimodal, complex free-response questions to form a reasoning-focused benchmark in the broader hardware domain (Section 2.2, Pages 4-5).
* Provides an organized taxonomy to study model reasoning and systematically identify failure modes (Figure 5, Page 6; Appendix Figure 16, Page 24; Appendix C.2-5, Pages 31-53).
* Exposes weaknesses that traditional code-generation benchmarks (e.g., Python or Verilog generation) fail to capture. Our analysis shows that even frontier models struggle with higher-order reasoning required in systems tasks and details specific LLM limitations relevant for computer architecture with concrete examples (Section 4.3, Pages 8-9; Appendix C.2-5, Pages 31-53).

QuArch is the first step toward holistic evaluation of LLM agents for hardware and systems tasks, **complementary to implementation-focused efforts like VerilogEval \[1\] or SWE-bench \[2\].**

## **Summary of Responses**
| Reviewer Concern | Response Summary |
| --- | --- |
| [Evidence Connecting Reasoning Performance to System-Level Impact](https://openreview.net/forum?id=nhcz0uni55&noteId=I6KLxaIlNO) (Reviewer ZBy2) | We added a new case study (Section 4.4 & Figure 7, Page 9\) showing **QuArch-Reasoning scores correlate with downstream memory controller design quality**. This canonical design space exploration task requires architectural reasoning to arrive at optimal designs. We also provide evidence from work that explicitly incorporating reasoning agents in multi-agent loops accelerates innovation in system design. |
| [Ecological Validity](https://openreview.net/forum?id=nhcz0uni55&noteId=cGdttjzml6) (Reviewer ZBy2) | QuArch is designed to evaluate the conceptual reasoning that underlies real architectural decision-making and design. **The questions are directly drawn from the same artifacts used to train human architects**, which include university architecture exams and expert-authored assessments. **We’ve added a section with concrete examples from QuArch (Section 2.4 & Figure 6, Page 6\) to illustrate this,** which were previously left in the Appendix. |
| [Benchmark Breadth and Scalability](https://openreview.net/forum?id=nhcz0uni55&noteId=cGdttjzml6) (Reviewer ZBy2) | **QuArch provides unusually broad coverage for a domain-specific reasoning dataset,** containing 2,671 validated QAs (comparable to or larger than influential domain-specific benchmarks) across 13 architecture subdomains with multiple modalities and response formats. We emphasize that **QuArch is extensible by design**: Our public submission portal (Appendix B.1, Pages 19-23) is actively receiving new QA submissions from academic and industry researchers to ensure that QuArch can remain an enduring benchmark for the community. |
| [Benchmark Headroom](https://openreview.net/forum?id=nhcz0uni55&noteId=K8iTO4C13k) (Reviewer ivvM) | GPT-5 achieves 72% on QuArch-Reasoning, but other **SoTA models lag significantly (achieving \<60%)**. Smaller open models leave even more headroom. Even within GPT-5, **QuArch exposes significant subdomain-specific weaknesses** (e.g., interconnects, cache design), leaving opportunity for topic-specific improvement. |
| [Potential Methods to Improve Performance](https://openreview.net/forum?id=nhcz0uni55&noteId=pvfyq7Ysba) (Reviewer ivvM) | We specified **actionable paths for systems-related task improvement using QuArch**. We propose improving models during both training (SFT on QuArch answers and rationales) and inference time (e.g., RAG or planner agent integration). |
| [Implementation Skill Simplification](https://openreview.net/forum?id=nhcz0uni55&noteId=7HIZlO6ED3) (Reviewer YovW) | **QuArch targets upstream reasoning**, not full implementation. We acknowledge limitations of QA format, but assert that QuArch complements toolchain-heavy benchmarks like VerilogEval \[1\] and helps characterize model readiness for higher-level tasks. We added a new experimental case study (Section 4.4, Page 9\) showing **QuArch-Reasoning scores correlate with downstream architectural tasks**. |
| [Expert Validation Details](https://openreview.net/forum?id=nhcz0uni55&noteId=COpi5mA00X) (Reviewer YovW) | We clarify that all questions in QuArch were unanimously validated by experts. For LLM-as-a-judge validation **we provided additional inter-rater reliability analysis and per-topic disagreement rates across expert annotators.** |

---

> ### Author Response · Authors · 2025-12-03
> **Summary of Rebuttal Discussion for Area Chair (continued)**
>
> ### **Final Remarks**
> Reviewers acknowledged the significance of QuArch as an interesting and timely benchmark for evaluating LLMs in computer architecture, citing its expert-validated dataset, broad coverage of frontier models, and insights into higher-order reasoning and common LLM failure patterns. The main concern centered on real-world utility, which our response addresses through a new downstream case study (Section 4.4 & Figure 7, Page 9), included examples (Section 2.4 & Figure 6, Page 6), and clearer positioning. We believe these additions substantially strengthen the paper and clarify its contributions. We also responded comprehensively to all reviewer concerns to clarify ecological validity and benchmark headroom as well as outline actionable improvement methods. We thank the reviewers and area chair for their thoughtful feedback and time.
>
> Kind Regards,
>
> The Authors
>
> ---
>
> ### **References**
> \[1\] N. Pinckney, C. Batten, M. Liu, H. Ren, and B. Khailany, “Revisiting VerilogEval: A Year of Improvements in Large-Language Models for Hardware Code Generation,” ACM Trans. Des. Autom. Electron. Syst., vol. 30, no. 6, p. 91:1-91:20, Oct. 2025, doi: 10.1145/3718088.
>
> \[2\] C. E. Jimenez et al., “SWE-bench: Can Language Models Resolve Real-world Github Issues?,” presented at the The Twelfth International Conference on Learning Representations, Oct. 2023\. Accessed: Nov. 20, 2025\. \[Online\]. Available: https://openreview.net/forum?id=VTF8yNQM66

---

### Meta-Review · Area_Chair_EtJB · 2026-01-07

**Summary:**

This submission proposes QuArch, a benchmark for evaluating LLM reasoning in computer architecture, with expert-validated QA pairs and an extensive empirical analysis. Reviewers agree the topic is timely and the dataset is carefully constructed, and they appreciate the effort in expert validation and coverage. However, significant concerns remain regarding ecological validity, benchmark headroom, scalability, and the strength of evidence connecting benchmark performance to real-world architectural impact. While the authors provided a detailed rebuttal and added clarifications and case studies, none of the reviewers explicitly updated their assessments or scores. Therefore, the recommendation is reject.

**Reviewer Concerns:**

Concerns largely addressed by the rebuttal:

1. The lack of discussion on *how to improve LLM performance* (Reviewer ivvM): the authors clearly outlined concrete training-time and inference-time strategies, strengthening the paper’s practical guidance.
2. Benchmark headroom (Reviewer ivvM): additional analyses and breakdowns across skills and subdomains helped clarify that non-trivial headroom remains, especially for reasoning and design tasks.
3. Expert validation details (Reviewer YovW): the rebuttal provided clearer explanations of inter-rater reliability and validation procedures.

Concerns that remain outstanding:

1. Ecological validity (Reviewer ZBy2): despite added examples and a downstream case study, it remains unclear whether the QA-based benchmark adequately captures the iterative, trade-off–driven reasoning of real-world architectural design.
2. Connection to real-world impact (Reviewer ZBy2): the evidence linking improved benchmark performance to tangible advances in computer architecture practice is still indirect and largely speculative.
3. Long-term relevance and scalability (Reviewer ZBy2): while extensibility is claimed, concrete evidence that the benchmark can scale to cover the full breadth of evolving architectural domains remains limited.

**Reviewer Scores:**

Reviewer YovW (4, confidence 3): Likely to remain at  4. While clarification on expert validation and positioning helps, the core concern about simplified implementation remains, and the rebuttal does not fundamentally change the reviewer’s borderline stance.

Reviewer ivvM (6, confidence 2): Might stay at 6, or increase slightly. The rebuttal directly addressed headroom and improvement methods, but given the reviewer’s low confidence and already marginally positive score, a significant upward revision is unlikely.

Reviewer ZBy2 (2, confidence 5): Unlikely to change. Despite added case studies and clarifications, the rebuttal does not fully resolve the reviewer’s fundamental concerns about ecological validity and real-world impact, which appear to be principled objections rather than missing details.

---

### Decision · Program_Chairs · 2026-01-26

Reject